# Distributed Personalized Empirical Risk Minimization

**Yuyang Deng**
Pennsylvania State University
yzd82@psu.edu

**Mohammad Mahdi Kamani**
Wyze Labs
mmkamani@alumni.psu.edu

**Pouria Mahdavinia**
Pennsylvania State University
pxm5426@psu.edu

**Mehrdad Mahdavi**
Pennsylvania State University
mzm616@psu.edu

## Abstract

This paper advocates a new paradigm Personalized Empirical Risk Minimization (PERM) to facilitate learning from heterogeneous data sources without imposing stringent constraints on computational resources shared by participating devices. In PERM, we aim to learn a distinct model for each client by learning who to learn with and personalizing the aggregation of local empirical losses by effectively estimating the statistical discrepancy among data distributions, which entails optimal statistical accuracy for all local distributions and overcomes the data heterogeneity issue. To learn personalized models at scale, we propose a distributed algorithm that replaces the standard model averaging with model shuffling to simultaneously optimize PERM objectives for all devices. This also allows us to learn distinct model architectures (e.g., neural networks with different numbers of parameters) for different clients, thus confining underlying memory and compute resources of individual clients. We rigorously analyze the convergence of the proposed algorithm and conduct experiments that corroborate the effectiveness of the proposed paradigm.

## 1 Introduction

Recently *federated learning* (FL) has emerged as an alternative paradigm to centralized learning to encourage federated model sharing and create a framework to support edge intelligence by shifting model training and inference from data centers to potentially scattered—and perhaps self-interested— systems where data is generated [1]. While undoubtedly being a better paradigm than centralized learning, enabling the widespread adoption of FL necessitates foundational advances in the efficient use of statistical and computational resources to encourage a large pool of individuals or corporations to share their private data and resources. Specifically, due to *heterogeneity* of data and compute resources among participants, it is necessary, if not imperative, to develop distributed algorithms that are i) cognizant of statistical heterogeneity (*data-awareness*) by designing algorithms that effectively deal with highly heterogeneous data distributions across devices; and ii) confined to learning models that meet available computational resources of participant devices (*system-awareness*).

To mitigate the negative effect of data heterogeneity (non-IIDness), two common approaches are clustering and personalization. The key idea behind the clustering-based methods [2, 3, 4, 5] is to partition the devices into clusters (coalitions) of similar data distributions and then learn a single shared model for all clients within each cluster. While appealing, the partitioning methods are limited to heuristic ideas such as clustering based on the geographical distribution of devices without taking the actual data distributions into account and lack theoretical guarantees or postulate strong assumptions on initial models or data distributions [4, 5]. In personalization-based methods [6, 7, 8, 2, 9, 10, 11], the idea is to learn a distinct *personalized model* for each device alongside the global model, which

37th Conference on Neural Information Processing Systems (NeurIPS 2023).

can be unified as minimizing a bi-level optimization problem [12]. Personalization aims to learn a model that has the generalization capabilities of the global model but can also perform well on the specific data distribution of each participant suffers from a few key limitations. First, as the number of clients grows, while the number of training data increases, the number of parameters to be learned increases which limits to increase in the number of clients beyond a certain point to balance data and overall model complexity tradeoff– a phenomenon known as incidental parameters problem [13]. Moreover, since the knowledge transfer among data sources happens through a single global model, it might lead to suboptimal results. To see this, consider an extreme example, where half of the users have identical data distributions, say $\mathcal{D}$, while the other half share a data distribution that is substantially different, say $-\mathcal{D}$ (e.g. two distributions with same marginal distribution on features but opposite labeling functions). In this case, the global model obtained by naively aggregating local models (e.g., fixed mixture weights) converges to a solution that suffers from low test accuracy on all local distributions which makes it preferable to learn a model for each client solely based on its local data or carefully chosen subset of data sources.

Focusing on system heterogeneity, most existing works require learning models of identical architecture to be deployed across the clients and server (model-homogeneity) [14, 15], and mostly focus on reducing number [15] or size [16, 17, 18] of communications or sampling handling chaotic availability of clients [19, 20]. The requirement of the same model makes it infeasible to train large models due to *system heterogeneity* where client devices have drastically different computational resources. A few recent studies aim to overcome this issue either by leveraging knowledge distillation methods [21, 22, 23, 24] or partial training (PT) strategies via model subsampling (either static [25, 26], random [27], or rolling [28]). However, KD-based methods require having access to a public representative dataset of all local datasets at server and ignore data heterogeneity in the distillation stage to a large extent. The focus of PT training methods is mostly on learning a single server model using heterogeneous resources of devices and does not aim at deploying a model onto each client after the global server model is trained (which is left as a future direction in [28]). The aforementioned issues lead to a fundamental question: *"What is the best strategy to learn from heterogeneous data sources to achieve optimal accuracy w.r.t. each data source, without imposing stringent constraints on computational resources shared by participating devices?"*.

We answer this question affirmatively, by proposing a new *data&system-aware* paradigm dubbed Personalized Empirical Risk Minimization (PERM), to facilitate learning from massively fragmented private data under resource constraints. Motivated by generalization bounds in multiple source domain adaptation [29, 30, 31, 32], in PERM we aim to learn a distinct model for each client by *personalizing the aggregation* of empirical losses of different data sources which enables each client *to learn who to learn with* using an effective method to empirically estimate the statistical discrepancy between their associated data distributions. We argue that PERM entails optimal statistical accuracy for all local distributions, thus overcoming the data heterogeneity issue. PERM can also be employed in other learning settings with multiple heterogeneous sources of data such as domain adaptation and multi-task learning to entail optimal statistical accuracy. While PERM overcomes the data heterogeneity issue, the number of optimization problems (i.e., distinct personalized ERMs) to be solved scales linearly with the number of data sources. To simultaneously optimize all objectives in a scalable and computationally efficient manner, we propose a novel idea that replaces the standard *model averaging* in distributed learning with *model shuffling* and establish its convergence rate. This also allows us to learn distinct model architectures (e.g., neural networks with different number of parameters) for different clients, thus confining to underlying memory and compute resources of individual clients, and overcoming the system heterogeneity issue. This addresses an open question in [28] where only a single global model can be trained in a model-heterogeneous setting, while PERM allows deploying distinct models for different clients. We empirically evaluate the performance of PERM, which corroborates the statistical benefits of PERM in comparison to existing methods.

## 2 Personalized Empirical Risk Minimization

In this section, we formally state the problem and introduce PERM as an ideal paradigm for learning from heterogeneous data sources. We assume there are $N$ distributed devices where each holds a distinct data shard $\mathcal{S}_i = \{(\boldsymbol{x}_{i,j}, y_{i,j})\}_{j=1}^{n_i}$ with $n_i$ training samples that are realized by a local source distribution $\mathcal{D}_i$ over instance space $\Xi = \mathcal{X} \times \mathcal{Y}$. The data distributions across the devices are not independently and identically distributed (non-IID or *heterogeneous*), i.e., $\mathcal{D}_1 \neq \mathcal{D}_2 \neq$

$\ldots \neq \mathcal{D}_N$, and each distribution corresponds to a local *generalization error* or *true risk* $\mathcal{L}_i(h) = \mathbb{E}_{(\boldsymbol{x},y)\sim\mathcal{D}_i}[\ell(h(\boldsymbol{x}),y)], i = 1, 2, \ldots, N$ on *unseen* samples for any model $h \in \mathcal{H}$, where $\mathcal{H}$ is the hypothesis set (e.g., a linear model or a deep neural network) and $\ell : \mathcal{Y} \times \mathcal{Y} \to \mathbb{R}^+$ is a given convex or non-convex loss function. We use $\widehat{\mathcal{L}}_i(h) = (1/n_i) \sum_{(\boldsymbol{x},y)\in\mathcal{S}_i} \ell(h(\boldsymbol{x}),y))$ to denote the *local empirical risk* or training loss at $i$th data shard $\mathcal{S}_i$ with $n_i$ samples.

We seek to collaboratively learn a model or personalized models that entail a good generalization on all local distributions, i.e. minimizing true risk $\mathcal{L}_i(\cdot), i = 1, \ldots, N$ for all data sources (all-for-all [33]). A simple non-personalized solution, particularly in FL, aims to minimize a (weighted) *empirical risk* over all data shards in a communication-efficient manner [34]:

$$\arg\min_{h\in\mathcal{H}} \sum_{i=1}^N p(i)\widehat{\mathcal{L}}_i(h) \text{ with } \boldsymbol{p} \in \Delta_N, \tag{WERM}$$

where $\Delta_N = \{\boldsymbol{p} \in \mathbb{R}_+^N \mid \sum_{i=1}^N p(i) = 1\}$ denotes the simplex set.

It has been shown that a *single model* learned by WERM, for example by using fixed mixing weights $p(i) = n_i/n$, where $n$ is total number of training samples, or even agnostic to mixture of distributions [35, 36], while yielding a good performance on the *combined* datasets of all devices, can suffer from a poor generalization error on individual datasets by increasing the diversity among distributions [37, 38, 39, 40]. To overcome this issue, there has been a surge of interest in developing methods that personalize the global model to individual local distributions. These methods can be unified as the following bi-level problem (a similar unification has been made in [12]):

$$\arg\min_{h_1,h_2,\ldots,h_m\in\mathcal{H}} \widehat{\mathcal{F}}_i(h_i \oplus h_*) \quad \text{subject to} \quad h_* = \arg\min_{h\in\mathcal{H}} \sum_{j=1}^N \alpha(j)\widehat{\mathcal{L}}_j(h) \tag{BERM}$$

where $\oplus$ denotes the mixing operation to combine local and global models, and $\widehat{\mathcal{F}}_i$ is a modified local loss which is not necessarily same as local risk $\widehat{\mathcal{L}}_i$. By carefully designing the local loss $\widehat{\mathcal{F}}_i$ and mixing operation $\oplus$, we can develop different penalization schemes for FL including existing methods such as linearly interpolating global and local models [11, 2], multi-task learning [10] and meta-learning [9] as special cases. For example, BERM reduces to *zero-personalization* objective WERM when $h_i \oplus h_* = h_*$, and $\widehat{\mathcal{F}}_i = \widehat{\mathcal{L}}_i$. At the other end of the spectrum lies the *zero-collaboration* where the $i$th client trains its own model without any influence from other clients by setting $h_i \oplus h_* = h_i$, $\widehat{\mathcal{F}}_i = \widehat{\mathcal{L}}_i$. The personalized model with *interpolation* of global and local models can be recovered by setting $h_i \oplus h_* = \alpha h_i + (1-\alpha)h_*$, and $\widehat{\mathcal{F}}_i = \widehat{\mathcal{L}}_i$. While more effective than a single global model learned via WERM, personalization methods suffer from three key issues: i) the global model is still obtained by minimizing the average empirical loss which might limit the statistical benefits of collaboration, ii) overall model complexity increases linearly with number of clients, and iii) a same model space is shared across servers and clients.

To motivate our proposal, let us consider the empirical loss $\sum_{i=1}^N \alpha(i)\widehat{\mathcal{L}}_i(h)$ in WERM (or the inner level objective in BERM) with fixed mixing weights $\boldsymbol{\alpha} \in \Delta_N$, and denote the optimal solution by $\widehat{h}_{\boldsymbol{\alpha}}$. The excess risk of the learned model $\widehat{h}_{\boldsymbol{\alpha}}$ on $i$th local distribution $\mathcal{D}_i$ w.r.t. the optimal local model $h_i^* = \arg\min_{h\in\mathcal{H}} \mathcal{L}_{\mathcal{D}_i}(h)$ (i.e. *all-for-one*) can be bounded by (informal) [31]

$$\mathcal{L}_i\left(\widehat{h}_{\boldsymbol{\alpha}}\right) \le \mathcal{L}_i\left(h_i^*\right) + \sum_{j=1}^N \alpha(j)\mathrm{R}_j(\mathcal{H}) + 2\sum_{j=1}^N \alpha(j)\mathrm{disc}_{\mathcal{H}}\left(\mathcal{D}_j, \mathcal{D}_i\right) + C\sqrt{\sum_{j=1}^N \frac{\alpha(j)^2}{n_j}} \tag{GEN}$$

where $\mathrm{R}_j(\mathcal{H})$ is the empirical Rademacher complexity $\mathcal{H}$ w.r.t. $\mathcal{S}_j$, and $\mathrm{disc}_{\mathcal{H}}(\mathcal{D}_i, \mathcal{D}_j)$ is a pseudo-distance on the set of probability measures on $\Xi$ to assess the discrepancy between the distributions $\mathcal{D}_i$ and $\mathcal{D}_j$ with respect to the hypothesis class $\mathcal{H}$ as defined below [29]:

**Definition 1.** *For a model space $\mathcal{H}$ and $\mathcal{D}, \mathcal{D}'$ two probability distributions on $\Xi = \mathcal{X} \times \mathcal{Y}$,*

$$\mathrm{disc}_{\mathcal{H}}\left(\mathcal{D}, \mathcal{D}'\right) = \sup_{h\in\mathcal{H}} |\mathbb{E}_{\xi\sim\mathcal{D}}(\ell(h,\xi)) - \mathbb{E}_{\xi'\sim\mathcal{D}'}(\ell(h,\xi'))|$$

Intuitively, the discrepancy between the two distributions is large, if there exists a predictor that performs well on one of them and badly on the other. On the other hand, if all functions in the hypothesis class perform similarly on both, then $\mathcal{D}$ and $\mathcal{D}'$ have low discrepancy. The above metric which is a special case of a popular family of distance measures in probability theory and mathematical

statistics known as integral probability metrics (IPMs) [41], can be estimated from finite data by replacing the expected losses with their empirical counterparts (i.e. $\mathcal{L}_i$ with $\widehat{\mathcal{L}}_i$).

From GEN, it can be observed that a mismatch between pairs of distributions limits the benefits of ERM on all distributions. Indeed, the generalization risk w.r.t. $\mathcal{D}_i$ will significantly increase when the distribution divergence terms $\mathrm{disc}_{\mathcal{H}}(\mathcal{D}_j, \mathcal{D}_i)$ are large. It leads to an ideal sample complexity $1/\sqrt{n}$ where $n = n_1 + n_2 + \ldots + n_N$ is the total number of samples, which could have been obtained in the IID setting with $\alpha(j) = 1/N$ when the divergence is small as the pairwise discrepancies disappear. Also, we note that even if the global model achieves a small training error over the union of all data (e.g., over parametrized setting) and can entail a good generalization error with respect to *average distribution*, the divergence term still remains which illustrates the poor performance of the global model on all local distributions $\mathcal{D}_i, i = 1, 2, \ldots, N$. This implies that even personalization of the global model as in BERM can not entail a good generalization on all local distributions as there is no effective transfer of positive knowledge among data sources in the presence of high data heterogeneity among local distributions (similar impossibility results even under seemingly generous assumptions on how distributions relate have been made in multisource domain adaptation as well [42]).

Interestingly the bound suggests that seeking optimal accuracy on *all* local distributions requires choosing a distinct mixing of local losses for each client $i$ that minimizes the right-hand side of GEN. This indicates that in an ideal setting (i.e. *all-for-all*), we can achieve the best accuracy for each local distribution $\mathcal{D}_i$ by *personalizing* the WERM, i.e., (i) first estimating $\boldsymbol{\alpha}_i, i = 1, 2, \ldots, N$ for each client individually, then (ii) solving a variant of WERM for each client with obtained mixing parameters:

$$\arg \min_{h \in \mathcal{H}_i} \sum\nolimits_{j=1}^{N} \alpha_i(j) \widehat{\mathcal{L}}_j(h) \quad \text{for} \quad i = 1, 2, \ldots, N. \tag{PERM}$$

By doing this each device achieves the optimal local generalization error by **learning who to learn with** based on the number of samples at each source and the mismatch between its data distribution with other clients. We also note that compared to WERM and BERM, in PERM since we solve a different aggregated empirical loss for each client, we can pick a different model space/model architecture $\mathcal{H}_i$ for each client to meet its available computational resources.

While this two-stage method is guaranteed to entail optimal test accuracy for all local distributions $\mathcal{D}_i$, however, making it scalable requires overcoming two issues. First, estimating the statistical discrepancies between each pair of data sources (i.e., $\boldsymbol{\alpha}_i, i = 1, \ldots, N$) is a computing burden as it requires solving $O(N^2)$ difference of (non)-convex functions in a distributed manner and requires enough samples form each source to entail good accuracy on estimating pairwise discrepancies [41]. Second, we need to solve $N$ variants of the optimization problem in PERM, possibly each with a different model space, which is infeasible when the number of devices is huge (e.g., cross-device federated learning). In the next section, we propose a simple yet effective idea to overcome these issues in a computationally efficient manner.

## 3 PERM at Scale via Model Shuffling

In this section, we propose a method to efficiently estimate the empirical discrepancies among data sources followed by a model shuffling idea to simultaneously solve $N$ versions of PERM to learn a personalized model for each client. We first start by proposing a two-stage algorithm: estimating mixing parameters followed by model shuffling. Then, we propose a single loop unified algorithm that enjoys the same computation and communication overhead as BERM (twice communication of FedAvg). For ease of exposition, we discuss the proposed algorithms by assuming all the clients share the same model architecture and later on discuss the generalization to heterogeneous model spaces. Specifically, we assume that the model space $\mathcal{H}$ is a parameterized by a convex set $\mathcal{W} \subseteq \mathbb{R}^d$ and use $f_i(\boldsymbol{w}) := \widehat{\mathcal{L}}_i(\boldsymbol{w}) = \sum_{(\boldsymbol{x}, y) \in \mathcal{S}_i} \ell(\boldsymbol{w}; (\boldsymbol{x}, y))$ to denote the empirical loss at $i$th data shard.

### 3.1 Warmup: a two-stage algorithm

We start by proposing a two-stage method for solving $N$ variants of PERM in parallel. In the first stage, we propose an efficient method to learn the mixing parameters for all clients. Then, in stage two, we propose a model shuffling method to solve all personalized empirical losses in parallel.

**Stage 1: Mixing parameters estimation.** In the first stage we aim to efficiently estimate the pairwise discrepancy among local distributions to construct mixing parameters $\boldsymbol{\alpha}_i, i = 1, 2, \ldots N$. From

generalization bound GEN and Definition 1, a direct solution to estimate $\boldsymbol{\alpha}_i$ is to solve the following convex-nonconcave minimax problem for each client:

$$\boldsymbol{\alpha}_i^* = \arg \min_{\boldsymbol{\alpha} \in \Delta_N} \sum_{j=1}^N \alpha(j) \max_{\boldsymbol{w} \in \mathcal{W}} |f_i(\boldsymbol{w}) - f_j(\boldsymbol{w})| + \sum_{j=1}^N \alpha(j)^2 / n_j \tag{1}$$

where we estimate the true risks in pairwise discrepancy terms with their empirical counterparts and drop the complexity term as it becomes identical for all sources by fixing the hypothesis space $\mathcal{H}$ and bounding it with a computable distribution-independent quantity such as VC dimension [43], or it can be controlled by choice of $\mathcal{H}$ or through data-dependent regularization. However, solving the above minimax problem itself is already challenging: the inner maximization loop is a nonconcave (or difference of convex) problem, so most of the existing minimax algorithms will fail on this problem. To our best knowledge, the only provable deterministic algorithm is [44], and it is hard to generalize it to stochastic and distributed fashion. Moreover, since we have $N$ clients, we need to solve $N$ variants of (1), which makes designing a scalable algorithm even harder.

To overcome aforementioned issues, we make two relaxations to estimate the per client mixing parameters. First, we optimize an upper bound of pairwise empirical discrepancies $\sup_{\boldsymbol{w}} |f_i(\boldsymbol{w}) - f_j(\boldsymbol{w})|$ in terms of gradient dissimilarity between local objectives $\|\nabla f_i(\boldsymbol{w}) - \nabla f_j(\boldsymbol{w})\|$ [45], which quantifies how different the local empirical losses are and widely used in analysis of learning from heterogeneous losses as in FL [46]. Second, given that the discrepancy measure based on the supremum could be excessively pessimistic in real-world scenarios, and drawing inspiration from the concept of average drift at the optimal point as a right metric to measure the effect of data heterogeneity in federated learning [47], we propose to measure discrepancy at the optimal solution obtained by solving WERM, i.e., $\boldsymbol{w}^* := \arg \min_{\boldsymbol{w} \in \mathcal{W}} (1/N) \sum_{i=1}^N f_i(\boldsymbol{w})$. By doing this, the problem reduces to a simple minimization for each client, given the optimal global solution. These two relaxations lead to solving the following tractable optimization problem to decide the per-client mixing parameters:

$$\arg \min_{\boldsymbol{\alpha} \in \Delta_N} g_i(\boldsymbol{w}^*, \boldsymbol{\alpha}) := \sum_{j=1}^N \alpha(j) \|\nabla f_i(\boldsymbol{w}^*) - \nabla f_j(\boldsymbol{w}^*)\|^2 + \lambda \sum_{j=1}^N \alpha(j)^2 / n_j \tag{2}$$

where we added a regularization parameter $\lambda$ and used the squared of gradient dissimilarity for computational convenience. Thus, obtaining all $N$ mixing parameters requires solving a single ERM to obtain optimal global solution and $N$ variants of (2). To get the optimal solution in a communication-reduced manner, we adapt the Local SGD algorithm [48] (or FedAvg [14]) and find the optimal solution in intermittent communication setting [49] where the clients work in parallel and are allowed to make $K$ stochastic updates between two communication rounds for $R$ consecutive rounds. The detailed steps are given in Algorithm A1 in Appendix B for completeness. After obtaining the global model $\boldsymbol{w}^R$ we optimize over $\boldsymbol{\alpha}$ in $g_i(\boldsymbol{w}^R, \boldsymbol{\alpha})$ using $T_{\boldsymbol{\alpha}}$ iterations of GD to get $\hat{\boldsymbol{\alpha}}_i$. Actually, we will show that as long as $\boldsymbol{w}^R$ converge to $\boldsymbol{w}^*$, $\hat{\boldsymbol{\alpha}}_i, i = 1, \dots, N$ converges to solution of (2) very fast. Our proof idea is based on the following Lipschitzness observation:

$$\left\|\boldsymbol{\alpha}_{g_i}^*(\boldsymbol{w}^R) - \boldsymbol{\alpha}_{g_i}^*(\boldsymbol{w}^*)\right\|^2 \le 4L^2 \kappa_g^2 \sum_{j=1}^N \left( 2 \left\|\nabla f_i(\boldsymbol{w}^*) - \nabla f_j(\boldsymbol{w}^*)\right\|^2 + 4L^2 \left\|\boldsymbol{w}^R - \boldsymbol{w}^*\right\|^2 \right) \left\|\boldsymbol{w}^R - \boldsymbol{w}^*\right\|^2$$

where $\boldsymbol{\alpha}_{g_i}^*(\boldsymbol{w}) := \arg \min_{\boldsymbol{\alpha} \in \Delta_N} g_i(\boldsymbol{w}, \boldsymbol{\alpha})$ and $\kappa_g := n_{\max}/(2\lambda)$ is the condition number of $g_i(\boldsymbol{w}, \cdot)$ where $n_{\max} = \max_{i \in [N]} n_i$. The Lipschitz constant mainly depends on *gradient dissimilarity at optimum*. As $\boldsymbol{w}^R$ tends to $\boldsymbol{w}^*$, the $\boldsymbol{\alpha}_{g_i}^*(\cdot)$ becomes more Lipschitz continuous, i.e., the coefficient in front of $\left\|\boldsymbol{w}^R - \boldsymbol{w}^*\right\|^2$ getting smaller, thus leading to more accurate mixing parameters.

To establish the convergence, we make the following standard assumptions.

**Assumption 1** (Smoothness and strong convexity)**.** *We assume $f_i(\boldsymbol{x})$'s are L-smooth and $\mu$-strongly convex, i.e.,*

$$\forall \boldsymbol{x}, \boldsymbol{y} : \|\nabla f_i(\boldsymbol{x}) - \nabla f_i(\boldsymbol{y})\| \le L \|\boldsymbol{x} - \boldsymbol{y}\|.$$

$$\forall \boldsymbol{x}, \boldsymbol{y} : f_i(\boldsymbol{y}) \ge f_i(\boldsymbol{x}) + \langle \nabla f_i(\boldsymbol{x}), \boldsymbol{y} - \boldsymbol{x} \rangle + \frac{1}{2}\mu \|\boldsymbol{y} - \boldsymbol{x}\|^2$$

We denote the condition number by $\kappa = L/\mu$.

**Assumption 2** (Bounded variance)**.** *The variance of stochastic gradients computed at each local function is bounded, i.e., $\forall i \in [N], \forall \boldsymbol{w} \in \mathcal{W}, \mathbb{E}[\|\nabla f_i(\boldsymbol{w}; \xi) - \nabla f_i(\boldsymbol{w})\|^2] \le \delta^2$.*

**Assumption 3** (Bounded domain). *The domain $\mathcal{W} \subset \mathbb{R}^d$ is a bounded convex set, with diameter $D$ under $\ell_2$ metric, i.e., $\forall \boldsymbol{w}, \boldsymbol{w}' \in \mathcal{W}, \|\boldsymbol{w} - \boldsymbol{w}'\| \leq D$.*

**Definition 2** (Gradient dissimilarity). *We define the following quantities to measure the gradient dissimilarity among local functions:*

$$\zeta_{i,j}(\boldsymbol{w}) := \|\nabla f_i(\boldsymbol{w}) - \nabla f_j(\boldsymbol{w})\|^2, \quad \bar{\zeta}_i(\boldsymbol{w}) := \frac{1}{N}\sum_{j=1}^{N}\zeta_{i,j}(\boldsymbol{w}),$$

$$\zeta := \sup_{\boldsymbol{w}\in\mathcal{W}}\max_{i\in[N]}\|\nabla f_i(\boldsymbol{w}) - (1/N)\sum_{j=1}^{N}\nabla f_j(\boldsymbol{w})\|^2.$$

The following theorem gives the convergence rate of estimated discrepancies to optimal counterparts.

**Theorem 1.** *Under Assumptions 1-3, if we run Algorithm A1 on $F(\boldsymbol{w}) := \frac{1}{N}\sum_{j=1}^{N} f_j(\boldsymbol{w})$ with $\gamma = \Theta\left(\frac{\log(RK)}{\mu RK}\right)$ for $R$ rounds with synchronization gap $K$, for $\kappa_g = 1/(\lambda n_{\min})$, it holds that*

$$\mathbb{E}\|\boldsymbol{\alpha}_i^R - \boldsymbol{\alpha}_i^*\|^2 \leq \tilde{O}\left(\exp\left(-\frac{T_{\boldsymbol{\alpha}}}{\kappa_g}\right) + \kappa_g^2\bar{\zeta}_i(\boldsymbol{w}^*)L^2\left(\frac{D^2}{RK} + \frac{\kappa\zeta^2}{\mu^2 R^2} + \frac{\delta^2}{\mu^2 NRK}\right)\right) \quad \forall i \in [N].$$

An immediate implication of Theorem 1 is that even we solve (2) at $\boldsymbol{w}^R$, the algorithm will eventually converge to optimal solution of (2) at $\boldsymbol{w}^*$. The core technique in the proof, as we mentioned, is to show that for a parameter within a small region centered at $\boldsymbol{w}^*$, the function $\boldsymbol{\alpha}_{g_i}^*(\boldsymbol{w})$ becomes 'more Lipschitz'. The rigorous characterization of this property is captured by Lemma 3 in appendix.

**Stage 2: Scalable personalized optimization with model shuffling.** After obtaining the per client mixing parameters, in the second stage we aim at solving $N$ different personalized variants of PERM denoted by $\Phi(\hat{\boldsymbol{\alpha}}_1, \boldsymbol{v}), \Phi(\hat{\boldsymbol{\alpha}}_2, \boldsymbol{v}), \ldots \Phi(\hat{\boldsymbol{\alpha}}_N, \boldsymbol{v})$ to learn local models where

$$\min_{\boldsymbol{v}\in\mathcal{W}}\Phi(\hat{\boldsymbol{\alpha}}_i, \boldsymbol{v}) := \frac{1}{N}\sum_{j=1}^{N}\hat{\alpha}_i(j)f_j(\boldsymbol{v}). \tag{3}$$

Here we devise an iterative algorithm based on distributed SGD with periodic averaging (a.k.a. Local SGD [48]) to solve these $N$ optimization problems in parallel with *no extra overhead*. The idea is to replace the model averaging in vanilla distributed (Local) SGD with *model shuffling*. Specifically, as shown in Algorithm 1 the algorithm proceeds for $R$ epochs where each epoch runs for $N$ communication rounds. At the beginning of each epoch $r$ the server generates a random permutation $\sigma_r$ over $N$ clients. At each communication round $j$ within the epoch, the server sends the model of client $i$ to client $i_j = (i + j) \mod N$ in the permutation $\sigma_r$ along with $\alpha_i(i_j)$. After receiving a model from the server, the client updates the received model for $K$ local steps and returns it back to the server. As it can be seen, the updates of each loss $\Phi(\hat{\boldsymbol{\alpha}}_i, \boldsymbol{v}), i = 1, 2, \ldots, N$ during an epoch is equivalent to sequentially processing individual losses in (3) which can be considered as permutation-based SGD but with the different that each component now is updated for $K$ steps. By *interleaving the permutations*, we are able to simultaneously optimize all $N$ objectives. We note that the computation and communication complexity of the proposed algorithm is the same as Local SGD with two differences: the model averaging is replaced with model shuffling, and the algorithms run over a permutation of devices. The convergence rate of Local SGD is well-established in literature [50, 51, 52, 53, 54], but here we establish the convergence of permutation-based variant which is interesting by its own.

**Assumption 4** (Bounded Gradient). *The variance of stochastic gradients computed at each local function is bounded, i.e., $\forall i \in [N], \sup_{\boldsymbol{v}\in\mathcal{W}}\|\nabla f_i(\boldsymbol{v})\| \leq G$.*

We note that the Assumption 4 can be realized since we work with a bounded domain $\mathcal{W}$.

**Theorem 2.** *Let Assumptions 1- 4 hold. Assume $\boldsymbol{\alpha}_i^*$ is the solution of (2). Then if we run Algorithm 1 on the $\hat{\boldsymbol{\alpha}}_i$ obtained from Algorithm A1, then Algorithm 1 with $\eta = \Theta\left(\frac{\log(NKR^3)}{\mu R}\right)$ will output the solution $\hat{\boldsymbol{v}}_i, \forall i \in [N]$, such that with probability at least $1 - p$, the following statement holds:*

$$\mathbb{E}[\Phi(\boldsymbol{\alpha}_i^*, \hat{\boldsymbol{v}}_i) - \Phi(\boldsymbol{\alpha}_i^*, \boldsymbol{v}^*(\boldsymbol{\alpha}_i^*))] \leq \tilde{O}\left(\frac{D^2 L}{NKR^2}\right) + \frac{L\delta^2}{\mu^2 R} + \left(\frac{L^4 + N}{\mu^4 R^2}\right)LG^2 N\log(1/p)$$

$$+ \kappa_{\Phi}^2 L\tilde{O}\left(\exp\left(-\frac{T_{\boldsymbol{\alpha}}}{\kappa_g}\right) + \kappa_g^2\bar{\zeta}_i(\boldsymbol{w}^*)L^2\left(\frac{\kappa\zeta^2}{\mu^2 R^2} + \frac{\delta^2}{\mu^2 NRK}\right)\right),$$

---

**Algorithm 1:** Shuffling Local SGD

---

**Input:** Clients $1, ..., N$, Number of Local Steps $K$, Number of Epoch $R$, mixing parameter $\hat{\boldsymbol{\alpha}}_1, ..., \hat{\boldsymbol{\alpha}}_N$

**Epoch for** $r = 0, ..., R-1$ **do**

    Server generates permutation $\sigma_r : [N] \mapsto [N]$.

    **parallel for** $i = 1, ..., N$ **do**

        Client $i$ sets initial model $\boldsymbol{v}_i^{r,0} = \boldsymbol{v}_i^r$.

        **for** $j = 1, ..., N$ **do**

            Set indices $i_j = \sigma_r((i+j) \mod N)$.

            Server sends $\boldsymbol{v}_i^{r,j}$ to Client $i_j$.

            $\boldsymbol{v}_i^{r,j+1} = \texttt{SGD-Update}(\boldsymbol{v}_i^{r,j}, \eta, i_j, K, \hat{\boldsymbol{\alpha}}_i)$.

        Client $i$ does projection: $\boldsymbol{v}_i^{r+1} = \mathcal{P}_{\mathcal{W}}(\boldsymbol{v}_i^{r,N})$.

**Output:** $\hat{\boldsymbol{v}}_i = \mathcal{P}_{\mathcal{W}}(\boldsymbol{v}_i^R - (1/L)\nabla_{\boldsymbol{v}}\Phi(\hat{\boldsymbol{\alpha}}_i, \boldsymbol{v}_i^R)), \forall i \in [N]$.

$\texttt{SGD-Update}(\boldsymbol{v}, \eta, j, K, \boldsymbol{\alpha})$

    Initialize $\boldsymbol{v}^0 = \boldsymbol{v}$

    **for** $t = 0, ..., K-1$ **do**

        $\boldsymbol{v}^t = \boldsymbol{v}^{t-1} - \eta\alpha(j)N\nabla f_j(\boldsymbol{v}^{t-1}; \xi_j^{t-1})$

    **Output:** $\boldsymbol{v}^K$

---

where $\kappa = \frac{L}{\mu}, \kappa_g = \frac{n_{\max}}{2\lambda}$ and $\kappa_\Phi = \frac{\sqrt{N}G}{\mu}$, and the expectation is taken over randomness of Algorithm A1. That is, to guarantee $\mathbb{E}[\Phi(\boldsymbol{\alpha}_i^*, \hat{\boldsymbol{v}}_i) - \Phi(\boldsymbol{\alpha}_i^*, \boldsymbol{v}_i^*)] \leq \epsilon$, we choose $R = O\left(\max\left\{\frac{L\delta^2}{\mu\epsilon}, \frac{\kappa_\Phi^2\kappa_g^2 L^3\bar{\zeta}_i(\boldsymbol{w}^*)D^2}{\epsilon}\right\}\right)$ and $T_{\boldsymbol{\alpha}} = O\left(\kappa_g \log\left(\frac{L\kappa_\Phi^2}{\epsilon}\right)\right)$.

The above theorem shows that even though we run the optimization on $\Phi(\hat{\boldsymbol{\alpha}}_i, \boldsymbol{v})$, our obtained model $\hat{\boldsymbol{v}}_i$ will still converge to the optimal solution of $\Phi(\boldsymbol{\alpha}_i^*, \boldsymbol{v})$. The convergence rate is contributed from two parts: convergence of $\hat{\boldsymbol{\alpha}}_i$ (Algorithm A1) and convergence of personalized model $\hat{\boldsymbol{v}}_i$ (Algorithm 1). Notice that, for the convergence rate of $\hat{\boldsymbol{v}}_i$, we roughly recover the optimal rate of shuffling SGD [55], which is $O(1/R^2)$. However, we suffer from a $O(\delta^2/R)$ term since each client runs vanilla SGD on their local data (the $\texttt{SGD-Update}$ procedure in Algorithm 1). One medication for this variance term could be deploying variance reduction or shuffling data locally at each client before applying SGD. We notice that there is a recent work [56] also considering the client-level shuffling idea, but our work differs from it in two aspects: 1) they work with local SGD type algorithm and the shuffling idea is employed for model averaging within a subset of clients, while in our algorithm, during each local update period, each client runs shuffling SGD directly on other's model 2) from a theoretical perspective, we are mostly interested in investigating whether the algorithm can converge to the true optimal solution of $\Phi(\boldsymbol{\alpha}_i^*, \boldsymbol{v})$ if we only optimize on a surrogate function $\Phi(\hat{\boldsymbol{\alpha}}_i, \boldsymbol{v})$.

One drawback of Algorithm 1 is that we have to wait for Algorithm A1 to finish and output $\hat{\boldsymbol{\alpha}}_i$, so that we can proceed with Algorithm 1. However, if we are not satisfied with the precision of $\hat{\boldsymbol{\alpha}}_i$, we may not have a chance to go back to refine it. Hence in the next subsection, we propose to interleave Algorithm 1 and Algorithm A1, and introduce a single-loop variant of PERM which will jointly optimize mixture weights and learn personalized models in an interleaving fashion.

### 3.2 A unified single loop algorithm

We now turn to introducing a single-stage algorithm that jointly optimizes $\boldsymbol{\alpha}_i$s and $\boldsymbol{v}_i$s as depicted in Algorithm 2 by intertwining the two stages in Algorithm A1 and Algorithm 1 in a single unified method. The idea is to learn the global model, which is used to estimate mixing parameters, concurrent to personalized models. At each communication round, the clients compute gradients on the global model, on their data, after the server collects these gradients does a step mini-batch SGD update on the global model, and then updates the mixing parameters. Then we proceed to update the personalized models similar to Algorithm 1. We note that, unlike the two-stage method where the mixing parameters are computed at the final global model, here the mixing parameters are updated adaptively based on intermediate global models.

**Theorem 3.** *Let Assumptions 1 to 4 to be satisfied. Assume $\boldsymbol{\alpha}_i^*$ is the solution of (2). Then if we run Algorithm 2 with $\eta = \Theta\left(\frac{\log(NKR^3)}{\mu R}\right)$ and $\gamma = \Theta\left(\frac{\log(NKR^3)}{\mu R}\right)$, it will output the solution $\hat{\boldsymbol{v}}_i$,*

**Algorithm 2:** Single Loop PERM

**Input:** Clients $1, ..., N$, Number of Local Steps $K$, Number of Epoch $R$, Initial mixing
parameter $\boldsymbol{\alpha}_1^0 =, ..., \boldsymbol{\alpha}_N^0 = \bar{\boldsymbol{\alpha}} = [1/N, ..., 1/N]$.

**Epoch for** $r = 0, ..., R - 1$ **do**

    Server generates permutation $\sigma_r : [N] \mapsto [N]$.

    **parallel for** Client $i = 1, ..., N$ **do**

        Client $i$ sets initial model $\boldsymbol{v}_i^{r,0} = \boldsymbol{v}_i^r$.

        **for** $j = 1, ..., N$ **do**

            Set indices $i_j = \sigma_r((i + j) \mod N)$.

            Server sends $\boldsymbol{v}_i^{r,j}$ to client $i_j$ .

            $\boldsymbol{v}_i^{r,j+1} = \texttt{SGD-Update}(\boldsymbol{v}_i^{r,j}, \eta, i_j, K, \boldsymbol{\alpha}_i^r)$.     // Personalized model update

        Client $i$ does projection: $\boldsymbol{v}_i^{r+1} = \mathcal{P}_{\mathcal{W}}(\boldsymbol{v}_i^{r,N})$.

    $\boldsymbol{w}^{r+1} = \mathcal{P}_{\mathcal{W}}(\boldsymbol{w}^r - \gamma \frac{1}{N} \sum_{i=1}^N \frac{1}{M} \sum_{j=1}^M \nabla f_i(\boldsymbol{w}^r, \xi_{i,j}^r))$     // Global model update

    Compute $\boldsymbol{\alpha}_i^{r+1}$ by running $T_{\boldsymbol{\alpha}}$ steps GD on $g_i(\boldsymbol{w}^{r+1}, \boldsymbol{\alpha})$     // $\boldsymbol{\alpha}$ update

**Output:** $\hat{\boldsymbol{v}}_i = \mathcal{P}_{\mathcal{W}}(\boldsymbol{v}_i^R - (1/L)\nabla_{\boldsymbol{v}}\Phi(\boldsymbol{\alpha}_i^R, \boldsymbol{v}_i^R)), \hat{\boldsymbol{\alpha}}_i = \boldsymbol{\alpha}_i^R, \forall i \in [N]$.

$\texttt{SGD-Update}(\boldsymbol{v}, \eta, j, K, \boldsymbol{\alpha})$

    Initialize $\boldsymbol{v}_j^0 = \boldsymbol{v}$

    **for** $t = 0, ..., K - 1$ **do**

        $\boldsymbol{v}^t = \boldsymbol{v}^{t-1} - \eta\alpha(j)N\nabla f_j(\boldsymbol{v}_j^{t-1}; \xi_j^{t-1})$

    **Output:** $\boldsymbol{v}^K$

$\forall i \in [N]$, *such that with probability at least* $1 - p$, *the following statement holds:*

$$\mathbb{E}[\Phi(\boldsymbol{\alpha}_i^*, \hat{\boldsymbol{v}}_i) - \Phi(\boldsymbol{\alpha}_i^*, \boldsymbol{v}_i^*)] \leq O\left(\frac{LD^2}{NKR^3}\right) + \tilde{O}\left(\left(\frac{\kappa^4 L}{R^2} + \frac{NL}{\mu^2 R^2}\right)G^2 N \log(1/p) + \frac{L\delta^2}{\mu R}\right)$$

$$+ \kappa_\Phi^2 L\tilde{O}\left(\frac{\kappa^2 \kappa_g^2 L^2 \bar{\zeta}_i(\boldsymbol{w}^*)DG}{R} + R^2 \exp\left(-\frac{T_{\boldsymbol{\alpha}}}{\kappa_g}\right) + \frac{L\kappa^2 \kappa_g^2 \bar{\zeta}_i(\boldsymbol{w}^*)\delta^2}{\mu^2 M}\right),$$

*where* $\kappa = \frac{L}{\mu}, \kappa_g = \frac{n_{\max}}{2\lambda}, \kappa_\Phi = \frac{\sqrt{N}G}{\mu}$ *and the expectation is taken over the randomness of stochastic samples in Algorithm* 2. *That is, to guarantee* $\mathbb{E}[\Phi(\boldsymbol{\alpha}_i^*, \hat{\boldsymbol{v}}_i) - \Phi(\boldsymbol{\alpha}_i^*, \boldsymbol{v}_i^*)] \leq \epsilon$, *we choose* $M = O\left(\frac{L^2 \kappa^2 \kappa_g^2 \kappa_\Phi^2 \bar{\zeta}_i(\boldsymbol{w}^*)\delta^2}{\mu^2 \epsilon}\right)$, $R = O\left(\max\left\{\frac{L\delta^2}{\mu\epsilon}, \frac{\kappa_\Phi^2 \kappa^2 \kappa_g^2 L^3 \bar{\zeta}_i(\boldsymbol{w}^*)DG}{\epsilon}\right\}\right)$ *and* $T_{\boldsymbol{\alpha}} = O\left(\kappa_g \log\left(\frac{LR^2}{\epsilon}\right)\right)$.

Compared to Theorem 2, we achieve a slightly worse rate, since we need a large mini-batch when we update global model $\boldsymbol{w}$. However, the advantages of the single-loop algorithm are two-fold. First, as we mentioned in the previous subsection, we have the freedom to optimize $\hat{\boldsymbol{\alpha}}_i$ to arbitrary accuracy, while in double loop algorithm (Algorithm A1 + Algorithm 1), once we get $\hat{\boldsymbol{\alpha}}_i$, we do not have the chance to further refine it. Second, in practice, a single-loop algorithm is often easier to implement and can make better use of caches by operating on data sequentially, leading to improved performance, especially on modern processors with complex memory hierarchies.

### 3.3 Extension to heterogeneous model setting

In the homogeneous model setting, we assumed a shared model space $\mathcal{W}$ for clients and the server. However, in real-world FL applications, devices have diverse resources and can only train models that match their capacities. We demonstrate that the PERM paradigm can be extended to support learning in model-heterogeneous settings, where different models with varying capacities are used by the server and clients. Focusing on learning the global optimal model to estimate pairwise statistical discrepancies, we note that by utilizing partial training methods [28], where at each communication round a sub-model with a size proportional to resources of each client is sampled from the server's global model (extracted either random, static, or rolling) and is transmitted to be updated locally. Upon receiving updated sub-models, the server can simply aggregate (average) heterogeneous sub-model updates sent from the clients to update the global model. We can consider the complexity of

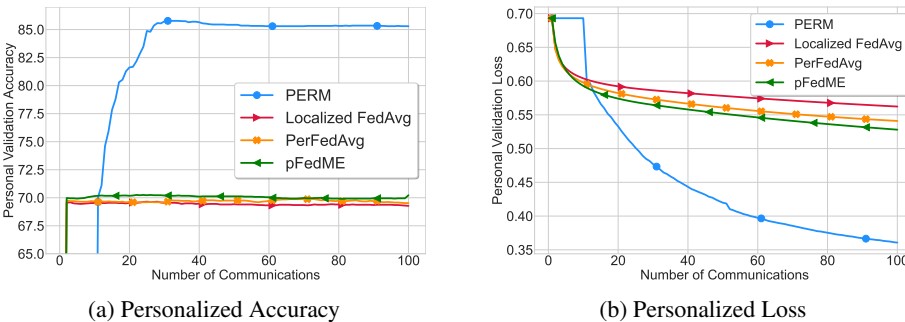

|                          |                          |
| :----------------------: | :----------------------: |
| (a) Personalized Accuracy | (b) Personalized Loss |

Figure 1: Comparative analysis of personalization methods, including our single-loop PERM algorithm, localized FedAvg, perFedAvg, and pFedME, with synthetic data. The disparity in personalized accuracy and loss highlights PERM's capability to leverage relevant client correlations.

models used by clients when estimating mixing parameters by solving a modified version of (2) as:

$$\sum_{j=1}^{N} \alpha(j)\sqrt{\mathsf{VC}(\mathcal{H}_j)/n_j} + \sum_{j=1}^{N} \alpha(j)\|\nabla f_i(\boldsymbol{m}_i \odot \boldsymbol{w}^*) - \nabla f_j(\boldsymbol{m}_j \odot \boldsymbol{w}^*)\|^2 + \lambda\sum_{j=1}^{N} \alpha(j)^2/n_j,$$

where we simply upper bounded the Rademacher complexity w.r.t. each data source in (GEN) with VC dimension [57]. Here $\boldsymbol{m}_i$ is the masking operator to extract a sub-model of the global model to compute local gradients at client $i$ based on its available resources. By doing so, we can adjust mixing parameters based on the complexity of underlying models, as different sub-models of the global model (i.e., $\boldsymbol{m}_i \odot \boldsymbol{w}^*$ versus $\boldsymbol{m}_j \odot \boldsymbol{w}^*$) are used to compute drift between pair of gradients at the optimal solution. With regards to training personalized models with heterogeneous local models, as we solve a distinct aggregated empirical loss for each client by interleaving permutations and shuffling models, we can utilize different model spaces $\mathcal{W}_i, i = 1, \ldots, N$ for different clients that meet their available resources with aforementioned partial training strategies.

## 4 Experimental Results

In this section we benchmark the effectiveness of PERM on synthetic data with 50 clients, where it notably outshone other renowned methods as evident in Figure 1. Our experiments concluded with the CIFAR10 dataset, employing a 2-layer convolutional neural network, where PERM, despite a warm-up phase, demonstrated unmatched convergence performance (Figure 2). Additional experiments are reported in the appendix. Across all datasets, the PERM algorithm consistently showcased its robustness and unmatched efficiency in the realm of personalized federated learning.

**Experiment on synthetic data.** To demonstrate the superior effectiveness of our proposed single-loop PERM algorithm compared to other existing personalization methods, we conducted an experiment using synthetic data generated according to the following specifications. We consider a scenario with a total of $N$ clients, where we draw samples from the distribution $\mathcal{N}(\boldsymbol{\mu}_1, \boldsymbol{\Sigma}_i)$ for half of the clients, denoted by $i \in [1, \frac{N}{2}]$, and from $\mathcal{N}(\boldsymbol{\mu}_2, \boldsymbol{\Sigma}_i)$ for the remaining clients, denoted by $i \in (\frac{N}{2}, N]$. Following the approach outlined in [58], we adopt a uniform variance for all samples, with $\Sigma_{k,k} = k^{-1.2}$. Subsequently, we generate a labeling model using the distribution $\mathcal{N}(\boldsymbol{\mu}_w, \boldsymbol{\Sigma}_w)$.

Given a data sample $\boldsymbol{x} \in \mathbb{R}^d$, the labels are generated as follows: clients $1, ..., \frac{N}{2}$ assign labels based on $y = \text{sign}(\boldsymbol{w}^\top \boldsymbol{x})$, while clients $\frac{N}{2} + 1, ..., N$ assign labels based on $y = \text{sign}(-\boldsymbol{w}^\top \boldsymbol{x})$. For this specific experiment, we set $\mu_1 = 0.2$, $\mu_2 = -0.2$, and $\mu_w = 0.1$. The data dimension is $d = 60$, and there are 2 classes in the output. We have a total of 50 clients, each generating 500 samples following the aforementioned guidelines. We train a logistic regression model on each client's data.

To demonstrate the superiority of our PERM algorithm, we conducted a performance comparison against other prominent personalized approaches, including the fined-tuned model of FedAvg [14] (referred to as localized FedAvg), perFedAg [9], and pFedME [7]. The results in Figure 1 highlight PERM's efficient learning of personalized models for individual clients. In contrast, competing methods relying on globally trained models struggle to match PERM's effectiveness in highly heterogeneous scenarios, as seen in personalized accuracy and loss. This showcases PERM's exceptional ability to leverage relevant client learning.

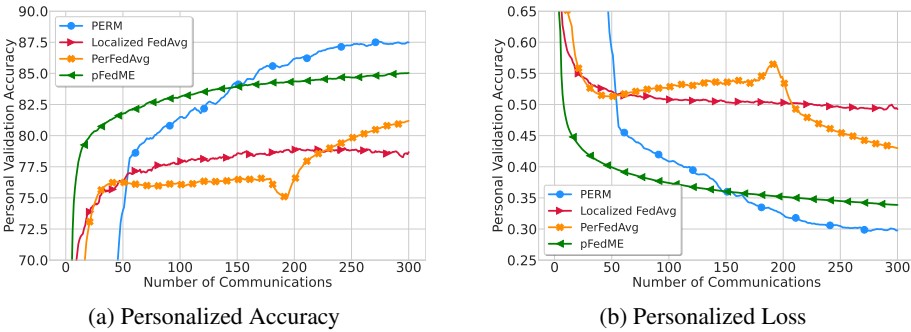

(a) Personalized Accuracy            (b) Personalized Loss

Figure 2: Comparative analysis of our single-loop PERM algorithm, localized FedAvg, PFedMe, and perFedAg, on CIFAR10 dataset and a 2-layer CNN model. Each client has access to only 2 classes of data. PERM rapidly catches up after 10 rounds of warmup without personalization involved.

**Experiment on CIFAR10 dataset.** We extend our experimentation to the CIFAR10 dataset using a 2-layer convolutional neural network. During this test, 50 clients participate, each limited to data from just 2 classes, resulting in a pronounced heterogeneous data distribution. We benchmark our algorithm against PerFedAvg, PFedMe, and the localized FedAvg. As illustrated in Figure 2, PERM demonstrates superior convergence performance compared to other personalized strategies. It's noteworthy that PERM's initial personalized validation is significantly lower than that of approaches like PerFedAvg and PFedMe. This discrepancy stems from our choice to implement 10 communica-

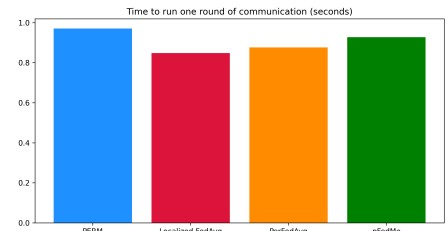

Figure 3: Runtime of different algorithms in a limited environment. We compare PERM (single loop), PerFedAvg, FedAvg, and pFedMe. PERM has a minimal overhead over FedAvg and is comparable to other personalization methods.

tion rounds as a warm-up phase before initiating personalization, whereas other models embark on personalization right from the outset.

**Computational overhead.** In demonstrating the computational efficiency of the proposed PERM algorithm, we present a comparison of wall-clock time of completing one round of communication of PERM and other methods. Each method undertakes 20 local steps along with their distinct computations for personalization. As depicted in Figure 3, the PERM (single loop) algorithm's runtime is compared against personalization methods such as PerFedAvg, FedAvg, and pFedMe. Remarkably, PERM maintains a notably minimal computational overhead. The run-time is slightly worse due to overhead of estimating mixing parameters.

## 5 Discussion & Conclusion

This paper introduces a new *data&system-aware* paradigm for learning from multiple heterogeneous data sources to achieve optimal statistical accuracy across all data distributions without imposing stringent constraints on computational resources shared by participating devices. The proposed PERM schema, though simple, provides an efficient solution to enable each client to learn a personalized model by *learning who to learn with* via personalizing the aggregation of data sources through an efficient empirical statistical discrepancy estimation module. To efficiently solve all aggregated personalized losses, we propose a model shuffling idea to optimize all losses in parallel. PERM can also be employed in other learning settings with multiple sources of data such as domain adaptation and multi-task learning to entail optimal statistical accuracy.

We would like to embark on the scalability of PERM. The compute burden on clients and servers is roughly the same as existing methods thanks to shuffling (except for extra overhead due to estimating mixing parameters which is the same as running FedAvg in a two-stage approach and an extra communication in an interleaved approach). The only hurdle would be the required *memory at server* to maintain mixing parameters, which scales proportionally to the square of the number of clients, which can be alleviated by clustering devices which we leave as a future work.

## Acknowledgement

This work was partially supported by NSF CAREER Award #2239374 NSF CNS Award #1956276.

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

# A  Additional Experiments

In addition to experiments on synthetic and CIFAR10 datasets reported before, we have also conducted experiments on the EMNIST dataset, highlighting PERM's capability to derive superior personalized models by tapping into inter-client data similarities. Additionally, further insights emerged from our tests on the MNIST dataset, revealing how PERM's learned mixture weights adeptly respond to both homogeneous and highly heterogeneous data scenarios.

**Experiment on EMNIST dataset**   In addition to the synthetic and CIFAR10 datasets discussed in the main body, we run experiments on the EMNIST dataset [59], which is naturally distributed in a federated setting. In this case, we chose 50 clients and use a 2-layer MLP model, each with 200 neurons. We compare the PERM algorithm with the localized model in FedAvg and perFedAvg [9]. As it can be seen in Figure 4, PERM can learn a better personalized model by attending to each client's data according to the similarity of the data distribution between clients. The learned values of $\alpha$, in Figure 5, show that the clients are learning from each others' data, and not focused on their own data only. This signifies that the distribution of data among clients in this dataset is not highly heterogeneous. Note that, since we are using a subset of clients in the EMNIST dataset for the training (only 50 clients for 100 rounds of communication), the results would be sub-optimal. Nonetheless, the experiments are designed to show the effectiveness of different algorithms. As it can be concluded, in terms of performance, PERM consistently excels beyond its peers, demonstrating exemplary results on various benchmark datasets.

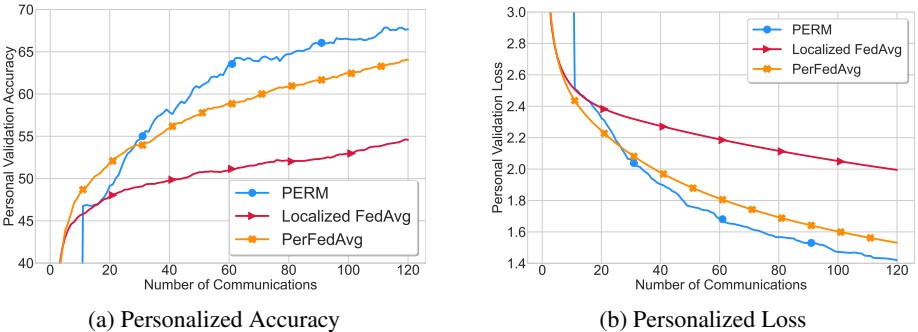

(a) Personalized Accuracy                    (b) Personalized Loss

Figure 4: Comparative Analysis of Personalization methods, including our single-loop PERM algorithm, localized FedAvg, and perFedAg, with EMNIST dataset. The disparity in personalized accuracy and loss highlights PERM's capability in leveraging relevant client correlations.

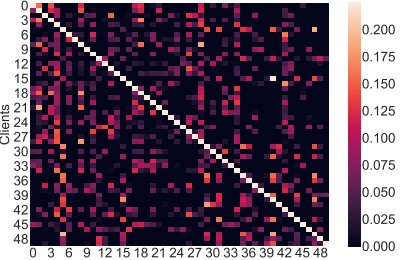

Figure 5: The heat map of the learned $\alpha$ values for the PERM algorithms on the EMNIST dataset with a 2-layer MLP model. The weights signify that clients mutually benefiting from one another's data, which also highlight that the distribution of data is not significantly heterogeneous in this dataset.

**The effectiveness of learned mixture weights**   To show the effectiveness of the two-stage PERM algorithm, as well as the effects of heterogeneity on the distribution of data among clients on the learned weights $\alpha$ in the algorithm, we run this algorithm on MNIST dataset. We use 50 clients, and the model is an MLP, similar to the EMNIST experiment. In this case, we consider two cases: distributing the data randomly across clients (homogeneous) and only allocating 1 class per client (highly heterogeneous). As it can be seen from Figure 6, when the data distribution is homogeneous the learned values of $\alpha$ as diffused across clients. However, when the data is highly heterogeneous, the learned $\alpha$ values will be highly sparse, indicating that each client is mostly learning from its own data and some other clients with partial distribution similarity. Notably, the matrix predominantly exhibits sparsity, indicating that each client selectively leverages information solely from a subset of other clients. This discernible pattern reinforces the inherent confidence that each client is effectively learning from a limited but strategically chosen group of clients.

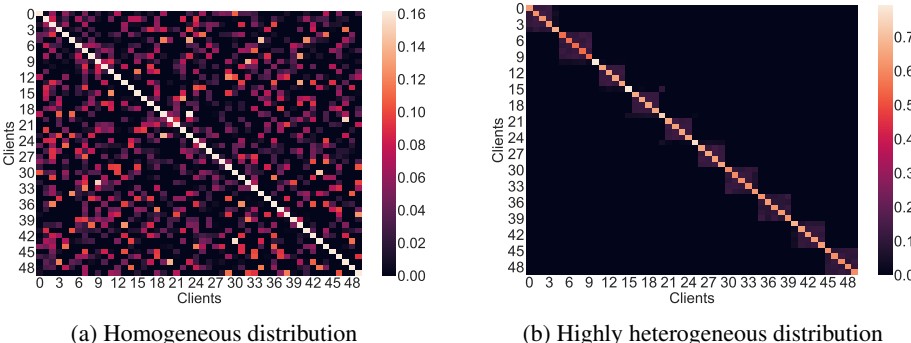

(a) Homogeneous distribution       (b) Highly heterogeneous distribution

Figure 6: Comparing the performance of two-stage PERM algorithm in learning $\alpha$ values on heterogeneous and homogeneous data distributions. We use MNIST dataset across 50 clients with homogeneous and heterogeneous distributions.

# B Proof of Two Stages Algorithm

In this section we provide the proof of convergence of two-stage implementation of PERM (computing mixing parameters followed by learning personalized models via model shuffling using permutation-based variant of distributed SGD with periodic communication).

## B.1 Technical Lemmas

**Lemma 1.** *Define $v^*(\alpha) := \arg\min_{v \in \mathcal{W}} \Phi(\alpha, v)$, and assume $\Phi(\alpha, \cdot)$ is $\mu$-strongly convex and $\nabla_v \Phi(\alpha, v)$ is $L$ Lipschitz in $\alpha$. Then, $v^*(\cdot)$ is $\kappa$-Lipschitz where $\kappa = L/\mu$.*

*Proof.* The proof is similar to Lin et al's result on minimax objective [60]. First, according to optimality conditions we have:

$$\langle v - v^*(\alpha), \nabla_2 \Phi(\alpha, v^*(\alpha)) \rangle \geq 0,$$
$$\langle v - v^*(\alpha'), \nabla_2 \Phi(\alpha', v^*(\alpha')) \rangle \geq 0$$

Substituting $v$ with $v^*(\alpha')$ and $v^*(\alpha)$ in the above first and second inequalities respectively yields:

$$\langle v^*(\alpha') - v^*(\alpha), \nabla_2 \Phi(\alpha, v^*(\alpha)) \rangle \geq 0,$$
$$\langle v^*(\alpha) - v^*(\alpha'), \nabla_2 \Phi(\alpha', v^*(\alpha')) \rangle \geq 0$$

Adding up the above two inequalities yields:

$$\langle v^*(\alpha') - v^*(\alpha), \nabla_2 \Phi(\alpha, v^*(\alpha)) - \nabla_2 \Phi(\alpha', v^*(\alpha')) \rangle \geq 0, \tag{4}$$

Since $\Phi(\alpha, \cdot)$ is $\mu$ strongly convex, we have:

$$\langle v^*(\alpha') - v^*(\alpha), \nabla_2 \Phi(\alpha, v^*(\alpha')) - \nabla_2 \Phi(\alpha, v^*(\alpha)) \geq \mu \| v^*(\alpha') - v^*(\alpha) \|^2. \tag{5}$$

Adding up (4) and (5) yields:

$$\langle v^*(\alpha') - v^*(\alpha), \nabla_2 \Phi(\alpha, v^*(\alpha')) - \nabla_2 \Phi(\alpha', v^*(\alpha')) \geq \mu \| v^*(\alpha') - v^*(\alpha) \|^2$$

Finally, using $L$ smoothness of $\Phi$ will conclude the proof:

$$L \| v^*(\alpha') - v^*(\alpha) \| \| \alpha - \alpha' \| \geq \mu \| v^*(\alpha') - v^*(\alpha) \|^2$$
$$\iff \kappa \| \alpha - \alpha' \| \geq \| v^*(\alpha') - v^*(\alpha) \|$$

$\square$

**Lemma 2** (Optimality Gap)**.** *Let $\Phi(\alpha, v)$ be defined in (3). Let $\hat{v} = \mathcal{P}_{\mathcal{W}}(\tilde{v} - \frac{1}{L} \nabla_v \Phi(\hat{\alpha}, \tilde{v}))$. If we assume each $f_i$ is $L$-smooth, $\mu$-strongly convex and with gradient bounded by $G$, then the following statement holds true:*

$$\Phi(\alpha^*, \hat{v}) - \Phi(\alpha^*, v^*) \leq 2L \| \tilde{v} - v^*(\hat{\alpha}) \|^2 + \left( 2\kappa_\Phi^2 L + \frac{4NG^2}{L} \right) \| \hat{\alpha} - \alpha^* \|^2.$$

*where $\kappa_\Phi = \frac{\sqrt{N}G}{\mu}$, $v^* = \arg\min_{v \in \mathcal{W}} \Phi(\alpha^*, v)$.*

*Proof.* First we show that $\nabla_{\boldsymbol{v}} \Phi(\boldsymbol{\alpha}, \boldsymbol{v})$ is $\sqrt{N}G$ Lipschitz in $\boldsymbol{\alpha}$. To see this:

$$\|\nabla_{\boldsymbol{v}} \Phi(\boldsymbol{\alpha}, \boldsymbol{v}) - \nabla_{\boldsymbol{v}} \Phi(\boldsymbol{\alpha}', \boldsymbol{v})\| = \left\| \sum_{j=1}^{N} \alpha_i(j) \nabla f_j(\boldsymbol{v}) - \sum_{j=1}^{N} \alpha_i'(j) \nabla f_j(\boldsymbol{v}) \right\|$$

$$\leq \sqrt{N}G \|\boldsymbol{\alpha}_i - \boldsymbol{\alpha}_i'\|.$$

Hence due to Lemma 1, we know $\boldsymbol{v}^*(\boldsymbol{\alpha})$ is $\kappa_\Phi := \frac{\sqrt{N}G}{\mu}$ Lipschitz. According to property of projection, we have:

$$0 \leq \langle \boldsymbol{v} - \hat{\boldsymbol{v}}, \, L(\hat{\boldsymbol{v}} - \tilde{\boldsymbol{v}}) + \nabla_{\boldsymbol{v}} \Phi(\hat{\boldsymbol{\alpha}}, \tilde{\boldsymbol{v}}) \rangle$$
$$= \underbrace{\langle \boldsymbol{v} - \hat{\boldsymbol{v}}, \, L(\hat{\boldsymbol{v}} - \tilde{\boldsymbol{v}}) + \nabla_{\boldsymbol{v}} \Phi(\boldsymbol{\alpha}^*, \tilde{\boldsymbol{v}}) \rangle}_{T_1} + \underbrace{\langle \boldsymbol{v} - \hat{\boldsymbol{v}}, \, \nabla_{\boldsymbol{v}} \Phi(\hat{\boldsymbol{\alpha}}, \tilde{\boldsymbol{v}}) - \nabla_{\boldsymbol{v}} \Phi(\boldsymbol{\alpha}^*, \tilde{\boldsymbol{v}}) \rangle}_{T_2}.$$

For $T_1$, we notice:

$$\langle \boldsymbol{v} - \hat{\boldsymbol{v}}, \, L(\hat{\boldsymbol{v}} - \tilde{\boldsymbol{v}}) + \nabla_{\boldsymbol{v}} \Phi(\boldsymbol{\alpha}^*, \tilde{\boldsymbol{v}}) \rangle = L \langle \boldsymbol{v} - \tilde{\boldsymbol{v}}, \, \hat{\boldsymbol{v}} - \tilde{\boldsymbol{v}} \rangle + \frac{1}{\eta} \langle \tilde{\boldsymbol{v}} - \hat{\boldsymbol{v}}, \, \hat{\boldsymbol{v}} - \tilde{\boldsymbol{v}} \rangle + \langle \boldsymbol{v} - \hat{\boldsymbol{v}}, \, \nabla_{\boldsymbol{v}} \Phi(\boldsymbol{\alpha}^*, \tilde{\boldsymbol{v}}) \rangle$$

$$= L \langle \boldsymbol{v} - \tilde{\boldsymbol{v}}, \, \hat{\boldsymbol{v}} - \tilde{\boldsymbol{v}} \rangle - L \|\tilde{\boldsymbol{v}} - \hat{\boldsymbol{v}}\|^2 + \langle \boldsymbol{v} - \hat{\boldsymbol{v}}_i, \, \nabla_{\boldsymbol{v}} \Phi(\boldsymbol{\alpha}^*, \tilde{\boldsymbol{v}}) \rangle$$

$$\leq L(\|\boldsymbol{v} - \tilde{\boldsymbol{v}}\|^2 + \frac{1}{4} \|\hat{\boldsymbol{v}} - \tilde{\boldsymbol{v}}\|^2) - L \|\tilde{\boldsymbol{v}} - \hat{\boldsymbol{v}}\|^2 + \underbrace{\langle \boldsymbol{v} - \hat{\boldsymbol{v}}, \, \nabla_{\boldsymbol{v}} \Phi(\boldsymbol{\alpha}^*, \tilde{\boldsymbol{v}}) \rangle}_{\spadesuit}$$

where at last step we used Young's inequality. To bound $\spadesuit$, we apply the $L$ smoothness and $\mu$ strongly convexity of $\Phi(\boldsymbol{\alpha}, \cdot)$:

$$\langle \boldsymbol{v} - \hat{\boldsymbol{v}}_i, \, \nabla_{\boldsymbol{v}} \Phi(\boldsymbol{\alpha}_i^*, \tilde{\boldsymbol{v}}_i) \rangle = \langle \boldsymbol{v} - \tilde{\boldsymbol{v}}_i, \, \nabla_{\boldsymbol{v}} \Phi(\boldsymbol{\alpha}^*, \tilde{\boldsymbol{v}}) \rangle + \langle \tilde{\boldsymbol{v}} - \hat{\boldsymbol{v}}, \, \nabla_{\boldsymbol{v}} \Phi(\boldsymbol{\alpha}^*, \tilde{\boldsymbol{v}}) \rangle$$

$$\leq \Phi(\boldsymbol{\alpha}^*, \boldsymbol{v}) - \Phi(\boldsymbol{\alpha}^*, \tilde{\boldsymbol{v}}) - \frac{\mu}{2} \|\tilde{\boldsymbol{v}} - \boldsymbol{v}\|^2 + \Phi(\boldsymbol{\alpha}^*, \tilde{\boldsymbol{v}}) - \Phi(\boldsymbol{\alpha}^*, \hat{\boldsymbol{v}}) + \frac{L}{2} \|\tilde{\boldsymbol{v}} - \hat{\boldsymbol{v}}\|^2$$

$$\leq \Phi(\boldsymbol{\alpha}^*, \boldsymbol{v}) - \Phi(\boldsymbol{\alpha}^*, \hat{\boldsymbol{v}}) - \frac{\mu}{2} \|\tilde{\boldsymbol{v}} - \boldsymbol{v}\|^2 + \frac{L}{2} \|\tilde{\boldsymbol{v}} - \hat{\boldsymbol{v}}\|^2$$

Putting above bound back yields:

$$\left\langle \boldsymbol{v} - \hat{\boldsymbol{v}}, \, \frac{1}{\eta}(\hat{\boldsymbol{v}} - \tilde{\boldsymbol{v}}) + \nabla_{\boldsymbol{v}} \Phi(\boldsymbol{\alpha}^*, \tilde{\boldsymbol{v}}) \right\rangle \leq \Phi(\boldsymbol{\alpha}^*, \boldsymbol{v}) - \Phi(\boldsymbol{\alpha}^*, \hat{\boldsymbol{v}}) + \frac{1}{2\eta} \|\tilde{\boldsymbol{v}} - \boldsymbol{v}\|^2 - \left( \frac{3L}{4} - \frac{L}{2} \right) \|\tilde{\boldsymbol{v}} - \hat{\boldsymbol{v}}\|^2$$

Now we switch to bounding $T_2$. Applying Cauchy-Schwartz yields:

$$\langle \boldsymbol{v} - \hat{\boldsymbol{v}}_i, \, \nabla_{\boldsymbol{v}} \Phi(\hat{\boldsymbol{\alpha}}_i, \tilde{\boldsymbol{v}}_i) - \nabla_{\boldsymbol{v}} \Phi(\boldsymbol{\alpha}_i^*, \tilde{\boldsymbol{v}}_i) \rangle \leq \frac{L}{4} \|\boldsymbol{v} - \tilde{\boldsymbol{v}}_i\|^2 + \frac{L}{4} \|\tilde{\boldsymbol{v}} - \hat{\boldsymbol{v}}_i\|^2 + \frac{4}{L} \|\nabla_{\boldsymbol{v}} \Phi(\hat{\boldsymbol{\alpha}}_i, \tilde{\boldsymbol{v}}_i) - \nabla_{\boldsymbol{v}} \Phi(\boldsymbol{\alpha}_i^*, \tilde{\boldsymbol{v}}_i)\|^2$$

$$\leq \frac{L}{4} \|\boldsymbol{v} - \tilde{\boldsymbol{v}}_i\|^2 + \frac{L}{4} \|\tilde{\boldsymbol{v}} - \hat{\boldsymbol{v}}_i\|^2 + \frac{4NG^2}{L} \|\hat{\boldsymbol{\alpha}}_i - \boldsymbol{\alpha}_i^*\|^2$$

where at last step we apply $\sqrt{N}G$ smoothness of $\Phi(\cdot, \boldsymbol{v})$. Putting pieces together yields:

$$0 \leq \Phi(\boldsymbol{\alpha}_i^*, \boldsymbol{v}) - \Phi(\boldsymbol{\alpha}_i^*, \hat{\boldsymbol{v}}_i) + \frac{L}{2} \|\tilde{\boldsymbol{v}}_i - \boldsymbol{v}\|^2 + \frac{L}{2} \|\boldsymbol{v} - \tilde{\boldsymbol{v}}_i\|^2 + \frac{4NG^2}{L} \|\hat{\boldsymbol{\alpha}}_i - \boldsymbol{\alpha}_i^*\|^2$$

Re-arranging terms and setting $\boldsymbol{v} = \boldsymbol{v}^*(\boldsymbol{\alpha}^*) = \arg\min_{\boldsymbol{v} \in \mathcal{W}} \Phi(\boldsymbol{\alpha}^*, \boldsymbol{v})$ yields:

$$\Phi(\boldsymbol{\alpha}^*, \hat{\boldsymbol{v}}) - \Phi(\boldsymbol{\alpha}^*, \boldsymbol{v}^*) \leq L \|\tilde{\boldsymbol{v}} - \boldsymbol{v}^*\|^2 + \frac{4NG^2}{L} \|\hat{\boldsymbol{\alpha}} - \boldsymbol{\alpha}^*\|^2.$$

At last, due to the $\kappa_\Phi$-Lipschitzness property of of $\boldsymbol{v}^*(\cdot)$ as shown in Lemma 1, it follows that:

$$L\|\tilde{\boldsymbol{v}} - \boldsymbol{v}^*(\boldsymbol{\alpha}^*)\|^2 \leq L\|\tilde{\boldsymbol{v}} - \boldsymbol{v}^*(\hat{\boldsymbol{\alpha}})\|^2 + L\|\boldsymbol{v}^*(\hat{\boldsymbol{\alpha}}) - \boldsymbol{v}^*(\boldsymbol{\alpha}^*)\|^2$$
$$\leq 2L\|\tilde{\boldsymbol{v}} - \boldsymbol{v}^*(\hat{\boldsymbol{\alpha}})\|^2 + 2\kappa_\Phi^2 L\|\hat{\boldsymbol{\alpha}} - \boldsymbol{\alpha}^*\|^2,$$

as desired.

$\square$

**Algorithm A1:** Discrepancy Estimation at Optimum

**Input:** Number of clients $N$, number of local steps $K$, number of communications rounds $R$

**for** $r = 0, \ldots, R - 1$ **do**

 **parallel for** client $i = 0, ..., N - 1$ **do**

  Client $i$ initializes model $\boldsymbol{w}_i^{r,0} = \boldsymbol{w}_i^r$.

  **for** $t = 0, ..., K - 1$ **do**

   $\boldsymbol{w}_i^{r,t+1} = \boldsymbol{w}_i^{r,t} - \gamma \nabla f_i(\boldsymbol{w}_i^{r,t}; \xi_i^{r,t})$ where $\xi_i^{r,t}$ is a mini-batch sampled from $\mathcal{S}_i$.

  Client $i$ sends $\boldsymbol{w}_i^{r,K}$ to Server.

 Server computes $\boldsymbol{w}^{r+1} = \mathcal{P}_{\mathcal{W}} \left( \frac{1}{N} \sum_{i=1}^{N} \boldsymbol{w}_i^{r,K} \right)$

 Server broadcasts $\boldsymbol{w}^{r+1}$ to all clients.

Server computes $\hat{\boldsymbol{\alpha}}_i, i = 1, 2, \ldots, N$ by running $T_{\boldsymbol{\alpha}}$ steps of GD on $g_i(\boldsymbol{w}^R, \boldsymbol{\alpha})$.

**Output:** $\hat{\boldsymbol{\alpha}}_1, \ldots, \hat{\boldsymbol{\alpha}}_N$.

## B.2   Proof of Convergence of Theorem 1

In this section we are going to prove the result in Theorem 1. To this end, we need to show that mixing parameters we compute by first learning the global model and then solving the optimization problem in objective (2) (as depicted in Algorithm A1) converges to optimal values. Notice that in Algorithm A1 we do not solve $g_i(\boldsymbol{w}^*, \boldsymbol{\alpha})$ directly, but optimize $g_i(\boldsymbol{w}^R, \boldsymbol{\alpha})$ on $\boldsymbol{\alpha}$ for $T_{\boldsymbol{\alpha}}$ iterations of GD. Hence, firstly we need to show that optimizing the surrogate function will also guarantee the convergence of output of algorithm $\widehat{\boldsymbol{\alpha}}$ to $\boldsymbol{\alpha}^*$ by deriving a property of the objective in (2). Formally the property is captured by the following lemma.

**Lemma 3.** *Let $g(\boldsymbol{w}, \boldsymbol{\alpha}) := \sum_{j=1}^{N} \alpha_j \|\nabla f_i(\boldsymbol{w}) - \nabla f_j(\boldsymbol{w})\|^2 + \lambda \sum_{j=1}^{N} \alpha_j^2/n_j$ and $\boldsymbol{\alpha}_g^*(\boldsymbol{w}) = \arg\min_{\boldsymbol{\alpha} \in \Delta_N} g(\boldsymbol{w}, \boldsymbol{\alpha})$. Let $\boldsymbol{w}^R$ be the output of Algorithm A1. Then the following statement holds:*

$$\left\| \boldsymbol{\alpha}_g^*(\boldsymbol{w}^R) - \boldsymbol{\alpha}_g^*(\boldsymbol{w}^*) \right\| \leq \kappa_g^2 \sum_{j=1}^{N} \left( 2 \|\nabla f_i(\boldsymbol{w}^*) - \nabla f_j(\boldsymbol{w}^*)\|^2 + 4L^2 \|\boldsymbol{w}^R - \boldsymbol{w}^*\|^2 \right) 4L \|\boldsymbol{w}^R - \boldsymbol{w}^*\|^2$$

*where $\kappa_g := \frac{n_{\max}}{2\lambda}$.*

*Proof.* Define function

$$W(\boldsymbol{z}, \boldsymbol{\alpha}) = \sum_{j=1}^{N} \alpha_j z_j + \lambda \sum_{j=1}^{N} \alpha_j^2/n_j \tag{6}$$

Apparently, $W(\boldsymbol{z}, \boldsymbol{\alpha})$ is linear in $\boldsymbol{z}$ and $2\frac{\lambda}{n_{\max}}$ strongly convex in $\boldsymbol{z}$. Next we show that $\nabla_{\boldsymbol{\alpha}} W(\boldsymbol{z}, \boldsymbol{\alpha})$ is Lipschitz in $\boldsymbol{w}$. To see this,

$$\|\nabla_{\boldsymbol{\alpha}} W(\boldsymbol{z}, \boldsymbol{\alpha}) - \nabla_{\boldsymbol{\alpha}} W(\boldsymbol{z}', \boldsymbol{\alpha})\| = \|[z_1, ..., z_N] - [z_1', ..., z_N']\|$$
$$\leq \|\boldsymbol{z} - \boldsymbol{z}'\|.$$

Then, according to Proposition 1, $\boldsymbol{\alpha}_W^*(\boldsymbol{z}) := \arg\min_{\boldsymbol{\alpha} \in \Delta_N} W(\boldsymbol{z}, \boldsymbol{\alpha})$ is $\kappa_g$ lipschitz in $\boldsymbol{z}$ where $\kappa_g = \frac{n_{\max}}{2\lambda}$, i.e., $\|\boldsymbol{\alpha}_W^*(\boldsymbol{z}) - \boldsymbol{\alpha}_W^*(\boldsymbol{z}')\| \leq \kappa_g \|\boldsymbol{z} - \boldsymbol{z}'\|$. Now, let us consider the objective (2):

$$g(\boldsymbol{w}, \boldsymbol{\alpha}) := \sum_{j=1}^{N} \alpha_j \|\nabla f_i(\boldsymbol{w}) - \nabla f_j(\boldsymbol{w})\|^2 + \lambda \sum_{j=1}^{N} \alpha_j^2/n_j$$

We define $\boldsymbol{\alpha}_g^*(\boldsymbol{w}) = \arg\min_{\boldsymbol{\alpha} \in \Delta_N} g(\boldsymbol{w}, \boldsymbol{\alpha})$.

We set

$$\boldsymbol{z}^R = \left[ \|\nabla f_i(\boldsymbol{w}^R) - \nabla f_1(\boldsymbol{w}^R)\|^2, ..., \|\nabla f_i(\boldsymbol{w}^R) - \nabla f_N(\boldsymbol{w}^R)\|^2 \right],$$
$$\boldsymbol{z}^* = \left[ \|\nabla f_i(\boldsymbol{w}^*) - \nabla f_1(\boldsymbol{w}^*)\|^2, ..., \|\nabla f_i(\boldsymbol{w}^*) - \nabla f_N(\boldsymbol{w}^*)\|^2 \right].$$

Then we know that

$$\left\|\boldsymbol{\alpha}_g^*(\boldsymbol{w}^R) - \boldsymbol{\alpha}_g^*(\boldsymbol{w}^*)\right\|^2 = \left\|\boldsymbol{\alpha}_W^*(\boldsymbol{z}^R) - \boldsymbol{\alpha}_W^*(\boldsymbol{z}^*)\right\|^2 \leq \kappa_g^2 \left\|\boldsymbol{z}^R - \boldsymbol{z}^*\right\|^2 \tag{7}$$

$$\leq \kappa_g^2 \sum_{j=1}^{N} \left| \left\|\nabla f_i(\boldsymbol{w}^R) - \nabla f_j(\boldsymbol{w}^R)\right\|^2 - \left\|\nabla f_i(\boldsymbol{w}^*) - \nabla f_j(\boldsymbol{w}^*)\right\|^2 \right|^2 \tag{8}$$

$$\leq \kappa_g^2 \sum_{j=1}^{N} \left| \left(\nabla f_i(\boldsymbol{w}^R) - \nabla f_j(\boldsymbol{w}^R) + \nabla f_i(\boldsymbol{w}^*) - \nabla f_j(\boldsymbol{w}^*)\right) \right.$$

$$\left. \times \left(\nabla f_i(\boldsymbol{w}^R) - \nabla f_j(\boldsymbol{w}^R) - \nabla f_i(\boldsymbol{w}^*) + \nabla f_j(\boldsymbol{w}^*)\right) \right|^2$$

$$\leq \kappa_g^2 \sum_{j=1}^{N} \left\|\nabla f_i(\boldsymbol{w}^R) - \nabla f_j(\boldsymbol{w}^R) + \nabla f_i(\boldsymbol{w}^*) - \nabla f_j(\boldsymbol{w}^*)\right\|^2 4L^2 \left\|\boldsymbol{w}^R - \boldsymbol{w}^*\right\|^2$$

Since $\left\|\nabla f_i(\boldsymbol{w}^R) - \nabla f_j(\boldsymbol{w}^R)\right\| \leq \left\|\nabla f_i(\boldsymbol{w}^*) - \nabla f_j(\boldsymbol{w}^*)\right\| + 2L \left\|\boldsymbol{w}^R - \boldsymbol{w}^*\right\|$, we can conclude that

$$\left\|\boldsymbol{\alpha}_g^*(\boldsymbol{w}^R) - \boldsymbol{\alpha}_g^*(\boldsymbol{w}^*)\right\| \leq \kappa_g^2 \sum_{j=1}^{N} \left(2 \left\|\nabla f_i(\boldsymbol{w}^*) - \nabla f_j(\boldsymbol{w}^*)\right\|^2 + 4L^2 \left\|\boldsymbol{w}^R - \boldsymbol{w}^*\right\|^2\right) 4L \left\|\boldsymbol{w}^R - \boldsymbol{w}^*\right\|^2.$$

$\square$

With above lemma, to show the convergence of $\hat{\boldsymbol{\alpha}}$ to $\boldsymbol{\alpha}^*$, we do the following decomposition

$$\left\|\hat{\boldsymbol{\alpha}} - \boldsymbol{\alpha}^*\right\|^2 \leq 2 \left\|\hat{\boldsymbol{\alpha}} - \boldsymbol{\alpha}_g^*(\boldsymbol{w}^R)\right\|^2 + 2 \left\|\boldsymbol{\alpha}_g^*(\boldsymbol{w}^R) - \boldsymbol{\alpha}_g^*(\boldsymbol{w}^*)\right\|^2$$

$$\leq 2(1 - \mu\eta_\alpha)^K + 2\kappa_g^2 \sum_{j=1}^{N} \left(2 \left\|\nabla f_i(\boldsymbol{w}^*) - \nabla f_j(\boldsymbol{w}^*)\right\|^2 + 4L^2 \left\|\boldsymbol{w}^R - \boldsymbol{w}^*\right\|^2\right) 4L \left\|\boldsymbol{w}^R - \boldsymbol{w}^*\right\|^2.$$

Now it remains to show the convergence of Local SGD *last iterate* $\boldsymbol{w}^R$ to optimal solution $\boldsymbol{w}^*$. By convention, we use $\boldsymbol{w}^t = \frac{1}{N} \sum_{i=1}^{N} \boldsymbol{w}_i^t$ to denote the virtual average iterates.

**Lemma 4** (One iteration analysis of Local SGD). *Under the condition of Theorem 1, the following statement holds true for any $t \in [T]$:*

$$\mathbb{E} \left\|\boldsymbol{w}^{r,t+1} - \boldsymbol{w}^*\right\|^2 \leq (1 - \mu\gamma)\mathbb{E} \left\|\boldsymbol{w}^{r,t} - \boldsymbol{w}^*\right\|^2 - (2\gamma - 4\gamma^2 L)\mathbb{E} \left(F(\boldsymbol{w}^*) - F(\boldsymbol{w}^{r,t})\right)$$

$$+ (\gamma L + 2\gamma^2 L^2)\frac{1}{N} \sum_{i=1}^{N} \left\|\boldsymbol{w}_i^{r,t} - \boldsymbol{w}^{r,t}\right\|^2 + \gamma^2 \frac{\delta^2}{N}.$$

*Proof.* According to updating rule in Algorithm A1, we have the following identity:

$$\mathbb{E} \left\|\boldsymbol{w}^{r,t+1} - \boldsymbol{w}^*\right\|^2 = \mathbb{E} \left\|\boldsymbol{w}^{r,t} - \boldsymbol{w}^*\right\|^2 - 2\gamma\mathbb{E} \left\langle \frac{1}{N} \sum_{i=1}^{N} \nabla f_i(\boldsymbol{w}_i^{r,t}; z_i^{r,t}), \boldsymbol{w}^{r,t} - \boldsymbol{w}^* \right\rangle + \gamma^2 \mathbb{E} \left\|\frac{1}{N} \sum_{i=1}^{N} \nabla f_i(\boldsymbol{w}_i^{r,t})\right\|^2$$

$$\tag{9}$$

$$= \mathbb{E} \left\|\boldsymbol{w}^t - \boldsymbol{w}^*\right\|^2 \underbrace{- 2\gamma \left\langle \frac{1}{N} \sum_{i=1}^{N} \nabla f_i(\boldsymbol{w}_i^{r,t}), \boldsymbol{w}^{r,t} - \boldsymbol{w}^* \right\rangle}_{T_1}$$

$$\underbrace{+ \gamma^2 \mathbb{E} \left\|\frac{1}{N} \sum_{i=1}^{N} \nabla f_i(\boldsymbol{w}_i^{r,t}; z_i^{r,t})\right\|^2}_{T_2} + \gamma^2 \frac{\delta^2}{N}. \tag{10}$$

For $T_1$, since each $f_j$ is $L$ smooth and $\mu$ strongly convex, we have:

$$-2\gamma \left\langle \frac{1}{N}\sum_{i=1}^{N}\nabla f_i(\boldsymbol{w}_i^{r,t}), \boldsymbol{w}^t - \boldsymbol{w}^* \right\rangle = -2\gamma \left\langle \frac{1}{N}\sum_{i=1}^{N}\nabla f_i(\boldsymbol{w}_i^t), \boldsymbol{w}^t - \boldsymbol{w}_i^t + \boldsymbol{w}_i^{r,t} - \boldsymbol{w}^* \right\rangle$$

$$\leq -2\gamma \left\langle \frac{1}{N}\sum_{i=1}^{N}\nabla f_i(\boldsymbol{w}_i^t), \boldsymbol{w}^{r,t} - \boldsymbol{w}_i^{r,t} + \boldsymbol{w}_i^{r,t} - \boldsymbol{w}^* \right\rangle$$

$$\leq 2\gamma\frac{1}{N}\sum_{i=1}^{N}\left( f_i(\boldsymbol{w}^*) - f_i(\boldsymbol{w}^{r,t}) - \frac{\mu}{2}\left\| \boldsymbol{w}_i^{r,t} - \boldsymbol{w}^* \right\|^2 + \frac{L}{2}\left\| \boldsymbol{w}_i^{r,t} - \boldsymbol{w}^{r,t} \right\|^2 \right).$$

Due to Jensen's inequality we know: $-\frac{1}{N}\sum_{i=1}^{N}\frac{\mu}{2}\left\| \boldsymbol{w}_i^t - \boldsymbol{w}^* \right\|^2 \leq -\frac{\mu}{2}\left\| \boldsymbol{w}^t - \boldsymbol{w}^* \right\|^2$. Hence we know:

$$-2\gamma \left\langle \frac{1}{N}\sum_{i=1}^{N}\nabla f_i(\boldsymbol{w}_i^{r,t}), \boldsymbol{w}^{r,t} - \boldsymbol{w}^* \right\rangle \leq 2\gamma \left( F(\boldsymbol{w}^*) - F(\boldsymbol{w}^{r,t}) - \frac{\mu}{2}\left\| \boldsymbol{w}^{r,t} - \boldsymbol{w}^* \right\|^2 + \frac{L}{2}\frac{1}{N}\sum_{i=1}^{N}\left\| \boldsymbol{w}_i^{r,t} - \boldsymbol{w}^{r,t} \right\|^2 \right).$$

For $T_2$, we have:

$$\mathbb{E}\left\| \frac{1}{N}\sum_{i=1}^{N}\nabla f_i(\boldsymbol{w}_i^{r,t}) \right\|^2 = 2\mathbb{E}\left\| \frac{1}{N}\sum_{i=1}^{N}\nabla f_i(\boldsymbol{w}_i^{r,t}) - \nabla F(\boldsymbol{w}^{r,t}) \right\|^2 + 2\mathbb{E}\left\| \nabla F(\boldsymbol{w}^{r,t}) \right\|^2$$

$$\leq 2L^2\frac{1}{N}\sum_{i=1}^{N}\mathbb{E}\left\| \boldsymbol{w}_i^{r,t} - \boldsymbol{w}^{r,t} \right\|^2 + 4L\left( F(\boldsymbol{w}^{r,t}) - F(\boldsymbol{w}^*) \right).$$

Now, plugging $T_1$ and $T_2$ back to (10) yields:

$$\mathbb{E}\left\| \boldsymbol{w}^{r,t+1} - \boldsymbol{w}^* \right\|^2 \leq (1-\mu\gamma)\mathbb{E}\left\| \boldsymbol{w}^{r,t} - \boldsymbol{w}^* \right\|^2 - (2\gamma - 4\gamma^2 L)\mathbb{E}\left( F(\boldsymbol{w}^*) - F(\boldsymbol{w}^{r,t}) \right)$$

$$+ (\gamma L + 2\gamma^2 L^2)\frac{1}{N}\sum_{i=1}^{N}\left\| \boldsymbol{w}_i^{r,t} - \boldsymbol{w}^{r,t} \right\|^2 + \gamma^2\frac{\delta^2}{N}.$$

$\square$

**Lemma 5.** *[50, Lemma 8] For the iterates $\{\boldsymbol{w}_i^{r,t}\}$ generated in Algorithm A1, the following statement holds true:*

$$\frac{1}{N}\sum_{i=1}^{N}\left\| \boldsymbol{w}_i^{r,t} - \boldsymbol{w}^{r,t} \right\|^2 \leq 3K\gamma^2\delta^2 + 6K^2\gamma^2\zeta^2.$$

**Lemma 6** (Last iterate convergence of Local SGD). *Under the conditions of Theorem 1, the following statement holds true for the iterates in Algorithm A1:*

$$\mathbb{E}\left\| \boldsymbol{w}^R - \boldsymbol{w}^* \right\|^2 \leq (1-\mu\gamma)^{RK}\mathbb{E}\left\| \boldsymbol{w}^0 - \boldsymbol{w}^* \right\|^2 + \frac{1}{\mu\gamma}(\gamma L + 2\gamma^2 L^2)\left( 3K\gamma^2\delta^2 + 6K^2\gamma^2\zeta^2 \right) + \frac{\gamma\delta^2}{\mu N}$$

*Proof.* We first unroll the recursion in Lemma 4 from $t = K$ to $0$, within one communication round:

$$\mathbb{E}\left\| \boldsymbol{w}^{r,K} - \boldsymbol{w}^* \right\|^2 = (1-\mu\gamma)^K\mathbb{E}\left\| \boldsymbol{w}^{r,0} - \boldsymbol{w}^* \right\|^2 - \sum_{t=0}^{K-1}(1-\mu\gamma)^{K-t}(2\gamma - 4\gamma^2 L)\mathbb{E}\left( F(\boldsymbol{w}^*) - F(\boldsymbol{w}^{r,t}) \right)$$

$$+ \sum_{t=0}^{K-1}(1-\mu\gamma)^{K-t}(\gamma L + 2\gamma^2 L^2)\frac{1}{N}\sum_{i=1}^{N}\left\| \boldsymbol{w}_i^{r,t} - \boldsymbol{w}^{r,t} \right\|^2 + \sum_{t=0}^{K-1}(1-\mu\gamma)^{K-t}\gamma^2\frac{\delta^2}{N}$$

Since we choose $\gamma \leq \frac{1}{2L}$, we know $\sum_{t=0}^{K-1}(1-\mu\gamma)^{K-t}(2\gamma - 4\gamma^2 L)\mathbb{E}\left( F(\boldsymbol{w}^*) - F(\boldsymbol{w}^{r,t}) \right) \geq 0$. Plugging in Lemma 5 yields:

$$\mathbb{E}\left\| \boldsymbol{w}^R - \boldsymbol{w}^* \right\|^2 = (1-\mu\gamma)^{RK}\mathbb{E}\left\| \boldsymbol{w}^0 - \boldsymbol{w}^* \right\|^2 + \frac{1}{\mu\gamma}(\gamma L + 2\gamma^2 L^2)\left( 3K\gamma^2\delta^2 + 6K^2\gamma^2\zeta^2 \right) + \frac{\gamma\delta^2}{\mu N},$$

---

**Algorithm A2:** Shuffling Local SGD (`One Client`)

---

**Input:** Clients $0, ..., N-1$, Number of Local Steps $K$ , Number of Epoch $R$, Mixing parameter $\hat{\alpha}$

**Epoch for** $r = 0, ..., R-1$ **do**

    Server generates permutation $\sigma_r : [N] \mapsto [N]$.

    Client sets initial model $\boldsymbol{v}^{r,0} = \boldsymbol{v}^r$.

    **for** $j = 0, ..., N-1$ **do**

        Server sends $\boldsymbol{v}^{r,j}$ to Client $\sigma_r(j)$.

        $\boldsymbol{v}^{r,j+1} = $ `SGD-Update`$(\boldsymbol{v}^{r,j}, \eta, \sigma_r(j), K, \hat{\alpha})$.

    Client $i$ does projection: $\boldsymbol{v}^{r+1} = \mathcal{P}_{\mathcal{W}}(\boldsymbol{v}^{r,N})$.

**Output:** $\hat{v} = \boldsymbol{v}^R$.

`SGD-Update`$(\boldsymbol{v}, \eta, j, K, \boldsymbol{\alpha})$

    Initialize $\boldsymbol{v}^0 = \boldsymbol{v}$

    **for** $t = 0, ..., K-1$ **do**

        $\boldsymbol{v}^t = \boldsymbol{v}^{t-1} - \eta\alpha(j)N\nabla f_j(\boldsymbol{v}^{t-1}; \xi^{t-1})$

    **Output** $\boldsymbol{v}^K$

---

Plugging in $\gamma = \frac{\log(RK)}{\mu RK}$ gives the convergence rate:

$$\mathbb{E}\left\|\boldsymbol{w}^R - \boldsymbol{w}^*\right\|^2 \leq \tilde{O}\left(\frac{\mathbb{E}\left\|\boldsymbol{w}^0 - \boldsymbol{w}^*\right\|^2}{RK} + \kappa\left(\frac{\delta^2}{\mu^2 R^2 K} + \frac{\zeta^2}{\mu^2 R^2}\right) + \frac{\delta^2}{\mu^2 NRK}\right),$$

which concludes the proof. $\qquad\qquad\qquad\qquad\qquad\qquad\qquad\qquad\qquad\qquad\qquad\qquad\square$

Equipped with above results, we are now ready to provide the convergence of main theorem.

*Proof of Theorem 1.* The proof simply follows from Lemma 3:

$$\left\|\hat{\boldsymbol{\alpha}} - \boldsymbol{\alpha}^*\right\|^2 \leq 2\left\|\hat{\boldsymbol{\alpha}} - \boldsymbol{\alpha}_g^*(\boldsymbol{w}^R)\right\|^2 + 2\left\|\boldsymbol{\alpha}_g^*(\boldsymbol{w}^R) - \boldsymbol{\alpha}_g^*(\boldsymbol{w}^*)\right\|^2$$

$$\leq 2(1-\mu\eta_{\boldsymbol{\alpha}})^{T_{\boldsymbol{\alpha}}} + 8L\kappa_g^2 \sum_{j=1}^{N}\left(2\left\|\nabla f_i(\boldsymbol{w}^*) - \nabla f_j(\boldsymbol{w}^*)\right\|^2 + 4L^2\left\|\boldsymbol{w}^R - \boldsymbol{w}^*\right\|^2\right)\left\|\boldsymbol{w}^R - \boldsymbol{w}^*\right\|^2$$

$$\leq 2(1-\mu\eta_{\boldsymbol{\alpha}})^{T_{\boldsymbol{\alpha}}} + 8L\kappa_g^2\left(2\bar{\zeta}_i(\boldsymbol{w}^*) + 4NL^2\left\|\boldsymbol{w}^R - \boldsymbol{w}^*\right\|^2\right)\left\|\boldsymbol{w}^R - \boldsymbol{w}^*\right\|^2$$

Plugging in the convergence of $\left\|\boldsymbol{w}^R - \boldsymbol{w}^*\right\|^2$ from Lemma 6, and the stepsize $\eta_{\boldsymbol{\alpha}} = \frac{1}{L_g}$ for $\boldsymbol{\alpha}$ yields:

$$\mathbb{E}\|\boldsymbol{\alpha}_i^R - \boldsymbol{\alpha}_i^*\|^2 \leq \tilde{O}\left(\exp(-\frac{T_{\boldsymbol{\alpha}}}{\kappa_g}) + \kappa_g^2\bar{\zeta}_i(\boldsymbol{w}^*)L^2\left(\frac{D^2}{RK} + \kappa\left(\frac{\delta^2}{\mu^2 R^2 K} + \frac{\zeta^2}{\mu^2 R^2}\right) + \frac{\delta^2}{\mu^2 NRK}\right)\right).$$

$$\qquad\qquad\qquad\qquad\qquad\qquad\qquad\qquad\qquad\qquad\qquad\qquad\qquad\qquad\qquad\qquad\qquad\qquad\square$$

## B.3    Proof of Convergence of Shuffling Local SGD

In this section, we are going to prove the convergence of proposed shuffled variant of Local SGD (Theorem 2). The whole proof framework follows the analysis of vanilla shuffling SGD, but notice that there are two differences. First, in vanilla shuffling SGD, in each epoch, algorithm only updates on each component function $f_j$ once, while here we have to take $K$ steps of SGD update on each component function. Second, we are considering a weighted sum objective in contrary to averaged objective in [56], which means we need to rescale the objective when we apply without-replacement concentration inequality. Even though our algorithm solves models for $N$ clients, for the sake of simplicity, throughout the proof we only show the convergence of one client's model. The algorithm from one client point of view is described in Algorithm A2, where we drop the client index for notational convenience.

**Proposition 1.** *Assume a sequence* $\{\boldsymbol{w}^t\}_{t=1}^K$ *is obtained by*

$$\boldsymbol{w}^t = \boldsymbol{w}^{t-1} - \eta\alpha N\nabla f(\boldsymbol{w}^{t-1}; \xi^{t-1}), \ t = 1, \ldots, K,$$

*then we have*

$$\boldsymbol{w}^{t+1} = \boldsymbol{w}^0 - \left( \sum_{\tau=0}^{t} \prod_{t'=t}^{\tau+1} (\mathbf{I} - \alpha N \eta \mathbf{H}_{t'}) \right) \eta \alpha N \nabla f(\boldsymbol{w}^0) - \sum_{\tau=0}^{t} \prod_{t'=t}^{\tau+1} (\mathbf{I} - \alpha N \eta \mathbf{H}_{t'}) \eta \alpha N \boldsymbol{\delta}^t, \quad \forall 0 \le t \le K-1,$$

*where $\boldsymbol{\delta}^t := \nabla f(\boldsymbol{w}^t; \xi^t) - \nabla f(\boldsymbol{w}^t)$, and by convention, we define $\prod_{j=a}^{b} \boldsymbol{A}_j = \mathbf{I}$ if $a < b$.*

*Proof.* According to updating rule, we have:

$$
\begin{aligned}
\boldsymbol{w}^{t+1} - \boldsymbol{w}^0 &= \boldsymbol{w}^t - \boldsymbol{w}^0 - \eta \alpha N \nabla f(\boldsymbol{w}^t; \xi^t) \\
&= \boldsymbol{w}^t - \boldsymbol{w}^0 - \eta \alpha N \nabla f(\boldsymbol{w}^t) - \eta \alpha N \boldsymbol{\delta}^t \\
&= \boldsymbol{w}^t - \boldsymbol{w}^0 - \eta \alpha N \nabla f(\boldsymbol{w}^0) - \eta \alpha N (\nabla f(\boldsymbol{w}^t) - \nabla f(\boldsymbol{w}^0)) - \eta \alpha N \boldsymbol{\delta}^t.
\end{aligned}
$$

Since $f$ is $L$ smooth, and according to Mean Value Theorem, there is a matrix $\mathbf{H}_t$ satisfying $\mu \mathbf{I} \preceq \mathbf{H}_t \preceq L\mathbf{I}$, such that $\nabla f(\boldsymbol{w}^t) - \nabla f(\boldsymbol{w}^0) = \mathbf{H}_t(\boldsymbol{w}^t - \boldsymbol{w}^0)$. Hence we have:

$$\boldsymbol{w}^{t+1} - \boldsymbol{w}^0 = (\mathbf{I} - \eta \alpha N \mathbf{H}_t)(\boldsymbol{w}^t - \boldsymbol{w}^0) - \eta \alpha N \nabla f(\boldsymbol{w}^0) - \eta \alpha N \boldsymbol{\delta}^t.$$

Unrolling the recursion from $t$ to $0$ will conclude the proof. $\square$

The following lemma establishes the updating rule of models between epochs $r$ and $r+1$. For notational convenience, whenever there is no confusion, we drop the superscript $r$ in $\sigma^r$.

**Lemma 7** (One epoch updating rule). *Let $\boldsymbol{v}^r$ and $\boldsymbol{v}^{r+1}$ be two iterates generated by Shuffling Local SGD (Algorithm A2), then the following updating rule holds:*

$$\boldsymbol{v}^{r+1} = \boldsymbol{v}^r - \sum_{j=1}^{N} \prod_{j'=N}^{j+1} (\mathbf{I} - \boldsymbol{Q}_{j'} \mathbf{H}_{j'})(\boldsymbol{Q}_j \nabla f_{\sigma(j)}(\boldsymbol{v}^r) - \boldsymbol{\delta}_j),$$

*where*

$$\boldsymbol{Q}_j := \left( \sum_{\tau=0}^{K-1} \prod_{t'=K-1}^{\tau+1} (\mathbf{I} - \eta \hat{\alpha}(\sigma(j)) N \mathbf{H}_{t'}) \right) \eta \hat{\alpha}(\sigma(j)) N,$$

$$\boldsymbol{\delta}_j := \sum_{\tau=0}^{K-1} \prod_{t'=t}^{\tau+1} (\mathbf{I} - \hat{\alpha}(\sigma(j)) N \eta \mathbf{H}_{t'}) \eta \hat{\alpha}(\sigma(j)) N \boldsymbol{\delta}_{\sigma(j)}^t,$$

*by convention, we define $\prod_{j=a}^{b} \boldsymbol{A}_j = \mathbf{I}$ if $a < b$.*

*Proof.* According to Proposition 1, we have

$$
\begin{aligned}
\boldsymbol{v}^{r,j+1} = \boldsymbol{v}^{r,j} &- \left( \sum_{\tau=0}^{K-1} \prod_{t'=t}^{\tau+1} (\mathbf{I} - \hat{\alpha}(\sigma(j)) N \eta \mathbf{H}_{t'}) \right) \eta \hat{\alpha}(\sigma(j)) N \nabla f_{\sigma(j)}(\boldsymbol{v}^{r,j}) \\
&- \sum_{\tau=0}^{K-1} \prod_{t'=t}^{\tau+1} (\mathbf{I} - \hat{\alpha}(\sigma(j)) N \eta \mathbf{H}_{t'}) \eta \hat{\alpha}(\sigma(j)) N \boldsymbol{\delta}_{\sigma(j)}^t.
\end{aligned}
$$

Plugging our definition of $\boldsymbol{Q}_j$ and $\boldsymbol{\delta}_j$ yields:

$$\boldsymbol{v}^{r,j+1} - \boldsymbol{v}^r = \boldsymbol{v}^{r,j} - \boldsymbol{v}^r - \boldsymbol{Q}_j \nabla f_{\sigma(j)}(\boldsymbol{v}^{r,j}) - \boldsymbol{\delta}_j.$$

Following the same reasoning in the proof of Proposition 1 will conclude the proof. $\square$

**Lemma 8** (Summation by parts). *Let $\boldsymbol{A}_j$ and $\mathbf{B}_j$ be complex valued matrices. Then the following fact holds:*

$$\sum_{j=1}^{N} \boldsymbol{A}_j \mathbf{B}_j = \boldsymbol{A}_N \sum_{j=1}^{N} \mathbf{B}_j - \sum_{n=1}^{N-1} (\boldsymbol{A}_{n+1} - \boldsymbol{A}_n) \sum_{j=1}^{n} \mathbf{B}_j.$$

**Proposition 2** (Spectral bound of polynomial expansion). *Given a collection of matrices $\{A_t\}$ and $\{\mathbf{B}_t\}$, such that $A_t \preceq L\mathbf{I}$ and $\mathbf{B}_t \preceq L\mathbf{I}$, the following bound hold:*

$$\left\| \prod_{t=l}^{h} (\mathbf{I} - aA_t) - \mathbf{I} \right\| \leq \sum_{m=1}^{h-l} \left( \frac{e(h-l)}{m} \right)^m (aL)^m,$$

$$\left\| \prod_{t=l}^{h} (\mathbf{I} - aA_t) - \prod_{t=l}^{h} (\mathbf{I} - b\mathbf{B}_t) \right\| \leq \sum_{m=1}^{h-l} \left( \frac{e(h-l)}{m} \right)^m (aL)^m + \sum_{m=1}^{h-l} \left( \frac{e(h-l)}{m} \right)^m (bL)^m.$$

*Proof.* We start with proving the first statement. Expanding the product yields:

$$\prod_{t=l}^{h} (\mathbf{I} - aA_t) = \mathbf{I} + \sum_{m=1}^{h-l} (-1)^m a^m \sum_{|S|=m, |S| \subseteq \{l,\ldots,h\}} \prod_{m' \in S} A_{m'}.$$

Hence we have:

$$\left\| \prod_{t=l}^{h} (\mathbf{I} - aA_t) - \mathbf{I} \right\| = \left\| \sum_{m=1}^{h-l} (-1)^m a^m \sum_{|S|=m, |S| \subseteq \{l,\ldots,h\}} \prod_{m' \in S} A_{m'} \right\| \leq \sum_{m=1}^{h-l} \binom{h-l}{m} (aL)^m$$

According to the upper bound for binomial coefficients: $\binom{h-l}{m} \leq \left( \frac{e(h-l)}{m} \right)^m$, we have:

$$\sum_{m=1}^{h-l} \binom{h-l}{m} (aL)^m \leq \sum_{m=1}^{h-l} \left( \frac{e(h-l)}{m} \right)^m (aL)^m.$$

Then we switch to the second one. Using the same expanding product yields:

$$\left\| \prod_{t=l}^{h} (\mathbf{I} - aA_t) - \prod_{t=l}^{h} (\mathbf{I} - b\mathbf{B}_t) \right\|$$

$$= \left\| \sum_{m=1}^{h-l} (-1)^m a^m \sum_{|S|=m, |S| \subseteq \{l,\ldots,h\}} \prod_{m' \in S} A_{m'} - \sum_{m=1}^{h-l} (-1)^m b^m \sum_{|S|=m, |S| \subseteq \{l,\ldots,h\}} \prod_{m' \in S} \mathbf{B}_{m'} \right\|$$

$$\leq \sum_{m=1}^{h-l} \binom{h-l}{m} (aL)^m + \sum_{m=1}^{h-l} \binom{h-l}{m} (bL)^m$$

$$\leq \sum_{m=1}^{h-l} \left( \frac{e(h-l)}{m} \right)^m (aL)^m + \sum_{m=1}^{h-l} \left( \frac{e(h-l)}{m} \right)^m (bL)^m.$$

$\square$

The following concentration result is the key to bound variance during shuffling updating. The original result holds for the average of gradients, and we will later on generalize it to an arbitrary weighted sum of gradients.

**Lemma 9** ([61, Theorem 2]). *Suppose $n \geq 2$. Let $g_1, g_2, \ldots, g_n \in \mathbb{R}^d$ satisfy $\|g_j\| \leq G$ for all $j$. Let $\bar{g} = \frac{1}{n} \sum_{j=1}^{n} g_j$. Let $\sigma \in S_n$ be a uniform random permutation of $n$ elements. Then, for $i \leq n$, with probability at least $1 - p$, we have*

$$\left\| \frac{1}{i} \sum_{j=1}^{i} g_{\sigma(j)} - \bar{g} \right\| \leq G \sqrt{\frac{8(1 - \frac{i-1}{n}) \log \frac{2}{p}}{i}}.$$

**Lemma 10** (Concentration of partial sum of gradients). *Given a uniformly randomly generated permutation $\sigma$, and simplex vector $\boldsymbol{\alpha}$, if we assume each $\sup_{\boldsymbol{v} \in \mathcal{W}} \|\nabla f_j(\boldsymbol{v})\| \leq G$, then the following statement holds true:*

$$\left\| \sum_{j=0}^{n} \hat{\alpha}(\sigma(j)) \nabla f_{\sigma(j)}(\boldsymbol{v}^r) \right\| \leq G \sqrt{8n \log(1/p)} + \frac{n}{N} \| \nabla \Phi(\hat{\boldsymbol{\alpha}}, \boldsymbol{v}^r) \|.$$

*Proof.* The proof works by re-writing weighted sum of vectors to average of the these vectors:

$$\left\| \sum_{j=0}^{n} \hat{\alpha}(\sigma(j)) \nabla f_{\sigma(j)}(\boldsymbol{v}^r) \right\| = \frac{1}{N} \left\| \sum_{j=0}^{n} \hat{\alpha}(\sigma(j)) N \nabla f_{\sigma(j)}(\boldsymbol{v}^r) \right\|$$

$$= \frac{1}{N} \left( \left\| \sum_{j=0}^{n} \hat{\alpha}(\sigma(j)) N \nabla f_{\sigma(j)}(\boldsymbol{v}^r) - n \nabla \Phi(\hat{\boldsymbol{\alpha}}, \boldsymbol{v}^r) \right\| + n \left\| \nabla \Phi(\hat{\boldsymbol{\alpha}}, \boldsymbol{v}^r) \right\| \right)$$

$$\leq G\sqrt{8n \log(1/p)} + \frac{n}{N} \left\| \nabla \Phi(\hat{\boldsymbol{\alpha}}, \boldsymbol{v}^r) \right\|.$$

$\square$

**Proposition 3** (Spectral norm bound of $\boldsymbol{Q}$)**.** *Let $\boldsymbol{Q}_j$ be defined in* (11)*. Then the following bound for the spectral norm of $\boldsymbol{Q}_j$ holds true for all $j \in [N]$:*

$$\|\boldsymbol{Q}_j\| \leq \eta \hat{\alpha}(\sigma(j)) N K (1 + \eta N L)^K$$

*Proof.* The proof can be completed by writing down the definitin of $\boldsymbol{Q}_j$ and applying Cauchy-Schwartz inequality:

$$\|\boldsymbol{Q}_j\| = \left\| \left( \sum_{\tau=0}^{K-1} \prod_{t'=K-1}^{\tau+1} (\mathbf{I} - \eta \hat{\alpha}(\sigma(j)) N' \mathbf{H}_{t'}) \right) \eta \hat{\alpha}(\sigma(j)) N \right\|$$

$$\leq \eta \hat{\alpha}(\sigma(j)) N \sum_{\tau=0}^{K-1} \prod_{t'=K-1}^{\tau+1} \|(\mathbf{I} - \eta \hat{\alpha}(\sigma(j)) N' \mathbf{H}_{t'})\|$$

$$\leq \eta \hat{\alpha}(\sigma(j)) N \sum_{\tau=0}^{K-1} \prod_{t'=K-1}^{\tau+1} (1 + \eta \hat{\alpha}(\sigma(j)) N' L)$$

$$\leq \eta \hat{\alpha}(\sigma(j)) N K (1 + \eta N L)^K.$$

The last step is due to we choose $\eta$ such that $\eta N L \leq \frac{1}{K}$.

$\square$

The following lemma establishes the bound regarding cumulative update between two epochs, namely, $\boldsymbol{v}^{r+1} - \boldsymbol{v}^r$. In particular, Lemma 11 below shows that: (a) in shuffling Local SGD, our update from $\boldsymbol{v}^r$ to $\boldsymbol{v}^{r+1}$ approximates performing $NK$ times of gradient descent with $\hat{\alpha}(j) N \nabla f_{\sigma(j)}(\boldsymbol{v}^r)$, namely, the bias is controlled, and (b) the update itself is bounded, and can be related to the norm of full gradient.

**Lemma 11.** *During the dynamic of Algorithm A2, the following statements hold true with probability at least $1 - p$:*

*(a)*

$$\left\| \sum_{j=1}^{N} \boldsymbol{Q}_j \nabla f_{\sigma(j)}(\boldsymbol{v}^r) - \eta N K \sum_{j=1}^{N} \hat{\alpha}(j) \nabla f_{\sigma(j)}(\boldsymbol{v}^r) \right\|^2 \leq 10\eta^2 N^2 K^2 \left( \frac{e}{4R - e} \right)^2 \|\nabla \Phi(\hat{\boldsymbol{\alpha}}, \boldsymbol{v}^r)\|^2$$

$$+ 128\eta^2 N^3 K^2 \left( \frac{e}{4R - e} \right)^2 G^2 \log(1/p).$$

*(b) for any $N'$ such that $0 \leq N' < N$*

$$\left\| \sum_{j=1}^{N'-1} \boldsymbol{Q}_j \nabla f_{\sigma(j)}(\boldsymbol{v}^r) \right\| \leq 3e\eta N K \left( \|\nabla \Phi(\hat{\boldsymbol{\alpha}}, \boldsymbol{v}^r)\| + G\sqrt{8N \log(1/p)} \right),$$

*where*

$$\boldsymbol{Q}_j := \left( \sum_{\tau=0}^{K-1} \prod_{t'=K-1}^{\tau+1} (\mathbf{I} - \eta \hat{\alpha}(\sigma(j)) N' \mathbf{H}_{t'}) \right) \eta \hat{\alpha}(\sigma(j)) N. \tag{11}$$

*Proof.* We start with proving statement (a). Let $\boldsymbol{A}_j = \frac{\boldsymbol{Q}_j}{\hat{\alpha}(\sigma(j))}$ and $\mathbf{B}_j = \hat{\alpha}(\sigma(j))\nabla f_{\sigma(j)}(\boldsymbol{v}^r)$, applying the identity of summation by parts yields:

$$\sum_{j=1}^{N} \boldsymbol{Q}_j \nabla f_{\sigma(j)}(\boldsymbol{v}^r) = \frac{\boldsymbol{Q}_{N-1}}{\hat{\alpha}(\sigma(N-1))} \sum_{j=1}^{N} \hat{\alpha}(\sigma(j))\nabla f_{\sigma(j)}(\boldsymbol{v}^r) - \sum_{n=1}^{N-1} \left( \frac{\boldsymbol{Q}_{n+1}}{\hat{\alpha}(\sigma(n+1))} - \frac{\boldsymbol{Q}_n}{\hat{\alpha}(\sigma(n))} \right) \sum_{j=1}^{n} \hat{\alpha}(\sigma(j))\nabla f_{\sigma(j)}(\boldsymbol{v}^r)$$

$$\left\| \sum_{j=1}^{N} \boldsymbol{Q}_j \nabla f_{\sigma(j)}(\boldsymbol{v}) - \eta NK \sum_{j=1}^{N} \hat{\alpha}(j)\nabla f_j(\boldsymbol{v}) \right\|^2$$

$$\leq 2 \underbrace{\left\| \left( \frac{\boldsymbol{Q}_{N-1}}{\hat{\alpha}(\sigma(N-1))} - \eta NK\mathbf{I} \right) \sum_{j=1}^{N} \hat{\alpha}(\sigma(j))\nabla f_{\sigma(j)}(\boldsymbol{v}^r) \right\|^2}_{T_1} + 2 \underbrace{\left\| \sum_{n=1}^{N-1} \left( \frac{\boldsymbol{Q}_{n+1}}{\hat{\alpha}(\sigma(n+1))} - \frac{\boldsymbol{Q}_n}{\hat{\alpha}(\sigma(n))} \right) \sum_{j=1}^{n} \hat{\alpha}(\sigma(j))\nabla f_{\sigma(j)}(\boldsymbol{v}^r) \right\|^2}_{T_2}.$$

According to Proposition 2, we have:

$$\left\| \prod_{t'=K-1}^{\tau+1} (\mathbf{I} - \eta\hat{\alpha}(\sigma(j))N\mathbf{H}_{t'}) - \mathbf{I} \right\| \leq \sum_{m=1}^{K-2-\tau} \left( \frac{e(K-2-\tau)}{m} \eta\hat{\alpha}(\sigma(j))NL \right)^m.$$

Since we choose $\eta \leq \frac{1}{4NKRL}$, we have:

$$\left\| \prod_{t'=K-1}^{\tau+1} (\mathbf{I} - \eta\hat{\alpha}(\sigma(j))N\mathbf{H}_{t'}) - \mathbf{I} \right\| \leq \sum_{m=1}^{K-2-\tau} \left( \frac{e}{4Rm} \right)^m \leq \frac{e}{4R-e}, \tag{12}$$

where we use the fact that $\sum_{m=1}^{K-2-\tau} \left( \frac{e}{4Rm} \right)^m \leq \sum_{m=1}^{K-2-\tau} \left( \frac{e}{4R} \right)^m \leq \frac{e}{4R} \frac{1}{1-e/4R}$. Hence we know:

$$T_1 \leq \left\| \left( \frac{\boldsymbol{Q}_{N-1}}{\hat{\alpha}(\sigma(N-1))} - \eta NK\mathbf{I} \right) \right\|^2 \left\| \sum_{j=1}^{N} \hat{\alpha}(\sigma(j))\nabla f_{\sigma(j)}(\boldsymbol{v}^r) \right\|^2$$

$$\leq \left\| \left( \sum_{\tau=0}^{K-1} \prod_{t'=K-1}^{\tau+1} (\mathbf{I} - \eta\hat{\alpha}(\sigma(N-1))N\mathbf{H}_{t'}) \right) \eta N - \eta NK\mathbf{I} \right\|^2 \left\| \sum_{j=1}^{N} \hat{\alpha}(\sigma(j))\nabla f_{\sigma(j)}(\boldsymbol{v}^r) \right\|^2$$

$$\leq \eta^2 N^2 K \sum_{\tau=0}^{K-1} \left\| \prod_{t'=K-1}^{\tau+1} (\mathbf{I} - \eta\hat{\alpha}(\sigma(N-1))N\mathbf{H}_{t'}) - \mathbf{I} \right\|^2 \left\| \sum_{j=1}^{N} \hat{\alpha}(\sigma(j))\nabla f_{\sigma(j)}(\boldsymbol{v}^r) \right\|^2$$

$$\leq \eta^2 N^2 K^2 \left( \frac{e}{4R-e} \right)^2 \left\| \sum_{j=1}^{N} \hat{\alpha}(\sigma(j))\nabla f_{\sigma(j)}(\boldsymbol{v}^r) \right\|^2.$$

Thus we have:

$$T_1 \leq \eta^2 N^2 K^2 \left( \frac{e}{4R-e} \right)^2 \|\nabla\Phi(\hat{\boldsymbol{\alpha}}, \boldsymbol{v}^r)\|^2.$$

For $T_2$, we first examine the bound of $\frac{\boldsymbol{Q}_{n+1}}{\hat{\alpha}(\sigma(n+1))} - \frac{\boldsymbol{Q}_n}{\hat{\alpha}(\sigma(n))}$:

$$\left\| \frac{\boldsymbol{Q}_{n+1}}{\hat{\alpha}(\sigma(n+1))} - \frac{\boldsymbol{Q}_n}{\hat{\alpha}(\sigma(n))} \right\| = \left\| \left( \sum_{\tau=0}^{K-1} \prod_{t'=K-1}^{\tau+1} (\mathbf{I} - \eta\hat{\alpha}(\sigma(n+1))N\mathbf{H}_{t'}) \right) \eta N - \left( \sum_{\tau=0}^{K-1} \prod_{t'=K-1}^{\tau+1} (\mathbf{I} - \eta\hat{\alpha}(\sigma(n))N\mathbf{H}_{t'}) \right) \eta N \right\|$$

$$= \eta N \left\| \left( \sum_{\tau=0}^{K-1} \prod_{t'=K-1}^{\tau+1} (\mathbf{I} - \eta\hat{\alpha}(\sigma(n+1))N\mathbf{H}_{t'}) \right) - \left( \sum_{\tau=0}^{K-1} \prod_{t'=K-1}^{\tau+1} (\mathbf{I} - \eta\hat{\alpha}(\sigma(n))N\mathbf{H}_{t'}) \right) \right\|$$

$$= \eta N \left\| \left( \sum_{\tau=0}^{K-1} \prod_{t'=K-1}^{\tau+1} (\mathbf{I} - \eta\hat{\alpha}(\sigma(n+1))N\mathbf{H}_{t'}) \right) - \left( \sum_{\tau=0}^{K-1} \prod_{t'=K-1}^{\tau+1} (\mathbf{I} - \eta\hat{\alpha}(\sigma(n))N\mathbf{H}_{t'}) \right) \right\|$$

$$\leq \eta N \sum_{\tau=0}^{K-1} \left( \sum_{m=1}^{K-2-\tau} \left( \frac{e(K-2-\tau)}{m} \eta\hat{\alpha}(\sigma(n))NL \right)^m + \sum_{m=1}^{K-2-\tau} \left( \frac{e(K-2-\tau)}{m} \eta\hat{\alpha}(\sigma(n+1))NL \right)^m \right).$$

where we evoke Proposition 2 at last step. Given that $\eta \leq \frac{1}{4NKRL}$ we have:

$$\left\|\frac{\boldsymbol{Q}_{n+1}}{\hat{\alpha}(\sigma(n+1))} - \frac{\boldsymbol{Q}_n}{\hat{\alpha}(\sigma(n))}\right\| \leq \eta N \sum_{\tau=0}^{K-1}\left(\sum_{m=1}^{K-2-\tau}\left(\frac{e}{4Rm}\hat{\alpha}(\sigma(n))\right)^m + \sum_{m=1}^{K-2-\tau}\left(\frac{e}{4Rm}\hat{\alpha}(\sigma(n+1))L\right)^m\right)$$

$$\leq \eta NK\left(\frac{\hat{\alpha}(\sigma(n))e}{4R-e} + \frac{\hat{\alpha}(\sigma(n+1))e}{4R-e}\right).$$

where we use the reasoining in (12). Hence for $\sqrt{T_2}$:

$$\sqrt{T_2} \leq \eta NK \sum_{n=1}^{N-1}\left(\frac{\hat{\alpha}(\sigma(n))e}{4R-e} + \frac{\hat{\alpha}(\sigma(n+1))e}{4R-e}\right)\left\|\sum_{j=1}^{n}\hat{\alpha}(\sigma(j))\nabla f_{\sigma(j)}(\boldsymbol{v}^r)\right\|$$

$$\leq \eta NK\frac{e}{4R-e}\sum_{n=1}^{N-1}\left(\hat{\alpha}(\sigma(n)) + \hat{\alpha}(\sigma(n+1))\right)\left(G\sqrt{8n\log(1/p)} + \frac{n}{N}\|\nabla\Phi(\hat{\boldsymbol{\alpha}},\boldsymbol{v}^r)\|\right)$$

$$\leq \eta NK\frac{2e}{4R-e}\left(G\sqrt{8N\log(1/p)} + \|\nabla\Phi(\hat{\boldsymbol{\alpha}},\boldsymbol{v}^r)\|\right).$$

where at last step we evoke Lemma 10. So we can conclude $T_2 \leq 2\eta^2 N^2 K^2\left(\frac{2e}{4R-e}\right)^2\left(G^2 8N\log(1/p) + \|\nabla\Phi(\hat{\boldsymbol{\alpha}},\boldsymbol{v}^r)\|^2\right)$. Putting the bounds of $T_1$ and $T_2$ together will conclude the proof for (a).

Now we switch to proving (b). Once again by the summation of parts identity we have:

$$\sum_{j=1}^{N'}\boldsymbol{Q}_j\nabla f_{\sigma(j)}(\boldsymbol{v}^r) = \frac{\boldsymbol{Q}_{N'}}{\hat{\alpha}(\sigma(N'))}\sum_{j=1}^{N'}\hat{\alpha}(\sigma(j))\nabla f_{\sigma(j)}(\boldsymbol{v}^r) - \sum_{n=1}^{N'-1}\left(\frac{\boldsymbol{Q}_{n+1}}{\hat{\alpha}(\sigma(n+1))} - \frac{\boldsymbol{Q}_n}{\hat{\alpha}(\sigma(n))}\right)\sum_{j=1}^{n}\hat{\alpha}(\sigma(j))\nabla f_{\sigma(j)}(\boldsymbol{v}^r).$$

Taking the norm of both side yields:

$$\left\|\sum_{j=1}^{N'}\boldsymbol{Q}_j\nabla f_{\sigma(j)}(\boldsymbol{v}^r)\right\| = \underbrace{\left\|\frac{\boldsymbol{Q}_{N'}}{\hat{\alpha}(\sigma(N'))}\sum_{j=1}^{N'}\hat{\alpha}(\sigma(j))\nabla f_{\sigma(j)}(\boldsymbol{v}^r)\right\|}_{B}$$

$$+ \underbrace{\left\|\sum_{n=1}^{N'-1}\left(\frac{\boldsymbol{Q}_{n+1}}{\hat{\alpha}(\sigma(n+1))} - \frac{\boldsymbol{Q}_n}{\hat{\alpha}(\sigma(n))}\right)\sum_{j=1}^{n}\hat{\alpha}(\sigma(j))\nabla f_{\sigma(j)}(\boldsymbol{v}^r)\right\|}_{C}.$$

Plugging our developed bound for $\|\boldsymbol{Q}_{N'}\|$ and $\left\|\sum_{n=1}^{N'-1}\left(\frac{\boldsymbol{Q}_{n+1}}{\hat{\alpha}(\sigma(n+1))} - \frac{\boldsymbol{Q}_n}{\hat{\alpha}(\sigma(n))}\right)\right\|$ yields:

$$B \leq \left\|\frac{\boldsymbol{Q}_{N'}}{\hat{\alpha}(\sigma(N'))}\right\|\left\|\sum_{j=1}^{N'}\hat{\alpha}(\sigma(j))\nabla f_{\sigma(j)}(\boldsymbol{v}^r)\right\|$$

$$\leq \eta NK(1+\eta NL)^K\left(G\sqrt{8N'\log(1/p)} + \frac{N'}{N}\|\nabla\Phi(\hat{\boldsymbol{\alpha}},\boldsymbol{v}^r)\|\right).$$

where at last step we evoke Lemma 10. And for C, we use the similar reasoning:

$$C \leq \sum_{n=1}^{N'-1}\left\|\left(\frac{\boldsymbol{Q}_{n+1}}{\hat{\alpha}(\sigma(n+1))} - \frac{\boldsymbol{Q}_n}{\hat{\alpha}(\sigma(n))}\right)\right\|\left\|\sum_{j=1}^{n}\hat{\alpha}(\sigma(j))\nabla f_{\sigma(j)}(\boldsymbol{v}^r)\right\|$$

$$\leq \sum_{n=1}^{N'-1}\eta NK\frac{e}{4R-e}\left(\hat{\alpha}(\sigma(n+1)) + \hat{\alpha}(\sigma(n))\right)\left(G\sqrt{8n\log(1/p)} + \frac{n}{N}\|\nabla\Phi(\hat{\boldsymbol{\alpha}},\boldsymbol{v}^r)\|\right)$$

$$\leq 2\eta NK\frac{e}{4R-e}\left(G\sqrt{8N\log(1/p)} + \|\nabla\Phi(\hat{\boldsymbol{\alpha}},\boldsymbol{v}^r)\|\right).$$

Putting these pieces together yields:

$$\left\| \sum_{j=1}^{N'} \boldsymbol{Q}_j \nabla f_{\sigma(j)}(\boldsymbol{v}^r) \right\| \le 3e\eta NK \left( \|\nabla\Phi(\hat{\boldsymbol{\alpha}}, \boldsymbol{v}^r)\| + G\sqrt{8N\log(1/p)} \right).$$

$\square$

**Lemma 12.** *During the dynamic of Algorithm A2, the following statements hold true with probability at least $1 - p$:*

$$\left\| \sum_{n=1}^{N-1} \left( \prod_{j'=N}^{n+2} (\mathbf{I} - \boldsymbol{Q}_{j'}\mathbf{H}_{j'}) \right) \boldsymbol{Q}_{n+1}\mathbf{H}_{n+1} \sum_{j=1}^{n} \boldsymbol{Q}_j \nabla f_{\sigma(j)}(\boldsymbol{v}^r) \right\|^2$$

$$\le 18e^6 \eta^4 N^4 K^4 L^4 \left( \|\nabla\Phi(\hat{\boldsymbol{\alpha}}, \boldsymbol{v}^r)\|^2 + 8G^2 N \log(1/p) \right)$$

*Proof.* We first apply Cauchy-Schwartz inequality:

$$\left\| \sum_{n=1}^{N-1} \left( \prod_{j'=N}^{n+2} (\mathbf{I} - \boldsymbol{Q}_{j'}\mathbf{H}_{j'}) \right) \boldsymbol{Q}_{n+1}\mathbf{H}_{n+1} \sum_{j=1}^{n} \boldsymbol{Q}_j \nabla f_{\sigma(j)}(\boldsymbol{v}^r) \right\|$$

$$\le \sum_{n=1}^{N-1} \left\| \left( \prod_{j'=N}^{n+2} (\mathbf{I} - \boldsymbol{Q}_{j'}\mathbf{H}_{j'}) \right) \right\| \|\boldsymbol{Q}_{n+1}\mathbf{H}_{n+1}\| \left\| \sum_{j=1}^{n} \boldsymbol{Q}_j \nabla f_{\sigma(j)}(\boldsymbol{v}^r) \right\|$$

$$\le \left( 1 + \eta NK + \eta^2 N^2 KL \right)^{2N} \eta NLK (1 + \eta NL)^K L \sum_{n=1}^{N-1} \hat{\alpha}(\sigma(n+1)) \left\| \sum_{j=1}^{n} \boldsymbol{Q}_j \nabla f_{\sigma(j)}(\boldsymbol{v}^r) \right\|$$

$$\le e^2 \eta NKL^2 \sum_{n=1}^{N-1} \hat{\alpha}(\sigma(n+1)) \left\| \sum_{j=1}^{n} \boldsymbol{Q}_j \nabla f_{\sigma(j)}(\boldsymbol{v}^r) \right\|.$$

We proceed by applying the bound from Lemma 11 (b):

$$\left\| \sum_{j=1}^{n} \boldsymbol{Q}_j \nabla f_{\sigma(j)}(\boldsymbol{v}^r) \right\| \le 3e\eta NK \left( \|\nabla\Phi(\hat{\boldsymbol{\alpha}}, \boldsymbol{v}^r)\| + G\sqrt{8N\log(1/p)} \right).$$

Therefore, it follows that:

$$\left\| \sum_{n=1}^{N-1} \left( \prod_{j'=N}^{n+2} (\mathbf{I} - \boldsymbol{Q}_{j'}\mathbf{H}_{j'}) \right) \boldsymbol{Q}_{n+1}\mathbf{H}_{n+1} \sum_{j=1}^{n} \boldsymbol{Q}_j \nabla f_{\sigma(j)}(\boldsymbol{v}^r) \right\|$$

$$\le e^2 \eta NKL^2 \sum_{n=1}^{N-1} \hat{\alpha}(\sigma(n+1)) \cdot 3e\eta NK \left( \|\nabla\Phi(\hat{\boldsymbol{\alpha}}, \boldsymbol{v}^r)\| + G\sqrt{8N\log(1/p)} \right)$$

$$\le 3e^3 \eta^2 N^2 K^2 L^2 \left( \|\nabla\Phi(\hat{\boldsymbol{\alpha}}, \boldsymbol{v}^r)\| + G\sqrt{8N\log(1/p)} \right)$$

$\square$

**Lemma 13** (Noise bound). *During the dynamic of Algorithm A2, the following statement for gradient noises holds true with probability at least $1 - p$:*

$$\mathbb{E}\left\| \sum_{j=1}^{N} \prod_{j'=N}^{j+1} (\mathbf{I} - \boldsymbol{Q}_{j'}\mathbf{H}_{j'})\boldsymbol{\delta}_j \right\| \le \eta NK e^2 \delta,$$

*where*

$$\boldsymbol{\delta}_j := \sum_{\tau=0}^{K-1} \prod_{t'=t}^{\tau+1} (\mathbf{I} - \hat{\alpha}(\sigma(j))N\eta\mathbf{H}_{t'})\, \eta\hat{\alpha}(\sigma(j))N\boldsymbol{\delta}_{\sigma(j)}^t.$$

*Proof.* According to triangle and Cauchy-Schwartz inequalities we have:

$$\left\| \sum_{j=1}^{N} \prod_{j'=N}^{j+1} (\mathbf{I} - \boldsymbol{Q}_{j'}\mathbf{H}_{j'})\boldsymbol{\delta}_j \right\| \leq \sum_{j=1}^{N} \prod_{j'=N}^{j+1} \|(\mathbf{I} - \boldsymbol{Q}_{j'}\mathbf{H}_{j'})\| \, \|\boldsymbol{\delta}_j\|$$

$$\leq \sum_{j=1}^{N} \left(1 + (\eta\hat{\alpha}(\sigma(j))NK(1+\eta NL)^K)L\right)^N \|\boldsymbol{\delta}_j\|$$

$$\leq \sum_{j=1}^{N} \underbrace{\left(1 + (\eta\hat{\alpha}(\sigma(j))NK(1+\eta NL)^K)L\right)^N}_{\leq e} \cdot \eta\hat{\alpha}(\sigma(j))NK \underbrace{(1+\eta NL)^K}_{\leq e} \delta$$

$$\leq \eta NK e^2 \delta.$$

$\square$

## B.4 Proof of Theorem 2

*Proof.* For notational convenience, let us define

$$\boldsymbol{g}^r := \sum_{j=1}^{N} \prod_{j'=N}^{j+1} (\mathbf{I} - \boldsymbol{Q}_{j'}\mathbf{H}_{j'})\boldsymbol{Q}_j \nabla f_{\sigma(j)}(\boldsymbol{v}^r),$$

$$\boldsymbol{\delta}^r := \sum_{j=1}^{N} \prod_{j'=N}^{j+1} (\mathbf{I} - \boldsymbol{Q}_{j'}\mathbf{H}_{j'})\boldsymbol{\delta}_j.$$

Then we recall the updating rule of $\boldsymbol{v}$ (Lemma 7):

$$\boldsymbol{v}^{r+1} = \mathcal{P}_{\mathcal{W}}\left(\boldsymbol{v}^r - \boldsymbol{g}^r - \boldsymbol{\delta}^r\right)$$

Hence we have:

$$\mathbb{E}\left\|\boldsymbol{v}^{r+1} - \boldsymbol{v}^*(\hat{\boldsymbol{\alpha}})\right\|^2 = \mathbb{E}\left\|\mathcal{P}_{\mathcal{W}}\left(\boldsymbol{v}^r - \boldsymbol{g}^r - \boldsymbol{\delta}^r - \boldsymbol{v}^*(\hat{\boldsymbol{\alpha}})\right)\right\|^2$$

$$\leq \mathbb{E}\left\|\boldsymbol{v}^r - \boldsymbol{g}^r - \boldsymbol{\delta}^r - \boldsymbol{v}^*(\hat{\boldsymbol{\alpha}})\right\|^2$$

$$\leq \mathbb{E}\left\|\boldsymbol{v}^r - \boldsymbol{v}^*(\hat{\boldsymbol{\alpha}})\right\|^2 - 2\mathbb{E}\langle \boldsymbol{g}^r, \boldsymbol{v}^r - \boldsymbol{v}^*(\hat{\boldsymbol{\alpha}})\rangle + \mathbb{E}\left\|\boldsymbol{g}^r\right\|^2 + \mathbb{E}\left\|\boldsymbol{\delta}^r\right\|^2$$

$$\leq \mathbb{E}\left\|\boldsymbol{v}^r - \boldsymbol{v}^*(\hat{\boldsymbol{\alpha}})\right\|^2 - 2\mathbb{E}\langle \eta NK\nabla\Phi(\hat{\boldsymbol{\alpha}}, \boldsymbol{v}^r), \boldsymbol{v}^r - \boldsymbol{v}^*(\hat{\boldsymbol{\alpha}})\rangle - 2\mathbb{E}\langle \boldsymbol{g}^r - \eta NK\nabla\Phi(\hat{\boldsymbol{\alpha}}, \boldsymbol{v}^r), \boldsymbol{v}^r - \boldsymbol{v}^*(\hat{\boldsymbol{\alpha}})\rangle$$

$$+ \mathbb{E}\left\|\boldsymbol{g}^r\right\|^2 + \mathbb{E}\left\|\boldsymbol{\delta}^r\right\|^2.$$

Now, applying strongly convexity of $\Phi(\hat{\boldsymbol{\alpha}}, \cdot)$ and Cauchy-Schwartz inequality yields:

$$\mathbb{E}\left\|\boldsymbol{v}^{r+1} - \boldsymbol{v}^*(\hat{\boldsymbol{\alpha}})\right\|^2 \leq (1 - \mu\eta NK)\mathbb{E}\left\|\boldsymbol{v}^r - \boldsymbol{v}^*(\hat{\boldsymbol{\alpha}})\right\|^2 - \eta NK\mathbb{E}[\Phi(\hat{\boldsymbol{\alpha}}, \boldsymbol{v}^r) - \Phi(\hat{\boldsymbol{\alpha}}, \boldsymbol{v}^*(\hat{\boldsymbol{\alpha}}))]$$

$$+ \frac{1}{2}\left(\frac{1}{\mu\eta NK}\mathbb{E}\|\boldsymbol{g}^r - \eta NK\nabla\Phi(\hat{\boldsymbol{\alpha}}, \boldsymbol{v}^r)\|^2 + \mu\eta NK\mathbb{E}\|\boldsymbol{v}^r - \boldsymbol{v}^*(\hat{\boldsymbol{\alpha}})\|^2\right)$$

$$+ \mathbb{E}\left\|\boldsymbol{g}^r\right\|^2 + \mathbb{E}\left\|\boldsymbol{\delta}^r\right\|^2$$

$$\leq (1 - \frac{1}{2}\mu\eta NK)\mathbb{E}\left\|\boldsymbol{v}^r - \boldsymbol{v}^*(\hat{\boldsymbol{\alpha}})\right\|^2 - \eta NK\mathbb{E}[\Phi(\hat{\boldsymbol{\alpha}}, \boldsymbol{v}^r) - \Phi(\hat{\boldsymbol{\alpha}}, \boldsymbol{v}^*(\hat{\boldsymbol{\alpha}}))]$$

$$+ \frac{1}{2\mu\eta NK}\mathbb{E}\|\boldsymbol{g}^r - \eta NK\nabla\Phi(\hat{\boldsymbol{\alpha}}, \boldsymbol{v}^r)\|^2$$

$$+ 2\mathbb{E}\left\|\boldsymbol{g}^r - \eta NK\nabla\Phi(\hat{\boldsymbol{\alpha}}, \boldsymbol{v}^r)\right\|^2 + 2\mathbb{E}\left\|\eta NK\nabla\Phi(\hat{\boldsymbol{\alpha}}, \boldsymbol{v}^r)\right\|^2 + \mathbb{E}\left\|\boldsymbol{\delta}^r\right\|^2.$$

Since $\Phi(\hat{\boldsymbol{\alpha}}, \cdot)$ is $L$ smooth, we have: $\mathbb{E}\left\|\nabla\Phi(\hat{\boldsymbol{\alpha}}, \boldsymbol{v}^r)\right\|^2 \leq 2L\mathbb{E}[\Phi(\hat{\boldsymbol{\alpha}}, \boldsymbol{v}^r) - \Phi(\hat{\boldsymbol{\alpha}}, \boldsymbol{v}^*(\hat{\boldsymbol{\alpha}}))]$. Therefore, we have:

$$\mathbb{E}\left\|\boldsymbol{v}^{r+1} - \boldsymbol{v}^*(\hat{\boldsymbol{\alpha}})\right\|^2 \leq (1 - \frac{1}{2}\mu\eta NK)\mathbb{E}\left\|\boldsymbol{v}^r - \boldsymbol{v}^*(\hat{\boldsymbol{\alpha}})\right\|^2 - (\eta NK - 4\eta^2 N^2 K^2 L)\mathbb{E}[\Phi(\hat{\boldsymbol{\alpha}}, \boldsymbol{v}^r) - \Phi(\hat{\boldsymbol{\alpha}}, \boldsymbol{v}^*(\hat{\boldsymbol{\alpha}}))]$$

$$(13)$$

$$+ \left(\frac{1}{2\mu\eta NK} + 2\right)\mathbb{E}\|\boldsymbol{g}^r - \eta NK\nabla\Phi(\hat{\boldsymbol{\alpha}}, \boldsymbol{v}^r)\|^2 + \mathbb{E}\left\|\boldsymbol{\delta}^r\right\|^2. \tag{14}$$

Now, we examine the term $\|\boldsymbol{g}^r - \eta N K \nabla \Phi(\hat{\boldsymbol{\alpha}}, \boldsymbol{v}^r)\|^2$. First according to summation by part (Lemma 8) by letting $\boldsymbol{A}_j := \prod_{j'=N}^{j+1}(\mathbf{I} - \boldsymbol{Q}_{j'}\mathbf{H}_{j'})$ and $\mathbf{B}_j = \boldsymbol{Q}_j \nabla f_{\sigma(j)}(\boldsymbol{v}^r)$, we have:

$$
\boldsymbol{g}^r = \sum_{j=1}^{N} \prod_{j'=N}^{j+1} (\mathbf{I} - \boldsymbol{Q}_{j'}\mathbf{H}_{j'}) \boldsymbol{Q}_j \nabla f_{\sigma(j)}(\boldsymbol{v}^r)
$$

$$
= \sum_{j=1}^{N} \boldsymbol{A}_j \mathbf{B}_j = \sum_{j=1}^{N} \boldsymbol{Q}_j \nabla f_{\sigma(j)}(\boldsymbol{v}^r) - \sum_{n=1}^{N-1} \left( \prod_{j'=N}^{n+2} (\mathbf{I} - \boldsymbol{Q}_{j'}\mathbf{H}_{j'}) - \prod_{j'=N}^{n+1} (\mathbf{I} - \boldsymbol{Q}_{j'}\mathbf{H}_{j'}) \right) \sum_{j=1}^{n} \boldsymbol{Q}_j \nabla f_{\sigma(j)}(\boldsymbol{v}^r)
$$

$$
= \sum_{j=1}^{N} \boldsymbol{Q}_j \nabla f_{\sigma(j)}(\boldsymbol{v}^r) - \sum_{n=1}^{N-1} \left( \prod_{j'=N}^{n+2} (\mathbf{I} - \boldsymbol{Q}_{j'}\mathbf{H}_{j'}) \right) \boldsymbol{Q}_{n+1}\mathbf{H}_{n+1} \sum_{j=1}^{n} \boldsymbol{Q}_j \nabla f_{\sigma(j)}(\boldsymbol{v}^r).
$$

Hence we have:

$$
\|\boldsymbol{g}^r - \eta N K \nabla \Phi(\hat{\boldsymbol{\alpha}}, \boldsymbol{v}^r)\|^2
$$

$$
= \left\| \eta N K \sum_{j=1}^{N} \hat{\alpha}(\sigma(j)) \nabla f_{\sigma(j)}(\boldsymbol{v}^r) - \left( \sum_{j=1}^{N} \boldsymbol{Q}_j \nabla f_{\sigma(j)}(\boldsymbol{v}^r) - \sum_{n=1}^{N-1} \left( \prod_{j'=N}^{n+2} (\mathbf{I} - \boldsymbol{Q}_{j'}\mathbf{H}_{j'}) \right) \boldsymbol{Q}_{n+1}\mathbf{H}_{n+1} \sum_{j=1}^{n} \boldsymbol{Q}_j \nabla f_{\sigma(j)}(\boldsymbol{v}^r) \right) \right\|^2
$$

$$
\overset{(1)}{=} 2 \left\| \left( \eta N K \sum_{j=1}^{N} \hat{\alpha}(\sigma(j)) \nabla f_{\sigma(j)}(\boldsymbol{v}^r) - \sum_{j=1}^{N} \boldsymbol{Q}_j \nabla f_{\sigma(j)}(\boldsymbol{v}^r) \right) \right\|^2 + 2 \left\| \sum_{n=1}^{N-1} \left( \prod_{j'=N}^{n+2} (\mathbf{I} - \boldsymbol{Q}_{j'}\mathbf{H}_{j'}) \right) \boldsymbol{Q}_{n+1}\mathbf{H}_{n+1} \sum_{j=1}^{n} \boldsymbol{Q}_j \nabla f_{\sigma(j)}(\boldsymbol{v}^r) \right\|^2
$$

$$
\overset{(2)}{\leq} \left( 20\eta^2 N^2 K^2 \left( \frac{e}{4R-e} \right)^2 + 36 e^6 \eta^4 N^4 K^4 L^4 \right) \|\nabla \Phi(\hat{\boldsymbol{\alpha}}, \boldsymbol{v}^r)\|^2 + 256\eta^2 N^3 K^2 \left( \frac{e}{4R-e} \right)^2 G^2 \log(1/p)
$$

$$
+ 244 e^6 \eta^4 N^4 K^4 L^4 G^2 N \log(1/p)
$$

$$
\overset{(3)}{\leq} \left( 20\eta^2 N^2 K^2 \left( \frac{e}{4R-e} \right)^2 + 36 e^6 \eta^4 N^4 K^4 L^4 \right) 2L \left( \Phi(\hat{\boldsymbol{\alpha}}, \boldsymbol{v}^r) - \Phi(\hat{\boldsymbol{\alpha}}, \boldsymbol{v}^*(\hat{\boldsymbol{\alpha}})) \right)
$$

$$
+ \left( 244 e^6 \eta^4 N^4 K^4 L^4 + 256\eta^2 N^3 K^2 \left( \frac{e}{4R-e} \right)^2 \right) G^2 N \log(1/p),
$$

where in (1) we apply Jensen's inequality, in (2) we plug in Lemma 11 (a), and Lemma 12, and in (3) we use the $L$-smoothness of $\Phi$. Plugging above bound back in (19) yields:

$$
\mathbb{E} \left\| \boldsymbol{v}^{r+1} - \boldsymbol{v}^*(\hat{\boldsymbol{\alpha}}) \right\|^2
$$

$$
\leq (1 - \frac{1}{2}\mu\eta N K)\mathbb{E} \left\| \boldsymbol{v}^r - \boldsymbol{v}^*(\hat{\boldsymbol{\alpha}}) \right\|^2 + \eta^2 N^2 K^2 e^4 \delta^2
$$

$$
- \underbrace{\left( \eta N K - 4\eta^2 N^2 K^2 L - \left( \frac{1}{2\mu\eta N K} + 2 \right) \left( 20\eta^2 N^2 K^2 \left( \frac{e}{4R-e} \right)^2 - 36 e^6 \eta^4 N^4 K^4 L^4 \right) \right)}_{T_1} \mathbb{E}[\Phi(\hat{\boldsymbol{\alpha}}, \boldsymbol{v}^r) - \Phi(\hat{\boldsymbol{\alpha}}, \boldsymbol{v}^*(\hat{\boldsymbol{\alpha}}))]
$$

$$
+ \left( \frac{1}{2\mu\eta N K} + 2 \right) \left( 244 e^6 \eta^4 N^4 K^4 L^4 + 256\eta^2 N^3 K^2 \left( \frac{e}{4R-e} \right)^2 \right) G^2 N \log(1/p).
$$

Since we choose $\eta = \frac{4\log(\sqrt{NK}R)}{\mu N K R}$, and large enough epoch number:

$$
R \geq \max \left\{ \left( \frac{40}{\mu} + 1 \right) e, 16\log(\sqrt{NK}R), 64\kappa \log(\sqrt{NK}R) \right\},
$$

we know that $T_1 \leq 0$. We thus have:

$$
\mathbb{E} \left\| \boldsymbol{v}^{r+1} - \boldsymbol{v}^*(\hat{\boldsymbol{\alpha}}) \right\|^2
$$

$$
\leq (1 - \frac{1}{2}\mu\eta N K)\mathbb{E} \left\| \boldsymbol{v}^r - \boldsymbol{v}^*(\hat{\boldsymbol{\alpha}}) \right\|^2 + \eta^2 N^2 K^2 e^4 \delta^2
$$

$$
+ \left( \frac{1}{2\mu\eta N K} + 2 \right) \left( 244 e^6 \eta^4 N^4 K^4 L^4 + 256\eta^2 N^3 K^2 \left( \frac{e}{4R-e} \right)^2 \right) G^2 N \log(1/p)
$$

Unrolling the recursion from $r = R$ to 0:

$$\mathbb{E}\left\|\boldsymbol{v}^R - \boldsymbol{v}^*(\hat{\boldsymbol{\alpha}})\right\|^2$$
$$\leq (1 - \frac{1}{2}\mu\eta NK)^R \mathbb{E}\left\|\boldsymbol{v}^0 - \boldsymbol{v}^*(\hat{\boldsymbol{\alpha}})\right\|^2 + \frac{2}{\mu}\eta NKe^4\delta^2$$
$$+ \frac{1}{\mu}\left(\frac{1}{2\mu\eta NK} + 2\right)\left(488e^6\eta^3 N^3 K^3 L^4 + 512\eta N^2 K\left(\frac{e}{4R-e}\right)^2\right)G^2 N\log(1/p).$$

Plugging in our choice of $\eta$ will conclude the proof:

$$\mathbb{E}\left\|\boldsymbol{v}^R - \boldsymbol{v}^*(\hat{\boldsymbol{\alpha}})\right\|^2 \leq \tilde{O}\left(\frac{\mathbb{E}\left\|\boldsymbol{v}^0 - \boldsymbol{v}^*(\hat{\boldsymbol{\alpha}})\right\|^2}{NKR^2} + \frac{\delta^2}{\mu^2 R} + \left(\frac{L^4 + N}{\mu^4 R^2}\right)G^2 N\log(1/p)\right).$$

Finally, according to Lemma 2 we can complete the proof:

$$\Phi(\boldsymbol{\alpha}_i^*, \hat{\boldsymbol{v}}_i) - \Phi(\boldsymbol{\alpha}_i^*, \boldsymbol{v}_i^*) \leq 2L\left\|\boldsymbol{v}_i^R - \boldsymbol{v}^*(\hat{\boldsymbol{\alpha}}_i)\right\|^2 + \left(2\kappa_\Phi^2 L + \frac{4NG^2}{L}\right)\|\hat{\boldsymbol{\alpha}}_i - \boldsymbol{\alpha}^*\|^2$$
$$\leq \tilde{O}\left(\frac{LD^2}{NKR^2} + \frac{L\delta^2}{\mu^2 R} + \left(\frac{L^4 + N}{\mu^4 R^2}\right)LG^2 N\log(1/p)\right)$$
$$+ \kappa_\Phi^2 L\tilde{O}\left(\exp\left(-\frac{T_{\boldsymbol{\alpha}}}{\kappa_g}\right) + \kappa_g^2\bar{\zeta}_i(\boldsymbol{w}^*)L^2\left(\frac{D^2}{RK} + \frac{\kappa\zeta^2}{\mu^2 R^2} + \frac{\delta^2}{\mu^2 NRK}\right)\right),$$

where we plug in the convergence result from Theorem 1 at last step. □

## C Proof of Convergence of Single Loop Algorithm

In this section, we turn to presenting the proof of single loop PERM algorithm (Algorithm 2) where the learning of mixing parameters and personalized models are coupled. Compared to Algorithm A2, here during the optimization of model, the mixing parameters are also being updated. As a result, we need to decouple the two updates which makes the analysis more involved. We begin with some technical lemmas that support the proof of main result.

### C.1 Technical Lemmas

**Proposition 4** (Basic Properties of SGD on Smooth Strongly Convex Function). *Let $\boldsymbol{w}^t$ to be the $t$th iterate of minibatch SGD on smooth and strongly convex function $F$, with minibatch size $M$ and learning rate $\gamma$. Also assume the variance is bounded by $\delta$. Then the following statements hold true after $T$ iterations of SGD:*

$$\mathbb{E}\|\nabla F(\boldsymbol{w}^T)\|^2 \leq 2L(1 - \mu\gamma)^T(F(\boldsymbol{w}^0) - F(\boldsymbol{w}^*)) + \frac{2\gamma\kappa\delta^2}{M} \tag{15}$$

$$\mathbb{E}\|\boldsymbol{w}^{T+1} - \boldsymbol{w}^T\|^2 \leq 2\gamma^2 L(1 - \mu\gamma)^T(F(\boldsymbol{w}^0) - F(\boldsymbol{w}^*)) + \frac{2\gamma^3\kappa\delta^2}{M} + \frac{\gamma^2\delta^2}{M} \tag{16}$$

$$\mathbb{E}\|\boldsymbol{w}^T - \boldsymbol{w}^*\|^2 \leq \frac{2}{\mu}(1 - \mu\gamma)^T(F(\boldsymbol{w}^0) - F(\boldsymbol{w}^*)) + 2\gamma\frac{\delta^2}{\mu^2 M}. \tag{17}$$

**Lemma 14** (Bounded iterates difference of $\boldsymbol{\alpha}$). *Let $\{\boldsymbol{\alpha}_i^r\}$ be iterates generated by Algorithm 2, then under conditions of Theorem 3, the following statement holds:*

$$\|\boldsymbol{\alpha}_i^r - \boldsymbol{\alpha}_i^{r-1}\|^2 \leq 6\left(1 - \frac{1}{\kappa_g}\right)^{T_{\boldsymbol{\alpha}}} + O\left(\kappa_g^2 L^2\bar{\zeta}_i(\boldsymbol{w}^*)\right)\left(\gamma^2 L(1 - \mu\gamma)^r(F(\boldsymbol{w}^0) - F(\boldsymbol{w}^*)) + \frac{\gamma^3\kappa\delta^2}{M} + \frac{\gamma^2\delta^2}{M}\right)$$

*Proof.* Define

$$\boldsymbol{z}^r = \left[\|\nabla f_i(\boldsymbol{w}^r) - \nabla f_1(\boldsymbol{w}^r)\|^2, ..., \|\nabla f_i(\boldsymbol{w}^r) - \nabla f_N(\boldsymbol{w}^r)\|^2\right].$$

According to updating rule of $\boldsymbol{\alpha}$ in Algorithm 2 and Lemma 3 we have:

$$\|\boldsymbol{\alpha}_i^r - \boldsymbol{\alpha}_i^{r-1}\|^2 \leq 3\|\boldsymbol{\alpha}_i^r - \boldsymbol{\alpha}_{g_i}^*(\boldsymbol{w}^r)\|^2 + 3\left\|\boldsymbol{\alpha}_{g_i}^*(\boldsymbol{w}^{r-1}) - \boldsymbol{\alpha}_{g_i}^*(\boldsymbol{w}^r)\right\|^2 + 3\|\boldsymbol{\alpha}_{g_i}^*(\boldsymbol{w}^{r-1}) - \boldsymbol{\alpha}_i^{r-1}\|^2$$

$$\leq 6(1 - \mu_g \eta_{\boldsymbol{\alpha}})^{T_{\boldsymbol{\alpha}}} + 3\left\|\boldsymbol{\alpha}_{g_i}^*(\boldsymbol{w}^{r-1}) - \boldsymbol{\alpha}_{g_i}^*(\boldsymbol{w}^r)\right\|^2$$

$$\leq 6(1 - \mu_g \eta_{\boldsymbol{\alpha}})^{T_{\boldsymbol{\alpha}}} + 3\kappa_g^2 \left\|\boldsymbol{z}^{r-1} - \boldsymbol{z}^r\right\|^2$$

$$\leq 6(1 - \mu_g \eta_{\boldsymbol{\alpha}})^{T_{\boldsymbol{\alpha}}} + 3\kappa_g^2 \sum_{j=1}^N \left\|\nabla f_i(\boldsymbol{w}^r) - \nabla f_j(\boldsymbol{w}^r) + \nabla f_i(\boldsymbol{w}^{r-1}) - \nabla f_j(\boldsymbol{w}^{r-1})\right\|^2 4L^2 \left\|\boldsymbol{w}^r - \boldsymbol{w}^{r-1}\right\|^2$$

where the third inequality follows from (8). Since $\|\nabla f_i(\boldsymbol{w}^r) - \nabla f_j(\boldsymbol{w}^r)\| \leq \|\nabla f_i(\boldsymbol{w}^*) - \nabla f_j(\boldsymbol{w}^*)\| + 2L \|\boldsymbol{w}^r - \boldsymbol{w}^*\|$, we can conclude that

$$\|\boldsymbol{\alpha}_i^r - \boldsymbol{\alpha}_i^{r-1}\|^2 \leq 6(1 - \mu_g \eta_{\boldsymbol{\alpha}})^{T_{\boldsymbol{\alpha}}}$$

$$+ 12L^2 \kappa_g^2 \sum_{j=1}^N \left(8\|\nabla f_i(\boldsymbol{w}^*) - \nabla f_j(\boldsymbol{w}^*)\|^2 + 8L^2 \|\boldsymbol{w}^r - \boldsymbol{w}^*\|^2 + 8L^2 \|\boldsymbol{w}^{r-1} - \boldsymbol{w}^*\|^2\right) \|\boldsymbol{w}^r - \boldsymbol{w}^{r-1}\|^2$$

$$\leq 6\left(1 - \frac{1}{\kappa_g}\right)^{T_{\boldsymbol{\alpha}}} + O\left(\kappa_g^2 L^2 \bar{\zeta}_i(\boldsymbol{w}^*) \|\boldsymbol{w}^r - \boldsymbol{w}^{r-1}\|^2\right)$$

$$\leq 6\left(1 - \frac{1}{\kappa_g}\right)^{T_{\boldsymbol{\alpha}}} + O\left(\kappa_g^2 L^2 \bar{\zeta}_i(\boldsymbol{w}^*)\right)\left(\gamma^2 L (1 - \mu\gamma)^r (F(\boldsymbol{w}^0) - F(\boldsymbol{w}^*)) + \frac{\gamma^3 \kappa \delta^2}{M} + \frac{\gamma^2 \delta^2}{M}\right)$$

where at last step we plug in Proposition 4 (16). $\qquad\square$

**Lemma 15** (Convergence of $\boldsymbol{\alpha}$). *Let $\{\hat{\boldsymbol{\alpha}}_i\}_{i=1}^N$ be the mixing parameters generated by Algorithm 2. Then under the conditions of Theorem 3, the following statement holds:*

$$\|\hat{\boldsymbol{\alpha}}_i - \boldsymbol{\alpha}^*\|^2 \leq 2(1 - \frac{1}{\kappa_g})^{T_{\boldsymbol{\alpha}}} + O\left(\kappa_g^2 \bar{\zeta}_i(\boldsymbol{w}^*) L^2 \frac{2}{\mu}(1 - \mu\gamma)^T + 2\gamma \frac{\delta^2}{\mu^2 M}\right), i \in [N]$$

*Proof.* We notice the following decomposition:

$$\|\hat{\boldsymbol{\alpha}}_i - \boldsymbol{\alpha}^*\|^2 = \|\boldsymbol{\alpha}_i^R - \boldsymbol{\alpha}_g^*(\boldsymbol{w}^*)\|^2$$

$$\leq 2\|\boldsymbol{\alpha}_i^R - \boldsymbol{\alpha}_g^*(\boldsymbol{w}^R)\|^2 + 2\|\boldsymbol{\alpha}_{g_i}^*(\boldsymbol{w}^R) - \boldsymbol{\alpha}_{g_i}^*(\boldsymbol{w}^*)\|^2$$

$$\leq 2(1 - \frac{1}{\kappa_g})^{T_{\boldsymbol{\alpha}}} + O\left(\kappa_g^2\left(\bar{\zeta}_i(\boldsymbol{w}^*) + NL^2 \|\boldsymbol{w}^R - \boldsymbol{w}^*\|^2\right) 4L \|\boldsymbol{w}^R - \boldsymbol{w}^*\|^2\right)$$

$$\leq 2(1 - \frac{1}{\kappa_g})^{T_{\boldsymbol{\alpha}}} + O\left(\kappa_g^2 \bar{\zeta}_i(\boldsymbol{w}^*) L^2 \frac{2}{\mu}(1 - \mu\gamma)^T + 2\gamma \frac{\delta^2}{\mu^2 M}\right),$$

where in the second inequality we apply Lemma 3, and in the third inequality we use Proposition 4 (17). $\qquad\square$

## C.2 Proof of Theorem 3

*Proof.* According to Lemma 2, we have:

$$\Phi(\boldsymbol{\alpha}_i^*, \hat{\boldsymbol{v}}_i) - \Phi(\boldsymbol{\alpha}_i^*, \boldsymbol{v}_i^*) \leq 2L \|\boldsymbol{v}_i^R - \boldsymbol{v}^*(\hat{\boldsymbol{\alpha}}_i)\|^2 + \left(2\kappa_\Phi^2 L + \frac{4NG^2}{L}\right)\|\hat{\boldsymbol{\alpha}}_i - \boldsymbol{\alpha}_i^*\|^2.$$

We first examine the convergence of $\|\boldsymbol{v}_i^R - \boldsymbol{v}^*(\hat{\boldsymbol{\alpha}}_i)\|^2$. Applying Cauchy-Schwartz inequality yields:

$$\|\boldsymbol{v}^{r+1} - \boldsymbol{v}^*(\boldsymbol{\alpha}^{r+1})\|^2 \leq \left(1 + \frac{1}{4a-2}\right)\|\boldsymbol{v}^{r+1} - \boldsymbol{v}^*(\boldsymbol{\alpha}^r)\|^2 + (1 + 4a - 2)\|\boldsymbol{v}^*(\boldsymbol{\alpha}^{r+1}) - \boldsymbol{v}^*(\boldsymbol{\alpha}^r)\|^2$$

$$\leq \left(1 + \frac{1}{4a-2}\right)\|\boldsymbol{v}^{r+1} - \boldsymbol{v}^*(\boldsymbol{\alpha}^r)\|^2 + (1 + 4a - 2)\kappa_\Phi^2 \|\boldsymbol{\alpha}^{r+1} - \boldsymbol{\alpha}^r\|^2 \tag{18}$$

where $a = \frac{1}{\mu \eta NK}$, and last step is due to that $\boldsymbol{v}^*(\boldsymbol{\alpha})$ is $\kappa_\Phi := \frac{\sqrt{N}G}{\mu}$ Lipschitz, as proven in Lemma 2
. Similar to the proof of Theorem 2, we first define

$$\boldsymbol{g}^r := \sum_{j=1}^{N} \prod_{j'=N-1}^{j+1} (\mathbf{I} - \boldsymbol{Q}_{j'}\mathbf{H}_{j'})\boldsymbol{Q}_j \nabla f_{\sigma(j)}(\boldsymbol{v}^r),$$

$$\boldsymbol{\delta}^r := \sum_{j=1}^{N} \prod_{j'=N-1}^{j+1} (\mathbf{I} - \boldsymbol{Q}_{j'}\mathbf{H}_{j'})\boldsymbol{\delta}_j.$$

Then we recall the updating rule of $\boldsymbol{v}$:

$$\boldsymbol{v}^{r+1} = \mathcal{P}_\mathcal{W}\left(\boldsymbol{v}^r - \boldsymbol{g}^r - \boldsymbol{\delta}^r\right).$$

Hence we have:

$$\begin{aligned}
\mathbb{E}\left\|\boldsymbol{v}^{r+1} - \boldsymbol{v}^*(\boldsymbol{\alpha}^r)\right\|^2 &= \mathbb{E}\left\|\mathcal{P}_\mathcal{W}\left(\boldsymbol{v}^r - \boldsymbol{g}^r - \boldsymbol{\delta}^r - \boldsymbol{v}^*(\boldsymbol{\alpha}^r)\right)\right\|^2 \\
&\leq \mathbb{E}\left\|\boldsymbol{v}^r - \boldsymbol{g}^r - \boldsymbol{\delta}^r - \boldsymbol{v}^*(\boldsymbol{\alpha}^r)\right\|^2 \\
&\leq \mathbb{E}\left\|\boldsymbol{v}^r - \boldsymbol{v}^*(\boldsymbol{\alpha}^r)\right\|^2 - 2\mathbb{E}\langle \boldsymbol{g}^r, \boldsymbol{v}^r - \boldsymbol{v}^*(\boldsymbol{\alpha}^r)\rangle + \mathbb{E}\left\|\boldsymbol{g}^r\right\|^2 + \mathbb{E}\left\|\boldsymbol{\delta}^r\right\|^2 \\
&\leq \mathbb{E}\left\|\boldsymbol{v}^r - \boldsymbol{v}^*(\boldsymbol{\alpha}^r)\right\|^2 - 2\mathbb{E}\langle \eta NK\nabla\Phi(\boldsymbol{\alpha}^r, \boldsymbol{v}^r), \boldsymbol{v}^r - \boldsymbol{v}^*(\boldsymbol{\alpha}^r)\rangle \\
&\quad - 2\mathbb{E}\langle \boldsymbol{g}^r - \eta NK\nabla\Phi(\boldsymbol{\alpha}^r, \boldsymbol{v}^r), \boldsymbol{v}^r - \boldsymbol{v}^*(\boldsymbol{\alpha}^r)\rangle + \mathbb{E}\left\|\boldsymbol{g}^r\right\|^2 + \mathbb{E}\left\|\boldsymbol{\delta}^r\right\|^2.
\end{aligned}$$

Now, applying strongly convexity of $\Phi(\boldsymbol{\alpha}^r, \cdot)$ and Cauchy-Schwartz inequality yields:

$$\begin{aligned}
\mathbb{E}\left\|\boldsymbol{v}^{r+1} - \boldsymbol{v}^*(\boldsymbol{\alpha}^r)\right\|^2 &\leq (1 - \mu\eta NK)\mathbb{E}\left\|\boldsymbol{v}^r - \boldsymbol{v}^*(\boldsymbol{\alpha}^r)\right\|^2 - \eta NK\mathbb{E}[\Phi(\boldsymbol{\alpha}^r, \boldsymbol{v}^r) - \Phi(\boldsymbol{\alpha}^r, \boldsymbol{v}^*(\hat{\boldsymbol{\alpha}}))] \\
&\quad + \frac{1}{2}\left(\frac{1}{\mu\eta NK}\mathbb{E}\|\boldsymbol{g}^r - \eta NK\nabla\Phi(\hat{\boldsymbol{\alpha}}, \boldsymbol{v}^r)\|^2 + \mu\eta NK\mathbb{E}\|\boldsymbol{v}^r - \boldsymbol{v}^*(\hat{\boldsymbol{\alpha}})\|^2\right) \\
&\quad + \mathbb{E}\left\|\boldsymbol{g}^r\right\|^2 + \mathbb{E}\left\|\boldsymbol{\delta}^r\right\|^2 \\
&\leq \left(1 - \frac{1}{2}\mu\eta NK\right)\mathbb{E}\left\|\boldsymbol{v}^r - \boldsymbol{v}^*(\boldsymbol{\alpha}^r)\right\|^2 - \eta NK\mathbb{E}[\Phi(\boldsymbol{\alpha}^r, \boldsymbol{v}^r) - \Phi(\boldsymbol{\alpha}^r, \boldsymbol{v}^*(\hat{\boldsymbol{\alpha}}))] \\
&\quad + \frac{1}{2\mu\eta NK}\mathbb{E}\|\boldsymbol{g}^r - \eta NK\nabla\Phi(\boldsymbol{\alpha}^r, \boldsymbol{v}^r)\|^2 \\
&\quad + 2\mathbb{E}\left\|\boldsymbol{g}^r - \eta NK\nabla\Phi(\boldsymbol{\alpha}^r, \boldsymbol{v}^r)\right\|^2 + 2\mathbb{E}\left\|\eta NK\nabla\Phi(\boldsymbol{\alpha}^r, \boldsymbol{v}^r)\right\|^2 + \mathbb{E}\left\|\boldsymbol{\delta}^r\right\|^2.
\end{aligned}$$

where in the first inequality we applied Cauchy-Schwartz inequality and strongly convexity. Since $\Phi(\boldsymbol{\alpha}^r, \cdot)$ is $L$ smooth, we have: $\mathbb{E}\left\|\nabla\Phi(\boldsymbol{\alpha}^r, \boldsymbol{v}^r)\right\|^2 \leq 2L\mathbb{E}[\Phi(\boldsymbol{\alpha}^r, \boldsymbol{v}^r) - \Phi(\boldsymbol{\alpha}^r, \boldsymbol{v}^*(\boldsymbol{\alpha}^r))]$. Therefore, it follows that:

$$\begin{aligned}
\mathbb{E}\left\|\boldsymbol{v}^{r+1} - \boldsymbol{v}^*(\hat{\boldsymbol{\alpha}})\right\|^2 &\leq \left(1 - \frac{1}{2}\mu\eta NK\right)\mathbb{E}\left\|\boldsymbol{v}^r - \boldsymbol{v}^*(\boldsymbol{\alpha}^r)\right\|^2 - (\eta NK - 4\eta^2 N^2 K^2 L)\mathbb{E}[\Phi(\boldsymbol{\alpha}^r, \boldsymbol{v}^r) - \Phi(\boldsymbol{\alpha}^r, \boldsymbol{v}^*(\boldsymbol{\alpha}^r))] \\
&\quad + \left(\frac{1}{2\mu\eta NK} + 2\right)\mathbb{E}\|\boldsymbol{g}^r - \eta NK\nabla\Phi(\boldsymbol{\alpha}^r, \boldsymbol{v}^r)\|^2 + \mathbb{E}\left\|\boldsymbol{\delta}^r\right\|^2 \qquad (19)
\end{aligned}$$

Now, we examine the term $\|\boldsymbol{g}^r - \eta NK\nabla\Phi(\boldsymbol{\alpha}^r, \boldsymbol{v}^r)\|^2$ in the right hand side of abovee inequality. First according to summation by part (Lemma 8): we let $\boldsymbol{A}_j := \prod_{j'=N-1}^{j+1}(\mathbf{I} - \boldsymbol{Q}_{j'}\mathbf{H}_{j'})$ and $\mathbf{B}_j = \boldsymbol{Q}_j \nabla f_{\sigma(j)}(\boldsymbol{v}^r)$, then we have:

$$\begin{aligned}
\boldsymbol{g}^r &= \sum_{j=1}^{N} \prod_{j'=N}^{j+1} (\mathbf{I} - \boldsymbol{Q}_{j'}\mathbf{H}_{j'})\boldsymbol{Q}_j \nabla f_{\sigma(j)}(\boldsymbol{v}^r) \\
&= \sum_{j=1}^{N} \boldsymbol{A}_j \mathbf{B}_j = \sum_{j=1}^{N} \boldsymbol{Q}_j \nabla f_{\sigma(j)}(\boldsymbol{v}^r) - \sum_{n=1}^{N-1}\left(\prod_{j'=N}^{n+2}(\mathbf{I} - \boldsymbol{Q}_{j'}\mathbf{H}_{j'}) - \prod_{j'=N}^{n+1}(\mathbf{I} - \boldsymbol{Q}_{j'}\mathbf{H}_{j'})\right)\sum_{j=1}^{n} \boldsymbol{Q}_j \nabla f_{\sigma(j)}(\boldsymbol{v}^r) \\
&= \sum_{j=1}^{N} \boldsymbol{Q}_j \nabla f_{\sigma(j)}(\boldsymbol{v}^r) - \sum_{n=1}^{N-1}\left(\prod_{j'=N}^{n+2}(\mathbf{I} - \boldsymbol{Q}_{j'}\mathbf{H}_{j'})\right)\boldsymbol{Q}_{n+1}\mathbf{H}_{n+1}\sum_{j=1}^{n} \boldsymbol{Q}_j \nabla f_{\sigma(j)}(\boldsymbol{v}^r).
\end{aligned}$$

Hence we have:

$$\|\boldsymbol{g}^r - \eta NK\nabla\Phi(\hat{\boldsymbol{\alpha}}, \boldsymbol{v}^r)\|^2$$

$$= \left\|\eta NK\nabla\Phi(\hat{\boldsymbol{\alpha}}, \boldsymbol{v}^r) - \sum_{j=1}^{N}\prod_{j'=N-1}^{j+1}(\mathbf{I} - \boldsymbol{Q}_{j'}\mathbf{H}_{j'})\boldsymbol{Q}_j\nabla f_{\sigma(j)}(\boldsymbol{v}^r)\right\|^2$$

$$= \left\|\eta NK\sum_{j=1}^{N}\hat{\alpha}(\sigma(j))\nabla f_{\sigma(j)}(\boldsymbol{v}^r) - \left(\sum_{j=1}^{N}\boldsymbol{Q}_j\nabla f_{\sigma(j)}(\boldsymbol{v}^r) - \sum_{n=1}^{N-1}\left(\prod_{j'=N}^{n+2}(\mathbf{I} - \boldsymbol{Q}_{j'}\mathbf{H}_{j'})\right)\boldsymbol{Q}_{n+1}\mathbf{H}_{n+1}\sum_{j=0}^{n}\boldsymbol{Q}_j\nabla f_{\sigma(j)}(\boldsymbol{v}^r)\right)\right\|^2$$

$$\overset{(1)}{\leq} 2\left\|\left(\eta NK\sum_{j=1}^{N}\hat{\alpha}(\sigma(j))\nabla f_{\sigma(j)}(\boldsymbol{v}^r) - \sum_{j=1}^{N}\boldsymbol{Q}_j\nabla f_{\sigma(j)}(\boldsymbol{v}^r)\right)\right\|^2$$

$$+ 2\left\|\sum_{n=1}^{N-1}\left(\prod_{j'=N}^{n+2}(\mathbf{I} - \boldsymbol{Q}_{j'}\mathbf{H}_{j'})\right)\boldsymbol{Q}_{n+1}\mathbf{H}_{n+1}\sum_{j=1}^{n}\boldsymbol{Q}_j\nabla f_{\sigma(j)}(\boldsymbol{v}^r)\right\|^2$$

$$\overset{(2)}{\leq} \left(20\eta^2 N^2 K^2\left(\frac{e}{4R-e}\right)^2 + 36e^6\eta^4 N^4 K^4 L^4\right)\|\nabla\Phi(\hat{\boldsymbol{\alpha}}, \boldsymbol{v}^r)\|^2 + 256\eta^2 N^3 K^2\left(\frac{e}{4R-e}\right)^2 G^2\log(1/p)$$

$$+ 244e^6\eta^4 N^4 K^4 L^4 G^2 N\log(1/p)$$

$$\overset{(3)}{\leq} \left(20\eta^2 N^2 K^2\left(\frac{e}{4R-e}\right)^2 + 36e^6\eta^4 N^4 K^4 L^4\right)2L\left(\Phi(\hat{\boldsymbol{\alpha}}, \boldsymbol{v}^r) - \Phi(\hat{\boldsymbol{\alpha}}, \boldsymbol{v}^*(\hat{\boldsymbol{\alpha}}))\right)$$

$$+ \left(244e^6\eta^4 N^4 K^4 L^4 + 256\eta^2 N^3 K^2\left(\frac{e}{4R-e}\right)^2\right)G^2 N\log(1/p)$$

where in (1) we apply Jensen's inequality, in (2) we plug in Lemma 11 (a), and Lemma 12, and in (3) we use the $L$-smoothness of $\Phi$. Plugging above bound back in (19) yields:

$$\mathbb{E}\left\|\boldsymbol{v}^{r+1} - \boldsymbol{v}^*(\boldsymbol{\alpha}^r)\right\|^2$$

$$\leq (1 - \frac{1}{2}\mu\eta NK)\mathbb{E}\|\boldsymbol{v}^r - \boldsymbol{v}^*(\boldsymbol{\alpha}^r)\|^2 + \eta^2 N^2 K^2 e^4\delta^2$$

$$- \left(\eta NK - 4\eta^2 N^2 K^2 L - \left(\frac{1}{2\mu\eta NK} + 2\right)\left(20\eta^2 N^2 K^2\left(\frac{e}{4R-e}\right)^2 - 36e^6\eta^4 N^4 K^4 L^4\right)\right)$$

$$\times \mathbb{E}[\Phi(\boldsymbol{\alpha}^r, \boldsymbol{v}^r) - \Phi(\boldsymbol{\alpha}^r, \boldsymbol{v}^*(\boldsymbol{\alpha}^r))]$$

$$+ \left(\frac{1}{2\mu\eta NK} + 2\right)\left(244e^6\eta^4 N^4 K^4 L^4 + 256\eta^2 N^3 K^2\left(\frac{e}{4R-e}\right)^2\right)G^2 N\log(1/p).$$

Since we choose $\eta = \frac{4\log(\sqrt{NK}R)}{\mu NKR}$, and

$$R \geq \max\left\{\frac{3}{8}e, \sqrt[3]{\frac{64\kappa^2\log(\sqrt{NK}R)e^6}{9\mu}}\right\},$$

hence we have:

$$\mathbb{E}\left\|\boldsymbol{v}^{r+1}-\boldsymbol{v}^*(\boldsymbol{\alpha}^r)\right\|^2$$

$$\leq (1-\frac{1}{2}\mu\eta NK)\mathbb{E}\left\|\boldsymbol{v}^r-\boldsymbol{v}^*(\boldsymbol{\alpha}^r)\right\|^2+\eta^2 N^2 K^2 e^4\delta^2-\frac{1}{2}\eta NK\underbrace{\mathbb{E}[\Phi(\hat{\boldsymbol{\alpha}},\boldsymbol{v}^r)-\Phi(\hat{\boldsymbol{\alpha}},\boldsymbol{v}^*(\hat{\boldsymbol{\alpha}}))]}_{\geq 0}$$

$$+\left(\frac{1}{2\mu\eta NK}+2\right)\left(244e^6\eta^4 N^4 K^4 L^4+256\eta^2 N^3 K^2\left(\frac{e}{4R-e}\right)^2\right)G^2 N\log(1/p)$$

$$\leq \left(1-\frac{1}{2}\mu\eta NK\right)\mathbb{E}\left\|\boldsymbol{v}^r-\boldsymbol{v}^*(\hat{\boldsymbol{\alpha}})\right\|^2+\eta^2 N^2 K^2 e^4\delta^2$$

$$+\left(\frac{1}{2\mu\eta NK}+2\right)\left(244e^6\eta^4 N^4 K^4 L^4+256\eta^2 N^3 K^2\left(\frac{e}{4R-e}\right)^2\right)G^2 N\log(1/p).$$

Putting above inequality back to (18) yields:

$$\|\boldsymbol{v}^{r+1}-\boldsymbol{v}^*(\boldsymbol{\alpha}^{r+1})\|^2\leq\left(1-\frac{1}{4a}\right)\|\boldsymbol{v}^r-\boldsymbol{v}^*(\boldsymbol{\alpha}^r)\|^2+2\eta^2 N^2 K^2 e^4\delta^2+(1+4a-2)\kappa_\Phi^2\|\boldsymbol{\alpha}^{r+1}-\boldsymbol{\alpha}^r\|^2$$

$$+2\left(\frac{1}{2\mu\eta NK}+2\right)\left(244e^6\eta^4 N^4 K^4 L^4+256\eta^2 N^3 K^2\left(\frac{e}{4R-e}\right)^2\right)G^2 N\log(1/p)$$

$$\leq\left(1-\frac{1}{4a}\right)\|\boldsymbol{v}^r-\boldsymbol{v}^*(\boldsymbol{\alpha}^r)\|^2+2\eta^2 N^2 K^2 e^4\delta^2$$

$$+2\left(\frac{1}{2\mu\eta NK}+2\right)\left(244e^6\eta^4 N^4 K^4 L^4+256\eta^2 N^3 K^2\left(\frac{e}{4R-e}\right)^2\right)G^2 N\log(1/p)$$

$$+O\left(\frac{\kappa_\Phi^2}{\mu\eta NK}\left(\left(1-\frac{1}{\kappa_g}\right)^{T_\alpha}+\kappa_g^2 L^2\bar{\zeta}_i(\boldsymbol{w}^*)\left(\gamma^2 L(1-\mu\gamma)^r DG+\frac{\gamma^3\kappa\delta^2}{M}+\frac{\gamma^2\delta^2}{M}\right)\right)\right)$$

where at second inequality we plug in Lemma 14. Unrolling the recursion from $r=R$ to 0, and plugging in $\eta=\frac{4\log(NKR^3)}{\mu NKR}$ yields:

$$\|\boldsymbol{v}^R-\boldsymbol{v}^*(\boldsymbol{\alpha}^R)\|^2$$

$$\leq\left(1-\frac{1}{4}\mu\eta NK\right)^R\|\boldsymbol{v}^0-\boldsymbol{v}^*(\boldsymbol{\alpha}^0)\|^2+\frac{1}{\mu}\eta NK e^4\delta^2$$

$$+8\frac{1}{\mu}\left(\frac{1}{2\mu\eta NK}+2\right)\left(244e^6\eta^3 N^3 K^3 L^4+256\eta N^2 K\left(\frac{e}{4R-e}\right)^2\right)G^2 N\log(1/p)$$

$$+O\left(\frac{\kappa_\Phi^2}{\mu\eta NK}\sum_{r=0}^R\left(1-\frac{1}{4a}\right)^{R-r}\left(\left(1-\frac{1}{\kappa_g}\right)^{T_\alpha}+\kappa_g^2 L^2\bar{\zeta}_i(\boldsymbol{w}^*)\left(\gamma^2 L(1-\mu\gamma)^r DG+\frac{\gamma^3\kappa\delta^2}{M}+\frac{\gamma^2\delta^2}{M}\right)\right)\right)$$

$$\leq O\left(\frac{\|\boldsymbol{v}^0-\boldsymbol{v}^*(\boldsymbol{\alpha}^0)\|^2}{NKR^3}\right)+\tilde{O}\left(\left(\frac{\kappa^4}{R^2}+\frac{N}{\mu^2 R^2}\right)G^2 N\log(1/p)+\frac{\delta^2}{\mu R}\right)$$

$$+\tilde{O}\left(R\kappa_\Phi^2\sum_{r=0}^R\left(1-\frac{\log(NKR^3)}{R}\right)^{R-r}\left(\left(1-\frac{1}{\kappa_g}\right)^{T_\alpha}+\kappa_g^2 L^2\bar{\zeta}_i(\boldsymbol{w}^*)\left(\gamma^2 L(1-\mu\gamma)^r DG+\frac{\gamma^3\kappa\delta^2}{M}+\frac{\gamma^2\delta^2}{M}\right)\right)\right)$$

Plugging in $\gamma = \frac{\log(NKR^3)}{\mu R}$ yields:

$$\|v^R - v^*(\alpha^R)\|^2 \leq O\left(\frac{\|v^0 - v^*(\alpha^0)\|^2}{NKR^3}\right) + \tilde{O}\left(\left(\frac{\kappa^4}{R^2} + \frac{N}{\mu^2 R^2}\right)G^2 N\log(1/p) + \frac{\delta^2}{\mu R}\right)$$

$$+ \tilde{O}\left(\kappa_\Phi^2 R\left(\gamma^2 L^2 \sum_{r=0}^{R}\left(1 - \frac{\log(NKR^3)}{R}\right)^R \kappa_g^2 L^2 \bar{\zeta}_i(w^*)DG\right.\right.$$

$$\left.\left. + \sum_{r=0}^{R}\left(1 - \frac{\log(NKR^3)}{R}\right)^{R-r}\left(\left(1 - \frac{1}{\kappa_g}\right)^{T_\alpha} + \frac{\kappa_g^2 L^2 \bar{\zeta}_i(w^*)\gamma^2\delta^2}{M}\right)\right)\right)$$

$$\leq O\left(\frac{D^2}{NKR^3}\right) + \tilde{O}\left(\left(\frac{\kappa^4}{R^2} + \frac{N}{\mu^2 R^2}\right)G^2 N\log(1/p) + \frac{\delta^2}{\mu R}\right)$$

$$+ \tilde{O}\left(\frac{\kappa_\Phi^2 \kappa^2 \kappa_g^2 L^2 \bar{\zeta}_i(w^*)DG}{R} + \kappa_\Phi^2 R^2\left(\left(1 - \frac{1}{\kappa_g}\right)^{T_\alpha} + \frac{\kappa_g^2 \kappa^2 \bar{\zeta}_i(w^*)\delta^2}{\mu^2 MR^2}\right)\right)$$

Since $\hat{v}_i = v^R$ and $\hat{\alpha}_i = \alpha^R$, we have the convergence of $\|\hat{v}_i - v^*(\hat{\alpha}_i)\|^2$. Plugging this convergence rate together with the convergence of $\|\hat{\alpha}_i - \alpha^*\|^2$ from Lemma 15:

$$\|\hat{\alpha}_i - \alpha^*\|^2 \leq O\left(2(1 - \frac{1}{\kappa_g})^{T_\alpha} + O\left(\kappa_g^2 \bar{\zeta}_i(w^*)L^2\frac{2}{\mu}(1 - \mu\gamma)^R + 2\gamma\frac{\delta^2}{\mu^2 M}\right)\right)$$

$$\leq \tilde{O}\left(\left(1 - \frac{1}{\kappa_g}\right)^{T_\alpha} + O\left(\kappa_g^2 \bar{\zeta}_i(w^*)L^2\frac{2}{\mu}\frac{1}{R} + \frac{\delta^2}{\mu^3 RM}\right)\right)$$

together with applying Lemma 2 leads to:

$$\Phi(\alpha_i^*, \hat{v}_i) - \Phi(\alpha_i^*, v_i^*) \leq 2L\|\hat{v}_i - v^*(\hat{\alpha}_i)\|^2 + \left(2\kappa_\Phi^2 L + \frac{4NG^2}{L}\right)\|\hat{\alpha}_i - \alpha_i^*\|^2$$

$$\leq O\left(\frac{LD^2}{NKR^3}\right) + \tilde{O}\left(\left(\frac{\kappa^4 L}{R^2} + \frac{NL}{\mu^2 R^2}\right)G^2 N\log(1/p) + \frac{L\delta^2}{\mu R}\right)$$

$$+ \tilde{O}\left(\frac{\kappa_\Phi^2 \kappa^2 \kappa_g^2 L^3 \bar{\zeta}_i(w^*)DG}{R} + \kappa_\Phi^2 LR^2\left(1 - \frac{1}{\kappa_g}\right)^{T_\alpha} + \frac{L\kappa_\Phi^2 \kappa_g^2 \kappa^2 \bar{\zeta}_i(w^*)\delta^2}{\mu^2 M}\right)$$

$$+ \left(2\kappa_\Phi^2 L + \frac{4NG^2}{L}\right)\tilde{O}\left(\left(1 - \frac{1}{\kappa_g}\right)^{T_\alpha} + \frac{\kappa_g^2 \bar{\zeta}_i(w^*)\kappa L}{R} + \frac{\delta^2}{\mu^3 RM}\right)$$

$$\leq O\left(\frac{LD^2}{NKR^3}\right) + \tilde{O}\left(\left(\frac{\kappa^4 L}{R^2} + \frac{NL}{\mu^2 R^2}\right)G^2 N\log(1/p) + \frac{L\delta^2}{\mu R}\right)$$

$$+ \tilde{O}\left(\frac{\kappa_\Phi^2 \kappa^2 \kappa_g^2 L^3 \bar{\zeta}_i(w^*)DG}{R} + \kappa_\Phi^2 LR^2\left(1 - \frac{1}{\kappa_g}\right)^{T_\alpha} + \frac{L^2 \kappa^2 \kappa_g^2 \kappa_\Phi^2 \bar{\zeta}_i(w^*)\delta^2}{\mu^2 M}\right).$$

thus completing the proof. $\qquad\square$