# OpenReview forum: "Distributed Personalized Empirical Risk Minimization"
_NeurIPS.cc/2023/Conference — NeurIPS 2023 poster_

### Official Review · Reviewer_XAoP · 2023-07-06

**Soundness:** 2 fair
**Presentation:** 3 good
**Contribution:** 2 fair
**Rating:** 5
**Confidence:** 3

**Summary:**

This paper proposes a new paradigm for learning from multiple heterogeneous data sources to achieve optimal statistical accuracy across all data distributions without imposing stringent constraints on computational resources shared by participating devices. The proposed Personalized Empirical Risk Minimization (PERM) schema provides an efficient solution to enable each client to learn a personalized model by learning who to learn with via personalizing the aggregation of data sources through an efficient empirical statistical discrepancy estimation module. The paper rigorously analyzes the convergence of the proposed algorithm and conducts experiments that corroborate the effectiveness of the proposed paradigm.

**Strengths:**

The strengths of this paper include:
1: The proposed PERM schema is a novel and practical methodology that addresses the data heterogeneity issue in distributed learning.

2: The paper provides insightful theoretical analysis of the proposed algorithm and rigorously proves its convergence.

3: The paper is well written and organized.

**Weaknesses:**

1: Inadequate evaluation and lack of verification on real data: Although the paper includes experiments showcasing the effectiveness of the proposed paradigm, it would greatly benefit from more comprehensive evaluation, particularly using real-world data.

2: The practical usefulness of Theorem 2 is limited due to its dependence on parameters related to the oracle models, which are typically unknown in real-world scenarios.

**Questions:**

1: Could you please provide additional details regarding the implementation of the proposed algorithm, including the specific hyperparameters utilized in the experiments?

2: How does the performance of the proposed paradigm compare to existing approaches in scenarios with highly heterogeneous data distributions?

3: What is the scalability of the proposed paradigm with respect to the number of participating devices? Are there any potential computational or communication bottlenecks that could hinder its scalability?

4: Why were real data experiments not conducted in this study?

**Limitations:**

See weaknesses.

---

> ### Author Rebuttal · Authors · 2023-08-10
>
> Thank you for your constructive comments! We will try to address your concerns as follows.
>
> **Inadequate evaluation and lack of verification on real data**
> We discuss this issue in Global comment. We have already reported results on real data sets in Appendix C of main paper and reported new results  in attached PDF as part of responses. We apologize for the confusion.
>
> **Details of implementation**
> For the synthetic dataset, we train using logistic regression with an $\ell_2$ regularization of $5\times10^{-4}$. The learning rate follows a convex decay schedule, initialized at $0.01$ and reduced by $10\%$ each iteration. We use a batch size of $128$ and perform $10$ local gradient descent steps on each communication round before model aggregation. The synthetic dataset itself is generated as detailed in the paper. For the EMNIST and MNIST datasets, we follow the same hyperparameters, except, we use the initial learning rate of $0.1$, and $20$ local steps in each communication round. Thank you for the question and we will gladly clarify this in the subsequent version.
>
>
> **Comparison with other approach on highly heterogeneous dataset**
> Theoretically, as we discussed in Section 2, when clients' data distribution are extremely heterogeneous, e.g., first half clients and second half clients have the opposite labels for the same input, then any existing works that relies on learning a global model then personalizing will fail as the global model may transfer *negative* knowledge from other highly irrelevant data sources. However, in PERM, this can be avoided by preperly estimating the mixing parameters (almost zero contribution from highly irrelevant data sources thus entailing better accuracy). This is also verified by our experiments on synthetic dataset, where meta-learning based algorithms and fine-tuned FedAvg model fail to achieve good accuracy. In the attached PDF (Figure 2) we included result on an extremely non-IID setting where the best model for each client is just the model trained solely based on its local data without collaboration, which further demonstrates the effectiveness of PERM compared to other methods that rely on learning a single global model to transfer knowledge among data sources (which might harm the performance due to negative effect of unrelated data sources which are eliminated in PERM by estimating the relatedness of data sources).
>
> **What is the scalability of the proposed paradigm with respect to the number of participating devices?**
>
> Thanks for the question. Please refer to Global comment for a detailed discussion on scalablity of PERM.  To summarize, the compute burden on clients and server is roughly same as existing methods thanks to shuffling (except extra overhead due to estimating mixing parameters which is same as running FedAvg in two-stage approach and an extra communication in interleaved approach). The only hurdle would be the required memory *at server* to maintain mixing parameters at server which scales at $O(N^2)$ which can be resolved by clustering devices or using approximate methods such as low rank approximation  which we leave as a future work.  To evaluate this empirically, we reported the **wall-clock time** of PERM and other methods in Figure.1 of attached PDF, demonstrating that the run-time is slightly worse due to overhead of estimating mixing parameters.
>
>
> **The practical usefulness of Theorem 2**
> The purpose of Theorem 2 is to show that PERM can provably converges  but in practice, we totally agree with your comment that some of the constants are usually unknown  which holds for most of theoretical analysis of SGD based methods, unless we utilize parameter-free or adaptive methods which is not the focus of work. We note that in implementing the PERM in conducting our experiments  we do not need the parameters of the function/oracle as discussed in your question about details of implementation. Our experiments validate the convergence of PERM as well.

---

> > ### Comment · Reviewer_XAoP · 2023-08-13
> >
> > Thank you very much for the response.

---

### Official Review · Reviewer_x8t7 · 2023-07-07

**Soundness:** 2 fair
**Presentation:** 3 good
**Contribution:** 2 fair
**Rating:** 5
**Confidence:** 1

**Summary:**

This paper studies a distributed training strategy to obtained a personalized model that shares a bases for all clients while an adaptor for each clients.

**Strengths:**

The overall derivation of the algorithm is theory-driven and reasonable.

Good experiment result in the simulation dataset.

**Weaknesses:**

This paper limits the mixing ops by a weighted summation, which great limits the scope and advantage of the proposed method.
Does your algorithm work for a more general class of mixing ops?

The experiment using the synthesis data looks great but as a paper aims at a very practical setting, could you show experiment on less-toyish benchmark？

**Questions:**

See above

**Limitations:**

See above

---

> ### Author Rebuttal · Authors · 2023-08-10
>
> Thank you for your valuable comments! We will try to address your concerns as follows.
>
> **Limitation of weight sum based mixing method**
> In multi-source learning, weighted summing source domains and learning on it is the most popular and natural method [1,2], due to its simple form, easy implementability and excellent performance. We agree that there are other methods to share the knowledges of source domains, but it is beyond the scope of this paper. We will consider it as a promising future work. Thank you for the suggestion!
>
>
> **Experiments on less-toyish practical cases**
> We discuss this issue in Global comment. We have already reported results on real data sets in Appendix C of main paper. We apologize for the inconvenience. Moreover, we provide new experiments on CIFAR10 in the attached PDF.
>
> [1] Konstantinov, Nikola, and Christoph Lampert. "Robust learning from untrusted sources." International conference on machine learning. PMLR, 2019.
> [2] Mansour, Yishay, et al. "A theory of multiple-source adaptation with limited target labeled data." International Conference on Artificial Intelligence and Statistics. PMLR, 2021.

---

> > ### Comment · Reviewer_x8t7 · 2023-08-18
> > **Thanks**
> >
> > Thanks the author for the rebuttal. My concern is partially addressed and I increase my score accordingly.

---

> > > ### Author Response · Authors · 2023-08-18
> > > **Thank you for your feedback**
> > >
> > > Thank you for your positive feedback and we are glad that our response addressed your concerns! We are also more than happy to engage in further discussion if you have any remaining concerns.

---

### Official Review · Reviewer_S5TK · 2023-07-22

**Soundness:** 4 excellent
**Presentation:** 3 good
**Contribution:** 4 excellent
**Rating:** 7
**Confidence:** 3

**Summary:**

This paper proposes an optimization strategy for federated learning (FL). A key challenge of FL is data heterogeneity, where each client device has different data distributions. Unlike a naive FL, which minimizes the empirical risk simply weighted over clients (called WERM), the proposed PERM first minimizes the empirical risk for each client individually. To mitigate the scalability problem in PERM, this paper proposes a practical algorithm using model shuffling. Experiments show better accuracy and loss compared to other personalized FL methods.

**Strengths:**

+ PERM seems theoretically sound, achieving the optimal solution for each client. The performance seems to be better than representative personalized FL methods.

+ In addition to data heterogeneity, the proposed method can be easily extended to heterogeneous devices and models since PERM minimizes the loss of each client individually.

+ This paper explains the theoretical aspects very clearly. Especially Section 2, which describes the key theoretical part, is carefully written and quite easy to understand, even for readers a bit outside the field (like me).

**Weaknesses:**

- The theoretical/practical merit compared to other personalized FL methods is not clearly mentioned (although experimental results show that the proposed method achieves better performance compared to representative methods of personalized FL.)
Especially, I am curious about the scalability of the proposed method. In the introductions, I understand that existing personalized methods lack scalability. However, if my understanding is correct, the proposed method also suffers from increasing the number of training data and devices. Rather, PERM theoretically has O(#devices^2) loss functions, as mentioned in the paper, implying that scalability is a significant limitation of PERM.

- The overall description was very clear and easy to read, but I sometimes struggled because of missing definitions and confusing notation. For example:
    * Weight parameter p vs. \alpha might be better for consistency. The definition of \alpha \in \Delta_N does not appear the first time it is used.
    * The procedure of PERM (L133-134 and Eq.(PERM)) was a bit unclear. In general, it was unclear to me how to determine \alpha_i before knowing the result of each client (i.e., h minimizing Eq.(PERM)) until I read the next section.

**Questions:**

Training time compared to other personalized FL methods. This would be an important aspect and might be related to a limitation of the proposed method. Also, I'm curious about the discussion of scalability against the number of clients, compared with the existing methods.

**Limitations:**

As described in the Weakness section, I suspect the proposed method may lack scalability.

---

> ### Author Rebuttal · Authors · 2023-08-10
>
> Thank you for your compliments and valuable comments! We will try to address your concerns as follows.
>
> **The scalability of the proposed method**
>  We discuss the scalability of PERM in comparison to existing methods in Global answer.  To summarize, the compute burden on clients and server is roughly same as existing methods thanks to shuffling (except extra overhead due to estimating mixing parameters which is same as running FedAvg in two-stage approach and an extra communication in interleaved approach).  To evaluate this empirically, we reported the **wall-clock time** of PERM and other methods in Figure.~1 of attached PDF, demonstrating that the run-time is slightly worse due to overhead of estimating mixing parameters.
>
> We are also sorry for the typos and will fix them in the revised version. Thanks for pointing them out!

---

### Official Review · Reviewer_DEoF · 2023-07-31

**Soundness:** 4 excellent
**Presentation:** 4 excellent
**Contribution:** 3 good
**Rating:** 6
**Confidence:** 4

**Summary:**

The paper proposes a novel learning framework for personalizing empirical risk minimization in a given federated learning problem where the data is heterogeneous (non-IID).  The core idea of the paper is to use optimization relaxations and data shuffling to achieve near-optimal solutions for the general NP-hard personalized federated learning formulation. The paper supports the framework's effectiveness by providing generalization bounds for strongly convex losses. Further, it evaluates the performance of the method on synthetic data where each client learns a logistic regression model.

**Strengths:**

- The paper reviews the most important limitations of the current federated learning literature on heterogeneous data, including clustering and personalization ideas.  Further, it demonstrates the main problems of the personalization frameworks (WERM and BERM).

- The theoretical study of the problem when the loss of each client is strongly convex is rigorous, and it offers reliable convergence guarantees for large-scale problems (huge number of data points and limited number of devices)

- The experiment results on the synthetic data are impressive, showing a significant improvement of the local losses compared to the baseline existing approaches.

**Weaknesses:**

- While the paper claims the method can be used for cases where different models have different architectures, the reliance on the strong concavity and smoothness of $f$ is a serious limitation (neural networks with ReLU activation function are not convex nor smooth).

- The experiments are only performed on the synthetic datasets where the underlying model in each client is logistic regression (convex). Therefore, it is not clear that the performance of the proposed algorithm is superior to other baselines if the data is real and the models are non-convex.

- While one of the motivations for relaxing is scalability, there is no experiment on many data points and clients. One suggestion is to report the algorithm's runtime aside from the other approaches in the current experiments.

Based on the above points, it is understandable that the theory might not be easily extendable to non-convex losses. However, it is crucial to add experiments on real data and with many data points on diverse neural network architectures for different clients.

**Questions:**

- What is the current work's main computational and theoretical advantage to [1]? The theoretical assumptions seem the same in both papers. Is there any improvement in the generalization bounds?


[1] Fallah, A., Mokhtari, A., & Ozdaglar, A. (2020). Personalized federated learning: A meta-learning approach. arXiv preprint arXiv:2002.07948.

**Limitations:**

- The main limitation of the paper is the reliance on the strong concavity of the model in both theory and experiments.

- There is no experiment on real datasets where each client can have its unique model or architecture.

- There is no experiment on large-scale datasets where the underlying model is non-smooth and non-convex (neural networks with ReLU activation function).

---

> ### Author Rebuttal · Authors · 2023-08-10
>
> Thank you for your valuable comments! We will try to address your concerns as follows.
>
> **The reliance on the strong concavity and smoothness is a serious limitation**
> Strongly convexity and smoothness are standard assumptions in the proof of shuffling-based optimization algorithm. Our objective is indeed a bi-level optimization: in inner-level we have to optimize $ \alpha $ to get an approximate solution for $ \alpha^*_i$, and then in outer-level we optimize model based on the approximate $ \alpha$. If we remove convexity of $f_i$, then our inner problem will be nonconvex.
>
> Our empirical results (included in Appendix C of main submission), demonstrate the effectiveness of PERM on non-smooth and non-convex objectives as well and we believe extending our theory to establish the convergence without smoothness or strong convexity is an interesting future work.
>
>
> **More experiments (real datasets, model heterogenity, non-smooth, non-convex)**
> As it was mentioned in Global comment, we do have empirical results on EMNIST, which is a real-world federated dataset, and the MLP model, which is a non-convex model. We also provide experiments on CIFAR10 in attached PDF, Figure.2. Due to space limit we only included part of results in main body and reported these results in Appendix C, and apologize for inconvenience.
>
> **Scalibity and reporting the algorithm's runtime**
> Thanks for the comment and suggestion. We discuss the scalability of PERM in comparison to existing methods in Global answer.  To summarize, the compute burden on clients and server is roughly same as existing methods thanks to shuffling (extra overhead due to estimating mixing parameters).
> Per you suggestion, we reported the **wall-clock time** of PERM and other methods in Figure 1 of attached PDF, demonstrating that the run-time is slightly worse  as mentioned above.
>
> **Advantage to meta-learning approach Fallah et al**
> As discussed in Section 2 of main body, Fallah et al fall in the category of BERM, in which they try to learn a global model (meta model) on the average loss, then personalized the
>  (meta)global model to adapt to local distribution. As we discussed in Section 2, when the clients' distribution is extremely heterogeneous, there is no good global model for personalization (please note that knowledge transfer among data sources still happens through the global model which is learned by optimizing vanilla empirical loss). For example, given a $x$, if the first half of clients are with labeling function $y=f(x)$ while the second half with labeling function $y=-f(x)$, the global model learned on the average of their empirical risks cannot have good performance. This is also verified by our experiments, where in high heterogeneous scenarios like synthetic datasets other personalization approaches like PerFedAvg fail. Even in lower heterogeneous cases such as EMNIST, our approach still outperforms PerFedAvg as we effectively estimate how much each data source contributes to training each model based on relatedness of their distributions.
>
> As for generalization of PERM, from algorithm-independent generalization (Theorem 1), PERM can potentially entail optimal generalization (except the error due to  our relaxation to estimate the mixing parameters). We note that while [1] establishes the convergence rate of optimization algorithm of proposed meta-learning method, no generalization guarantees are provided.  We believe understanding the generalization of both methods through the lens of stability  would an interesting future work.

---

> > ### Comment · Reviewer_DEoF · 2023-08-18
> > **Response to Authors**
> >
> > Thanks for the response. I found your explanations on comparing to Fallah's work fulfilling. I think it will be necessary to bring some of the experiments (EMNIST in particular) to the main body in the camera-ready version. Further, it would be nice if you mentioned the discussion you had on the scalability of the model in the limitations of the work.

---

> > > ### Author Response · Authors · 2023-08-18
> > > **Thanks for your feedback**
> > >
> > > We are happy that our answer was convincing. We will definitely revise the paper according to your suggestions. Thank you for your constructive comments again!

---

### Author Rebuttal · Authors · 2023-08-10

We would like to thank all reviewers for your careful and constructive comments. We appreciate the compliments from reviewers on our algorithmic novelty and theory and  will gladly incorporate the suggestions.

**Empirical evaluation on real data** We understand the main concern raised is that our experiments were only conducted on synthetic datasets. Importantly, we direct your attention to additional experiments *already included* in the **Appendix C** of main submission on EMNIST and MNIST using MLP models, which further validate our theoretical results.  We included additional results in an *extremely* non-IID setting in **Figure 2**  in attached PDF on CIFAR10 dataset using a 2-layer CNN model, which further demonstrates the effectiveness of PERM. To further address this concern more comprehensively, we are currently working to supplement our experiments with results on the CIFAR10 dataset using CNN models with varying heterogeneity among data sources and will include in the subsequent version. This will provide further evidence to underscore our theoretical findings.  We appreciate you highlighting this opportunity to strengthen our empirical evaluations.

**Scalability of PERM** Another common question is the scalability of our method. Since in PERM we have to solve $N$ weighted ERM problems to yields $N$ personalized model, naively solution will make client $i$ update $N$ models at the same time. When $N$ is a large number, it is not affordable. Our proposed method only requires each client to update one model at the same time thanks to model shuffling/swapping by server, and the convergence rate and the experimental results show that it can still converge very fast.

To better illustrate this, for simplicity, let us focus on the two-stage method. The first stage is simply running FedAvg to estimate the mixing parameters. In the second stage,  the compute burden of each client is exactly the same as FedAvg that optimizes a single ERM objective (without sampling), with the only difference  that in PERM the model each client receives at each communication round varies. On the server side, the averaging is replaced with model swapping which enjoys the same scalability as existing methods. The only issue would be the required *memory* at the server to maintain mixing parameters for all clients which can be reduced by low-rank approximation or clustering methods (which could be an interesting future work). A piece of empirical evidence to support this is included in the attached PDF (**Figure 1**) which is slightly worse due to the overhead of estimating mixing parameters in parallel to learning personalized models (i.e., two times more communication needed compared to FedAvg and same as  personalized methods such as personalization with interpolation).

---

### Decision · Program_Chairs · 2023-09-21

**Decision:**

Accept (poster)

**Comment:**

This paper studies "personalized ERM" for learning from heterogeneous datasets. Instead of pursuing bilevel optimization proposed in prior work, the authors focus on a new formulation based on the mixing of local losses personalized for each client. The convergence of the proposed algorithm is analyzed and the experiments demonstrate the practicality of the method in certain applications. The reviewers found the topic interesting, the presentation clear, and the contributions solid. The authors' rebuttal addressed the concerns of multiple reviewers. I recommend the acceptance of the paper. In your revision, please include discussions on prior work (e.g. Fallah et al) that was brought up in the rebuttal period. Also, please include additional rebuttal experiments and discussions related to the scalability/limitations of the proposed method.